# Bayesian association scan reveals loci associated with human lifespan and linked biomarkers

Aaron F. McDaid[1,2], Peter K. Joshi[3], Eleonora Porcu[2,4], Andrea Komljenovic[2,5], Hao Li[6], Vincenzo Sorrentino[6], Maria Litovchenko[2,7], Roel P.J. Bevers[2,7], Sina Rüeger[1,2], Alexandre Reymond[4], Murielle Bochud[1], Bart Deplancke[2,7], Robert W. Williams[8], Marc Robinson-Rechavi[2,5], Fred Paccaud[1], Valentin Rousson[1], Johan Auwerx[6], James F. Wilson[3,9] & Zoltán Kutalik[1,2]

The enormous variation in human lifespan is in part due to a myriad of sequence variants, only a few of which have been revealed to date. Since many life-shortening events are related to diseases, we developed a Mendelian randomization-based method combining 58 disease-related GWA studies to derive longevity priors for all HapMap SNPs. A Bayesian association scan, informed by these priors, for parental age of death in the UK Biobank study ($n = 116,279$) revealed 16 independent SNPs with significant Bayes factor at a 5% false discovery rate (FDR). Eleven of them replicate (5% FDR) in five independent longevity studies combined; all but three are depleted of the life-shortening alleles in older Biobank participants. Further analysis revealed that brain expression levels of nearby genes (*RBM6, SULT1A1* and *CHRNA5*) might be causally implicated in longevity. Gene expression and caloric restriction experiments in model organisms confirm the conserved role for *RBM6* and *SULT1A1* in modulating lifespan.

[1] Institute of Social and Preventive Medicine (IUMSP), Lausanne University Hospital, Lausanne 1010, Switzerland. [2] Swiss Institute of Bioinformatics, Lausanne 1015, Switzerland. [3] Centre for Global Health Research, Usher Institute for Population Health Sciences and Informatics, University of Edinburgh, Teviot Place, Edinburgh EH8 9AG, Scotland. [4] Center for Integrative Genomics, University of Lausanne, Lausanne 1015, Switzerland. [5] Department of Ecology and Evolution, University of Lausanne, Lausanne 1015, Switzerland. [6] Laboratory of Integrative and Systems Physiology, Institute of Bioengineering, Ecole Polytechnique Fédérale de Lausanne (EPFL), Lausanne 1015, Switzerland. [7] Laboratory of Systems Biology and Genetics, Institute of Bioengineering, Ecole Polytechnique Fédérale de Lausanne and Swiss Institute of Bioinformatics, Lausanne 1015, Switzerland. [8] Department of Genetics, Genomics and Informatics, University of Tennessee Health Science Center, Memphis, Tennessee 38163, USA. [9] MRC Human Genetics Unit, Institute of Genetics and Molecular Medicine, University of Edinburgh, Western General Hospital, Crewe Road, Edinburgh EH4 2XU, Scotland. Correspondence and requests for materials should be addressed to Z.K. (email: zoltan.kutalik@unil.ch).

Human lifespan is a highly variable and heterogeneous trait[1]. Until the 19th century average life expectancy was ~30–40 years[2]. The sharp rise in life expectancy over the past century is due primarily to improved housing, nutrition, hygiene and vaccines[3], which greatly enhanced the chances to survive childhood and young adulthood[4].

While most determinants of life expectancy are still environmental (for example, socio-economic status, smoking and other lifestyle factors and gender[1]), numerous twin studies[5] suggest that in modern societies 20–30% of human lifespan variation is due to genetic factors, perhaps up to 40% when looking at survival beyond 85–100 years, often termed longevity[1,6,7]. Longer lifespan is strongly associated with later disease onset and as such is likely influenced by many genetic variants via predisposition to, or protection from certain diseases[5], or their risk factors.

Genome-wide association studies (GWAS) have been instrumental in revealing disease susceptibility[8] and longevity-associated loci. However, these studies have so far revealed only a handful of robustly replicating genetic variants. Most studies have focussed on an extreme case–control design, where 'cases' were defined as individuals surviving past 90 years of age and controls were those who died at earlier ages, typically before 80. The bottleneck in these studies is the availability of genotyped samples from nonagenarians, however, increasing sample sizes from $n = 1,836$ (ref. 9; at which no variants were significant) to $n = 6,036$ (ref. 10) and $n = 7,729$ (ref. 11), revealed first the TOMM40/APOE/APOC1 (rs4420638) and FOXO3 (rs10457180) loci, although not at genome-wide significance level. A meta-analysis of European longevity cohorts confirmed APOE (rs4420638, $P < 3.4 \times 10^{-36}$) and revealed a new locus near EBF1 (rs2149954, $P = 1.74 \times 10^{-8}$)[11]. Another approach to identify lifespan-associated loci uses survival analysis, however an early genome-wide study of 25,007 individuals, of whom 8,444 died during the follow-up period failed to identify any significant findings, or overlaps with previous studies[12].

Two recent studies[13,14] analysed parental longevity in the UK Biobank[15]: both highlighted a disease locus, CHRNA3/5 ($P < 10^{-16}$), known to associate with lung cancer linked to smoking[16]. Presumably due to the more powerful method, using information from all parents, alive or dead, in a Cox-regression framework[14], both detected the well-known APOE region ($P < 10^{-19}$), also associated with Alzheimer's disease[17], whilst neither study replicated FOXO3 or EBF1. These findings suggest disease-associated loci may serve as good candidates for lifespan studies. Indeed, a recent GWAS on extreme longevity[18] proposed a weighted hypothesis-testing approach, in which single nucleotide polymorphisms (SNPs) are up-weighted if their association $P$ value combined over 14 disease-related traits is lower. This approach enabled the discovery of three novel loci implicated in extreme longevity.

Here we improve and extend these approaches using: (i) a Bayesian approach to derive a meaningful prior effect size for each SNP; (ii) estimating and accounting for all effect directions (that is, the SNP-phenotype effect sizes and the trait-lifespan causal effect sizes); (iii) analysing lifespan in a general population cohort of the UK Biobank study[14] rather than extreme longevity; (iv) looking up the 16 top associations in four additional longevity studies[10–12,18]; (v) performing extensive bio-informatics follow-up and transcriptomic experiments in mice.

The key aims of our study are to (i) discover new variants impacting mortality and doing this by developing a mechanistic model to estimate morbidity priors for each SNP; (ii) understand through which disease-predisposition they act; (iii) reveal the discrepancy between the life-shortening effect of a SNP and its effect on various life-shortening diseases; (iv) shed light on transcriptome biomarkers that may be causally involved in this process; (v) elucidate whether these processes are shared between mice and human.

## Results

**Methods overview**. We used parental lifespan-association summary statistics (Z-statistics of the linear regression) from the UK Biobank study[14]. For simplicity, in the following, we will use the term lifespan to refer to parental lifespan.

For each SNP, we compute a novel test statistic, based on a Bayes factor (BF). With simulations, we draw a large sample of how these statistics would be distributed if the null hypothesis were true for every SNP, that is, no effect of any SNP on lifespan. This allows us to compute a $P$ value for each observed test statistic, comparing the observed test statistic to the large sample of 'null' test statistics. In this framework, the goal is to compute a novel test statistic which is more powerful than the conventional lifespan Z-statistic of $\frac{\widehat{\beta}}{\sqrt{\mathrm{Var}(\widehat{\beta})}}$. Given two point hypotheses, the likelihood ratio test is the most powerful test[19]. We use a BF as the likelihood ratio. For each SNP, the two hypotheses of interest are the null hypothesis of no effect on lifespan and an alternative hypothesis, which we refer to as a *prior*. The prior is a prediction of the lifespan effect size, based on the effect sizes of the same SNP for other disease-related traits and an (out-of-sample) estimate of the causal effect of these traits on lifespan. The steps taken, in order, are: (1) Impute Z-statistics for all SNPs of interest in lifespan and in other traits of interest. (2) Estimate the causal effects of traits on lifespan using Mendelian randomization (MR). (3) Build the prior for each SNP. (4) Compute the BFs for each SNP. (5) Sample the 'null' BFs. (6) Compute $P$ values by comparing the observed BFs to the 'null' BFs. All these steps are explained in details in the 'Methods section'.

The first step is performed with a standard summary statistic imputation method[20]. To tackle step 2, we first collected a large compendium of 58 eligible association studies representing 36 distinct traits potentially impacting lifespan (Supplementary Table 1). These association scores were subjected to multivariate, out-of-sample MR analysis to estimate the causal effect of these traits on parental lifespan (Methods section). In step 3, we then defined the per-trait prior effect of each SNP as the product of its effect on a trait and the causal effect of this trait on lifespan. These per-trait priors were summed up to yield a prior effect estimate for each SNP on lifespan. Steps 2&3 are illustrated in Fig. 1.

Next, the derived priors were combined with the observed longevity association statistics to calculate the BF. The final step was to compare the obtained BFs to the distribution expected under the null hypothesis, allowing us to compute empirical $P$ values control the false discovery rate (FDR) regardless of the correctness of the priors. Aside from these permutation-based $P$ values, all $P$ values in this paper are based on a normally distributed test statistic and a null hypothesis of no effect of the SNP (or trait) on lifespan. The GWAS data we used for non-lifespan traits includes $P$ values; from these $P$ values and the effect direction we inferred the Z-statistics.

**Possible causal effects of GWAS traits on parental lifespan.** The instrument (SNPs) selection, a key component of any MR analysis, depends on the association $P$ value threshold and strength of linkage disequilibrium (LD) pruning. We have explored multiple settings (Supplementary Fig. 1) and, as a compromise, chose intermediate instrument strength ($P < 10^{-5}$) and moderate LD pruning (LD < 0.2). The multivariate MR

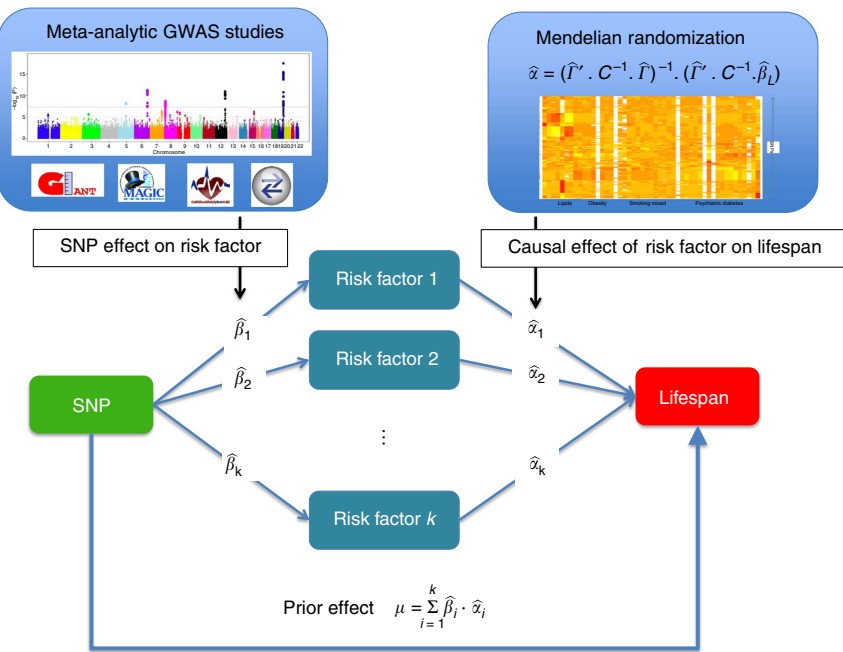

**Figure 1 | Analysis steps to obtain longevity prior effects for Bayesian analysis.** For each SNP its prior effect on lifespan is calculated as the product of the effect of the SNP $i$ on GWAS traits (risk factors) $[\hat{\Gamma}_{i,t}]$ and the causal effect of the trait $t$ on lifespan $[\hat{b}_t]$, summed over all available $T$ GWAS traits. The causal effects of the traits on lifespan were calculated via a leave-one-chromosome out multivariate Mendelian randomization.

**Table 1 | Multivariate causal effect estimates for the 11 traits chosen by the AIC-based stepwise model selection.**

| Trait | Effect size | s.e. | P value |
|---|---|---|---|
| Education level | 0.1810 | 0.0144 | 5.40E-36 |
| Cholesterol LDL | −0.0587 | 0.0050 | 5.90E-32 |
| BMI | −0.0958 | 0.0132 | 3.73E-13 |
| Smoking | −0.1575 | 0.0248 | 2.14E-10 |
| Coronary artery disease | −0.0934 | 0.0166 | 1.77E-08 |
| Type 2 diabetes | −0.0716 | 0.0158 | 5.9562E-06 |
| Schizophrenia | −0.0196 | 0.0068 | 0.0039 |
| Cholesterol HDL | 0.0223 | 0.0078 | 0.0041 |
| Height | −0.0131 | 0.0045 | 0.0041 |
| Triglycerides | −0.0240 | 0.0089 | 0.0071 |
| Glucose | −0.0433 | 0.0165 | 0.0086 |

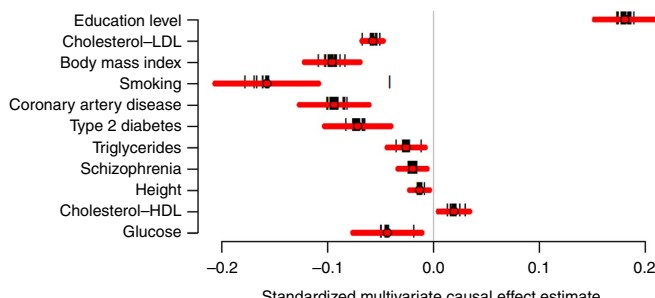

**Figure 2 | Multivariate MR causal effect estimates and 95% confidence intervals of the 11 significant traits on lifespan.** Effects are standardized such that they correspond to the square-root of the variance explained. In other terms, for example, 1 SD increase in BMI leads to 0.09 SD reduction in lifespan. Each black vertical bar represents the causal effect estimates obtained when leaving one chromosome in the estimation (Methods section).

identified 11 traits with significant causal effects on lifespan (Table 1 and Fig. 2). Note that the estimates presented here are multivariate, which is necessary to create priors that avoid double counting the effect of pleiotropic SNPs. In univariate MR, many other traits show significant univariate causal effect (Supplementary Table 1). The 11 traits were selected based on Akaike Information Criterion (AIC) after removing any traits with multivariate $P$ value > 0.05. The complete model explained 7.7% of the variability of lifespan effect sizes. Most causal effect directions were in line with the literature and mechanistic explanations: having higher education level, higher high-density lipoprotein (HDL)-cholesterol level and shorter stature improves lifespan; but higher body mass index (BMI), low-density lipoprotein (LDL)-cholesterol, triglyceride or glucose levels, susceptibility to coronary artery disease, major depression, type 2 diabetes (T2D) or schizophrenia all shorten lifespan. Smoking intensity (measured as number of cigarettes per day) showed a clear negative impact on lifespan. Less intuitively, the link between tall stature and shorter lifespan has been shown in several studies, some explaining the link by cancer risk[21].

Only one cancer study (neuroblastoma) passed our inclusion criteria, hence due to the weakness of the available instruments the impact of most cancers could not be investigated here. In order to provide further evidence that these causal estimates yield robust priors, we performed MR using instruments only from chromosomes 1 to 6 and another MR analysis with instruments only from chromosomes 7 to 22. Although the selected traits did not perfectly agree (Supplementary Table 2), both the causal effect sizes and the resulting priors were highly concordant (Supplementary Fig. 2). In addition, we examined the MR residuals to ensure that there was no instrument with indication to violate the third MR assumption, that is, that the instrument must be independent of disease-lifespan confounders. If such violation occurs, the instrument would have a disease-independent (pleiotropic) impact on lifespan and hence the observed SNP-lifespan effect would deviate substantially from the prior effect. We observed only a slight deviation for one

instrument (a SNP near *APOE*, rs7256200) on the residual qq-plot for all the 22 leave-one-chromosome-out estimation (Supplementary Fig. 3). Due to the large number of instruments, excluding this single SNP does not change noticeably the results.

While many of the obtained causal effect estimates seem plausible and in line with literature evidence, we do not claim that these causal effect estimates are unbiased, neither that no instruments violated MR assumptions. This step was necessary only to obtain sensible priors for the downstream analysis. If these estimates are wrong, our prior is less informative, yielding fewer discoveries.

**Computing prior effect estimates.** In total 77,963 SNPs had association *P* value $< 10^{-5}$ with at least one of the 11 traits identified in the previous step to be causally linked to lifespan. For each of these SNPs we calculated the product of its effect on a given trait (set to zero if the association *P* value $> 10^{-5}$) and the estimated causal effect of the trait on lifespan. These products were then summed up for all the 11 traits and used as prior effect estimate of the SNP on lifespan. For illustration we show how the effect sizes of these 78 K SNPs for four important traits compare to their effect on lifespan (Supplementary Fig. 4). Note that the agreement is far from perfect, but when combined together into a prior it correlates well with the estimated effect on lifespan in the UK Biobank study (Supplementary Fig. 5). We also computed priors for the remainder of the 2.3 million HapMap SNPs for which we imputed lifespan-association Z-statistics and effect estimates—their prior distribution was set to have zero mean as none of them were significant at $10^{-5}$ for any of the 11 traits.

**Loci surviving 5% false discovery rate.** Combining the priors with the observed longevity summary statistics we obtained BF for all 2.3 million SNPs and generated 1000 null GWAS associations to derive null BFs. The generated 2.3 billion (1,000 permutations × 2,350,352 SNPs) null BFs were used to assign *P* values to each observed BF. The Manhattan plot highlights the 16 associated loci surviving 5% FDR (Fig. 3). For each locus we selected the SNP with the largest BF, the resulting 16 SNPs are completely independent of each other and described in Table 2.

Conditional analysis performed at each locus revealed no secondary association signal beyond the top hit. In the QQ-plot, the permutation-based BF *P* values show good adherence to the null distribution for ∼99% of the selected SNPs with genomic lambda of 1.002 (Supplementary Fig. 6). If we use only the 77,963 SNPs with prior estimate different from zero, reveals a convincing enrichment of low lifespan-association *P* values (Supplementary Fig. 7). If one wishes to control FWER, 4 SNPs (in/near *LPA, CDKN2BAS, CHRNA5* and *APOE/APOC1*) survive the more stringent Bonferroni threshold ($5 \times 10^{-8}$).

Details of these top hits are shown in Table 2. Remarkably, 14 out of the 16 SNPs lie within 1 kb of a coding sequence, but none of them are coding variants. Since these SNPs were discovered due to their prior association with various diseases, we also checked which diseases these SNPs are already known to be associated with. Most of these SNPs (or their LD-proxies) have been previously associated with clinically important traits, most frequently: LDL-, triglyceride-, total cholesterol-, education levels, BMI, height and coronary artery disease (Table 2). Interestingly, an intronic SNP of the BMI-associated *FTO* gene also features in the list of 16 top hits. However, three of them (rs2909448 in *DPP4*, rs362296 in *MSANTD1*, rs729583 in *SNX29*) have not yet been associated with any trait at genome-wide significant level.

In Table 2 we also report the hazard ratio (HR) for survival beyond age 40 per allelic dosage in offspring for the 16 top hits. The HR was calculated directly in UK Biobank for the genetically British with valid mother/father survival information and maternal and paternal effects were combined. Using only dead parents we also estimated the effect of these SNPs in term of years of life lost per allele. For example, the strongest *APOE/APOC1* (rs4420638) variant has HR = 1.038 and each copy of the G allele decreases life by ∼5 months.

**Lookup of the 16 loci in other longevity studies.** Given the large sample size of the UK Biobank compared to other previous independent studies we first assessed power to replicate our findings in the combined CHARGE studies. First, we performed power calculations assuming a replication study with 14,000 samples selected from the top and bottom 5% tail of a general

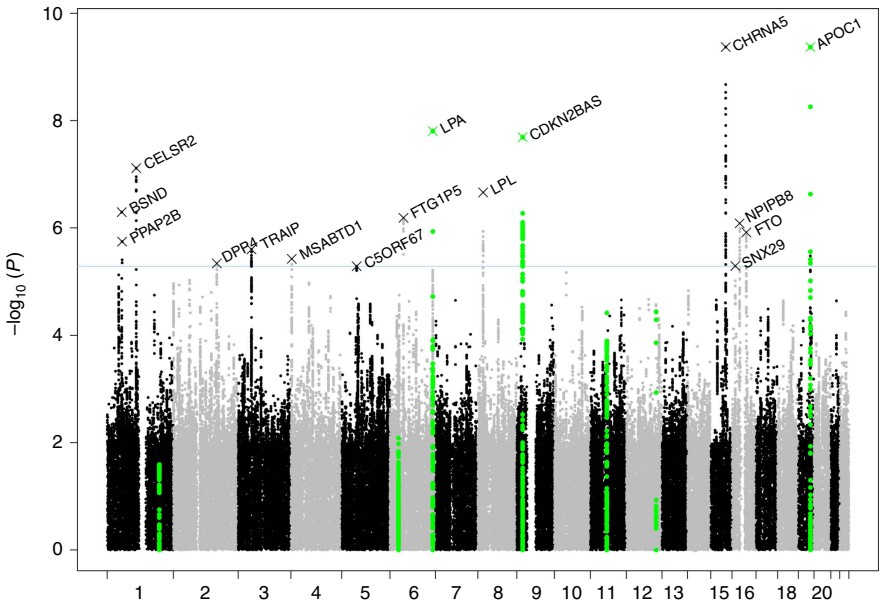

**Figure 3 | Manhattan plot of the permutation *P* values of the BF.** The nearest genes to the 16 significant loci are indicated next to the lead SNP. Regions implicated in a recent longevity study[18] are highlighted in green. X-axis represents the chromosome number and the physical position within each chromosome.

**Table 2 | List of the 16 lifespan-associated SNPs at 5% FDR level.**

| Chr | Position | Rs number | A1 | A2 | EAF | Gene | Category | Prior | s.e. prior | UKBB Z-stat | Log BF | P (BF) | UKBB HR | UKBB effect (month) | P rep | Trait GWAS | | |
|---|---|---|---|---|---|---|---|---|---|---|---|---|---|---|---|---|---|---|
| | | | | | | | | | | | | | | | | Study | Z-stat | Study P |
| 1 | 55487346 | rs12117661 | G | C | 0.24 | BSND | Intergenic | 0.552 | 0.362 | 4.792 | 3.471 | 5.47E-07 | 0.981 | 5.6 | 0.003 | LDL | − 9.13 | 7.00E-20 |
| 1 | 56962821 | rs17114036 | G | A | 0.09 | PPAP2B | Intron | 0.796 | 0.384 | 3.407 | 2.765 | 1.80E-06 | 0.978 | 1.8 | 0.325 | CARDIOGRAM | − 5.67 | 1.43E-08 |
| 1 | 109818530 | rs646776 | T | C | 0.78 | CELSR2 | nearGene-3 | − 2.921 | 0.436 | − 3.119 | 4.761 | 6.97E-08 | 1.015 | − 1.1 | 0.001 | LDL | 35.27 | 1.63E-272 |
| 2 | 162866255 | rs2909448 | T | C | 0.55 | DPP4 | Intron | 0.559 | 0.363 | 3.602 | 2.335 | 4.63E-06 | 0.987 | 3.3 | 0.252 | EDU_2016 | 4.86 | 1.20E-06 |
| 3 | 49890613 | rs2352974 | T | C | 0.49 | TRAIP | Intron | − 0.944 | 0.368 | − 3.018 | 2.597 | 2.53E-06 | 1.013 | − 3.2 | 0.012 | EDU_2016 | − 8.51 | 1.69E-17 |
| 4 | 3247007 | rs362296 | A | C | 0.38 | MSANTD1 | UTR-5 | 0.551 | 0.359 | 3.729 | 2.419 | 3.80E-06 | 0.987 | 4.8 | 0.093 | EDU_2016 | 4.95 | 7.45E-07 |
| 5 | 55860866 | rs3936510 | T | G | 0.2 | C5ORF67 | Intron | − 0.550 | 0.375 | − 3.547 | 2.286 | 5.19E-06 | 1.016 | − 3.4 | 0.003 | TG | 7.73 | 1.09E-14 |
| 6 | 50921602 | rs6904450 | T | A | 0.18 | FTH1P5 | Intergenic | − 0.654 | 0.366 | − 4.331 | 3.353 | 6.45E-07 | 1.019 | − 6.9 | 0.006 | BMI_2015 | 10.96 | 6.25E-28 |
| 6 | 161010118 | rs10455872 | G | A | 0.08 | LPA | Intron | − 1.340 | 0.398 | − 4.465 | 5.680 | 1.60E-08 | 1.034 | − 6.1 | 7E-05 | TC | 8.34 | 7.24E-17 |
| 8 | 19858499 | rs1581675 | A | T | 0.26 | LPL | Intergenic | 1.145 | 0.455 | 3.749 | 4.123 | 2.11E-07 | 0.984 | 3.0 | 0.167 | TG | − 27.20 | 7.40E-163 |
| 9 | 22119195 | rs1333045 | C | T | 0.51 | CDKN2BAS | Intron | − 1.430 | 0.495 | − 4.155 | 5.540 | 1.77E-08 | 1.014 | − 3.7 | 3E-05 | CARDIOGRAM | 11.79 | 4.63E-32 |
| 15 | 78878541 | rs951266 | A | G | 0.33 | CHRNA5 | Intron | − 0.978 | 0.888 | − 6.739 | 13.142 | 4.33E-10 | 1.028 | − 5.7 | 0.002 | Smoking | 12.01 | 3.00E-33 |
| 16 | 12532918 | rs729583 | G | C | 0.61 | SNX29 | Intron | − 0.503 | 0.362 | − 3.726 | 2.289 | 5.15E-06 | 1.014 | − 3.7 | 0.361 | EDU_2016 | − 4.43 | 9.35E-06 |
| 16 | 28825605 | rs2008514 | A | G | 0.41 | NPIPB8 | Intron | − 1.342 | 0.378 | − 2.970 | 3.184 | 8.22E-07 | 1.011 | − 1.1 | 0.003 | BMI_2015 | 10.00 | 1.45E-23 |
| 16 | 53800568 | rs9939973 | A | G | 0.42 | FTO | Intron | − 2.477 | 0.463 | − 2.470 | 2.953 | 1.22E-06 | 1.010 | − 1.1 | 2E-04 | BMI_2015 | 25.58 | 2.62E-144 |
| 19 | 45422946 | rs4420638 | G | A | 0.19 | APOC1 | nearGene-3 | − 1.567 | 0.422 | − 7.340 | 12.706 | 4.33E-10 | 1.038 | − 4.8 | <2E-06 | LDL | 28.49 | 1.51E-178 |

Columns represent: chromosome, position (hg19), rs number, effect allele (A1), other allele (A2), effect allele frequency (EAF), nearest gene, SNP category, prior effect size, s.e. of the prior effect, Z-statistic in the UK Biobank study, log BF, permutation P value of the log BF, Cox-regression HR in UK Biobank, months lost/allele effect in UK Biobank, replication P value (5 studies combined), GWAS that shows the strongest association with the SNP [CARDIOGRAM: coronary artery disease[60], smoking (cigarette per day)[61], LDL[62], BMI_2015[63], EDU_2016[44]], GWAS Z-statistic, GWAS P value.

lifespan distribution and a SNP explaining ∼1% of lifespan variability. These calculations showed that apart from the top two hits (APOE and CHRNA3/5) we have only 16% (for the FTO variant) to 73% (for the SNP near BSND) power to replicate our findings with a P value below 0.05/16 in a study very similar to the combined CHARGE data set. For this reason we call our analysis as a lookup, given that strict replication is impossible due to power considerations.

The 16 significant SNPs were tested against summary statistics from all available replication studies: the CHARGE study[10], the EU longevity[11], a genome-wide survival study[12] and a recent disease-informed extreme longevity GWAS[18] including two cohorts. We combined evidence at the P value level and meta-analysed P values using the Fisher's method[22] while accounting for study overlap (Methods section). Notably, 11 of the 16 SNPs can be declared significant based on its combined replication P value at 5% FDR (Table 2 and Supplementary Table 3).

Some of these 16 SNPs (near APOE/APOC1, LPA and FTO) seemed to have larger effects on extreme longevity (Supplementary Fig. 8). There are several explanations for this: (i) Markers related to late onset diseases (such as Alzheimer's) are bound to be less pronounced effect earlier. (ii) Inflated extreme longevity effects can also be put down to the fact that testing the extremes of a distribution versus the whole distribution by definition yields larger effects. (iii) Moreover, the UK Biobank study used offspring genotypes as proxy for parental genotypes, which results in estimates biased towards zero. On the other hand, we also observe a handful of SNPs (for example, in CHRNA5, LPL) impacting lifespan only in the normal range, but with negligible effect on extreme lifespan.

In summary, 12 out of the 16 hits replicate in at least one of the replication studies with nominal significant one-sided P value and 9 of these (in/near BSND, CELSR2, FTH1P5, LPA, CDKN2BAS, CHRNA5, NPIPB8, FTO and APOE/APOC1) survive Bonferroni correction (0.05/16).

Finally, we verified that 13 out of the 16 life-shortening alleles are depleted in older study participants of the UK Biobank (P = 0.0021) and five of them have nominally significant association with participant age (P = 8.09 × 10$^{-5}$). This observation is concordant with the selection bias of study participants: those that are able to make it to population cohort studies are in better health than the overall population. Hence, people carrying life-shortening (or disease-associated) alleles are less likely to make it to a cohort at an older age resulting in lower life-shortening allele frequency as a function of participant age in the UK Biobank (Fig. 4).

**Replication of previous associations.** As expected, the well-established APOE locus is convincingly confirmed in our study, being the top associated locus. The lifespan-shortening FOXO3 variant (rs10457180-A) did not replicate in our study (P = 0.41). The longevity-associated EBF1 variant (rs2149954-T) found in the largest European GWAS[11] in the UK Biobank analysis did not reach nominal significance (imputed P value = 0.16) and was assigned zero prior in our analysis.

A large study on the genetics of survival[12] put forward 14 candidate SNPs that reached a mild cutoff of P < 10$^{-5}$. None of these SNPs attained a nominally significant P value in our study (Supplementary Table 4), moreover the effect directions agreed only for five out of 12 SNPs we could impute.

A recent extreme longevity study[18] found four novel associated protective alleles near CDKN2B/ANRIL (rs4977756-G), SH2B3/ATXN2 (rs3184504-G), ABO (rs514659-A) and HLA (rs3763305-A). Remarkably, the first two variants replicated in our analysis with (one-sided) P values P = 4.34 × 10$^{-6}$, P = 1.08 × 10$^{-3}$, P = 0.03, P = 0.51, respectively.

**Gene expression and methylation as marker for biological age.** We applied MR to test whether the expression of certain genes at the 16 lifespan-associated loci may be causally driving lifespan. The rationale behind MR is that if a gene expression in a given tissue is causally related to lifespan, SNPs modulating its expression must have an effect on lifespan proportional to the effect on the expression. We investigated 91 gene-tissue pairs (equivalent to 57 independent tests) at the 16 loci that had more than five independent eQTL SNPs in a GTEx (ref. 23) tissue to be sure that the signal is not driven by pleiotropy. Three genes (SULT1A1, CHRNA5 and RBM6) showed significant negative causal effects (all in brain) on lifespan, that is, lower expression extending lifespan (Supplementary Table 5).

If the expression of a gene correlates with chronological age, it may be an indicator for the biological age and may be linked to lifespan[24]. This hypothesis was formally confirmed for age-associated methylation levels[25]. If the expression of age-correlated genes tends to modulate lifespan, SNPs regulating their expression may influence lifespan through the altered expression. To test this hypothesis, we explored whether SNPs that at least mildly regulate peripheral blood expression level[26] (P < 10$^{-4}$) of age-associated genes[24] are enriched for lower than expected longevity P values. This set of SNPs, however, showed no significant enrichment in low P values. Similarly, we found no significant enrichment of lower than expected lifespan P values

among SNPs that are methylation QTLs[27,28] ($P < 10^{-4}$) for at least one of the 353 age-associated methylation CpG sites[29]. These results indicate that many of the age-correlated biomarkers may simply change in response to aging or are driven by mechanisms that underlie aging.

**Linking expression level and lifespan in model organisms.** In this section we focus on the three genes (*SULT1A1, CHRNA5* and *RBM6*) that emerged as strong candidates, and analysed whether their expression levels (in brain) could be used as an early bio-marker for longevity. Specifically, we looked for evidence whether their expression or the suppression thereof may be linked to lifespan in animal models.

We compared the lifespan and the expression level of these focal genes in prefrontal cortex at 72 days of age in 35 strains of LXS mouse lines[30]. Remarkably, lower messenger RNA level of *RBM6* in prefrontal cortex at 72 days was strongly associated with lifespan in ($P = 4.1 \times 10^{-4}$, Supplementary Table 6 and Supplementary Fig. 9). The significant correlation persisted even after correcting for population structure through mixed effect models[31] ($P = 1.1 \times 10^{-4}$). The other two genes showed no significant association with lifespan.

Caloric restriction (CR), defined as a reduction of ∼20–40% of the regular caloric intake, can extend lifespan in a large range of organisms (*C. elegans*[32], *D. melanogaster*[33], mice[34], rhesus monkeys[35]). In humans, the evidence is less direct and based on the improvements of primary and secondary aging traits (fasting insulin level, body temperature, metabolic rates, inflammation markers and elasticity of the cardiac left ventricle)[36,37]. We therefore asked whether CR in animal models induces expression change in any of the three focal genes. Notably, *SULT1A1* expression levels are up-regulated upon CR diet in mouse (GSE11291), (Supplementary Fig. 10). No significant difference was found for the other genes, neither in fly (GSE26726) nor in worm (GSE27677).

**Genetics of lifespan versus extreme longevity.** One key question is whether there are specific genetic factors associated with extreme longevity[10,11], but not general lifespan[14]. One can imagine that some genetic factors (for example, *APOEε4*) trigger diseases with very late onset (such as Alzheimer's), hence these markers would not impact lifespan below 70 years of age. To check to what extent the genomic landscape of lifespan and extreme longevity overlap, we applied LD-score regression[38]. Despite these studies do not share top hits, the analysis comparing association summary statistics genome-wide revealed very strong genetic correlation ($r_G = 0.73$, s.e. = 0.11) between these two kinds of longevity measures, indicating mostly shared genetic mechanisms.

**Discussion**

In this work, we present a Bayesian approach to detect novel associations between genetic markers and lifespan. We have demonstrated that genome-wide disease GWAS results combined with Mendelian randomization yield very informative prior effect estimates for lifespan association. When using non-informative priors only two loci could be identified[14], but our approach revealed 16 loci associated with lifespan at 5% FDR. Moreover, if we were to use the standard association $P$ values, but reduce multiple testing burden to only 4% of the genome that had non-zero prior assigned, still only three of the 16 loci would have been recovered at 5% FDR (Supplementary Table 7).

One could argue that any SNP linked to a trait causally affecting lifespan will eventually be associated with lifespan. Our study points far beyond this basic statement: First, we provide

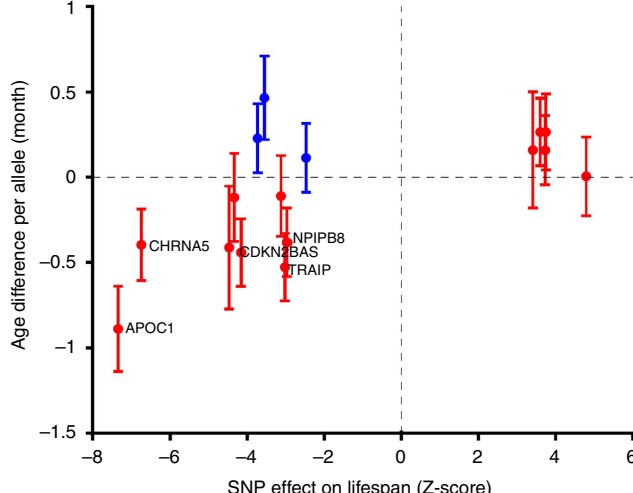

**Figure 4 | The frequency of life-shortening alleles decreases with the increasing age of study participants for 13 out of the 16 SNPs.** Each dot represents a SNP, with x-coordinate marking the Z-statistic in the lifespan association study and y-coordinate the age difference per allele (in month) with 95% confidence interval. SNPs whose effect direction agrees in the two studies are in red, others in blue. The nearest genes of the five SNPs with age-association $P < 0.05$ are indicated.

effect size estimates of the identified SNPs on various scales. Second, three of our top hits are not associated at genome-wide significant level with any individual disease and only two of them are top hit for some disease. Third, most of the 16 loci exhibit pleiotropic effects (Fig. 5): For example, the *APOE/APOC1* variant (rs4420638) decreases lifespan primarily through (beyond the well-known effect on Alzheimer predisposition) increasing LDL levels, but surprisingly protects from type 2 diabetes. The *CHRNA3/5* SNP (rs951266) mainly impacts lifespan through smoking and schizophrenia predisposition; the *SNX29* SNP rs729583 seems to have little impact on most of the 11 traits; the *FTO* SNP rs9939973 essentially exerts its effect on BMI and its downstream traits. Fourth, the actual effect size estimates of the discovered SNPs are quite far from both the priors and their individual effect on a trait (multiplied by the impact of the trait on lifespan; Supplementary Figs 4 and 5).

Given the size of all other published longevity GWAS, we had low power to replicate our findings. Despite this fact many of the discovered loci replicated in independent data sets[10,12,18,39]. We performed pathway enrichment analysis by PASCAL[40] using the BF permutation-based $P$ values (Supplementary Table 8). Only the lipoprotein metabolism pathway and lipid digestion, mobilization and transport showed statistically significant enrichment ($P = 3.1 \times 10^{-6}$, $1.5 \times 10^{-5}$, respectively). This is not unexpected, as longevity seems to be extremely complex, impacting multiple diseases and processes.

Surprisingly, the MR analysis did not yield a significant causal effect of alcohol consumption on lifespan, despite epidemiological observations[41]. This may be a false negative result and might be due to the fact that the GWAS on alcohol use did not reveal strong associations, rendering the MR less powerful. The strong link between education level and lifespan has been seen in other studies[42], but its interpretation is not straightforward. Educational attainment may have shared genetics with socio-economic status, which is a more plausible underlying cause. In line with our finding, a recent study demonstrated that genetic risk score for education level is a strong predictor of longevity[42]. Furthermore, some risk factors (lipids, blood pressure, obesity and diabetes) have shown opposite effect on lifespan when

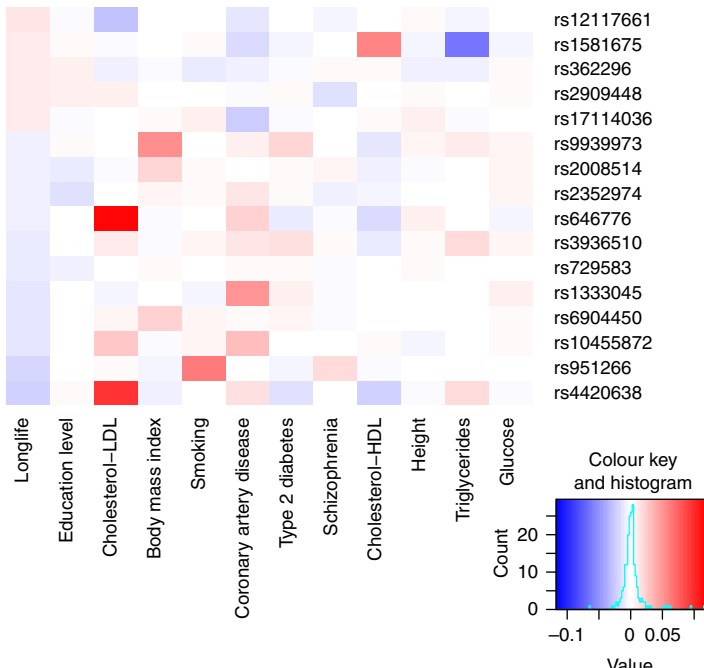

**Figure 5 | Heatmap of the standardized effects of the 16 lifespan-associated SNPs on the 11 lifespan-impacting traits and lifespan.** We plotted the standardized effects of the 16 lifespan-associated SNPs on the 11 traits altering lifespan. Trait-increasing (decreasing) effects are shown in red (blue). Most of these SNPs show extensive pleiotropic effects.

focusing on longevity only beyond 85 years[43], but <15% of UK Biobank parental lifespans fell into this category, hence this phenomenon could only mildly worsen our prior, which was based on younger populations. Some risk factors (for example, BMI) can have non-linear causal relationship with lifespan, which can render our approach less powerful.

The follow-up analyses revealed several novel insights into the genetic underpinnings of lifespan regulation. First, we found that while genetic effects on extreme longevity are generally larger than on general lifespan, their global genetics are largely shared ($r_G = 0.73$). Second, the squared genetic correlation between lifespan and the (causal effect) weighted combination of 11 traits/diseases examined in this study, calculated by LD-score regression[38], is surprisingly low 5.8% (Supplementary Fig. 5), but in line with the one 7.7% lifespan variance explained by the 11 traits in the multivariate MR model. Second, we have also seen that 13 of the 16 lifespan-shortening alleles are rarer in older individuals, a clear fingerprint of selection bias. The small extent of allele depletion (<1 months difference between genotype groups) also suggests that selection bias in general population cohorts would not strongly influence GWAS results. Third, our in-depth eQTL MR analysis revealed three potentially causal genes for lifespan (in brain). Finally, the similarity of the effect of some candidate genes and their expression levels in brain on lifespan and trends in CR in mouse indicate evolutionary conservation.

Our approach has several weaknesses: (i) Since the UK Biobank study is much larger than any other lifespan study ever conducted, meaningful replication is not possible due to these previous studies being massively underpowered for this purpose. (ii) We cannot identify longevity-associated SNPs whose effects are non-disease mediated or mediated by diseases whose genetic background is largely unknown or confounded by environmental factors. (iii) When applying MR we did not strictly verify whether all instruments are valid as our aim was only to obtain a sensible causal effect estimate to define the priors, rather than accurately establishing the true causal effect size. (iv) Sample size of the

available GWASs impacts our ability to detect traits causally affecting lifespan. This limitation precluded us from including most cancer GWASs to build the priors. Another example is educational attainment: when we used the recently published study on educational attainment[44] ($N = 293,723$) in our MR analysis, it revealed a much stronger causal evidence for lifespan than the earlier study[45] ($N = 101,069$). (v) Combining our results with expression QTL analysis does not provide definite evidence of the causal involvement of certain transcripts in lifespan modulation. In addition, tissues with larger sample size have more chance to be picked up than others. For example, in case of the lifespan-associated rs646776 at the 1p13.3 locus (near *CELSR2*) we could not detect a causally implicated mediator gene in any tissue. However, a previous study[46] showed that this variant might exert its effect on LDL cholesterol levels through modulating the expression of *SORT1* in the liver. We conducted MR analysis to confirm this gene as the best candidate for LDL, but not for lifespan: The reason for this is that *SORT1* has a much stronger eQTL SNP in liver (rs7528419), which is not associated with lifespan in our study. Thus unless rs7528419 is an invalid instrument, our observation excludes *SORT1* as causal gene for lifespan, despite being a strong candidate for LDL.

We expect that the growing number of traits subjected to GWASs and their increasing sample size will strengthen our approach to reveal many more longevity-associated SNPs that act through disease modulation. Importantly, our analysis strategy is not specific to longevity and can provide a new statistical framework to improve the power of future GWA studies of any disease with well-studied risk factors.

## Methods

**Methods overview.** A BF is computed for each SNP, compared to the distribution of BF obtained for simulated $P$ values under the null hypothesis of no SNP affecting lifespan. This procedure allows us to control the FDR. To compute the BF for a SNP, we need an alternative distribution of the SNP's effect on lifespan, alongside the conventional null distribution of zero effect. We use a *prior* distribution as our alternative distribution, as presented in Fig. 1. In the remainder of this section,

these steps will be described in more detail. Note that through the whole paper we utilize only association summary statistics, that is, the Z-statistics from the regression (coefficient estimate divided by its standard error), and not actual genetic data, except for the UK Biobank.

**Longevity studies used.** The primary longevity study used is based on 116,279 individuals in the UK Biobank[14]. In brief, the authors use the age-of-death for the parents as measure of longevity. Since not all parents were dead in the study, they calculated the Martingale residuals from a Cox regression to place alive and dead subjects (estimated) age-of-death on the same scale, thus incorporating useful additional information over and above considering the dead alone, especially for long-lived alive parents. Only those identified by UK Biobank[15] as unambiguously genetically British, with valid parent lifespan details and covariates were analysed. Covariates fitted in the model were subjects' (not parental) sex, indicators of assessment center and genotyping batch, as well as the first 15 principal components of the relationship matrix, as provided by the UK Biobank. One GWAS was run for paternal and one for maternal age-of-death. These UK Biobank analyses can be interpreted as a study on the parent's longevity, where we use the child's genome as a proxy for the parent's genome, akin to imputation. In our study we used the combined association results for both parents via inverse-variance weighted fixed effect meta-analysis adjusted for correlated test statistics[47]. The correlation between the maternal and paternal lifespan effect sizes was observed to be 0.11 under the null.

As replication we combined two (almost) independent studies performed by the CHARGE Consortium[10] and the EU Longevity Study[11]. Each study is a case–control analysis, comparing a set of cases that survived to at least 90 years of age to a set of (differently defined) younger controls. We meta-analysed these two studies, via inverse-variance weighted fixed effect meta-analysis adjusted for correlated test statistics[47] due to the fact that there are a small number of individuals that are in both studies, to construct a single GWAS. As further replication, we also looked up association results in the Walter et al.[12] study.

**Disease GWASs used.** To build priors for lifespan, we assembled a compendium of meta-analytic GWASs that are not related to lifespan nor longevity. We began with 207 studies, made up of three groups (Supplementary Table 1). The largest group is 144 studies that were publicly available on the dbGaP ftp site for a wide variety of traits. We considered only studies that had at least 90,000 unique SNPs and which had suitable column names (P value, effect allele, effect size or direction) defined. The direction of the effect ($\pm$), which is sufficient for our method, is defined as the sign of 'beta' fields (for continuous traits) or whether 'odds ratio' fields are greater ($+$) or less ($-$) than 1 (for binary traits).

We also included the 35 publicly available studies from the Psychiatric Genomics Consortium (PGC), and another collection of 36 studies of various consortia: anthropometric (GIANT), lipid (GLGC), metabolic (DIAGRAM, MAGIC), cardiovascular (CARDIOGRAM) and other traits.

Of these studies, 60 demonstrated heritability significantly different from zero ($P < 5e-4$) as estimated by LD-Score regression[48]. Two of these studies were discarded because of strand ambiguity yielding 58 suitable studies for our analysis.

**Imputation of association summary statistics.** To have summary statistics available for the same set of SNPs we performed summary statistic imputation[20]. This imputes Z-statistics (effect size/s.e.) by assuming that the correlation between Z-statistics is the same as the correlation between the corresponding SNPs in a suitable reference panel of genotypes. We will refer to the Z-statistic for SNP $i$ under disease $t$ as $w_{it}$. When we have observed Z-statistics for a set of SNPs M, we can impute an unobserved Z-statistic, $w_{id}$, based on the observed vector,

$$\widehat{w}_{id} = E[w_{id}|\mathbf{w}_{Md}] = \mathbf{c}'_{iM} C_{MM}^{-1} \mathbf{w}_{Md} \tag{1}$$

where $\mathbf{c}$ and $C$ record the SNP–SNP pairwise genetic correlation matrix of the relevant population.

We computed the correlations in the UK10K database of genotypes of 3,781 British individuals. For simplicity we chose to impute 2.1 million HapMap SNPs. We then convert all z-statistics, observed and imputed, into the standardized effect estimates that are used in the remainder of the method; $\widehat{\Gamma}_{it} \approx w_{it}/\sqrt{n_t}$ where $n_t$ is the sample size in the GWAS for disease $t$.

**MR estimation for causal effects of diseases on lifespan.** This method requires, for each SNP, an estimate of its effect on lifespan, which is independent of its summary statistic in the parental lifespan study. We refer to this estimate as the *prior* of the SNP. To compute this prior, using the estimated effects of the SNP on non-lifespan diseases and traits, we first need an estimate of the effect of each of those diseases on lifespan. In this section, we describe how to compute these causal effect estimates. As an individual SNP may affect lifespan via two different diseases, it is better to estimate all the causal effect estimates jointly to allow for this correlation. We refer to this joint estimation as multivariate Mendelian randomization (multivariate MR), and we applied it to the 58 studies with strong instruments. We established a set of SNPs that are strong instruments

(F-statistic > 10, which in our case was equivalent to $P < 10^{-5}$) for at least one of the 58 studies.

The conventional method to estimate the causal effects of diseases on lifespan is to use the data on each individual's (parental) lifespan and on each individual's disease status. We do not have access to such detailed data, but we describe here how we derive equivalent estimates of the causal effect by using the Z-statistic data that is available to us, which gives us the estimates of the effect of all the SNPs on lifespan and on the various diseases.

Given $n$ individuals, $m$ SNPs and $d$ diseases define the $n \times m$ matrix $Z$ which records the genotype of each individual, an $n \times d$ matrix $X$ which records the disease status of each individual for the $d$ diseases, and a vector $\mathbf{y}$ of length $n$ with the parental lifespans. Every column of $X$, $Z$ and $\mathbf{y}$ is assumed to have been standardized to have mean zero and variance one.

Using two-stage least squares, we first define

$$\widehat{X} = Z(Z'Z)^{-1}Z'X \tag{2}$$

With such data, as described in equations (8–9) in Section 8.3.4 of Green et al.[49], we would estimate the causal effects via

$$\mathbf{b} = \left(\widehat{X}'\widehat{X}\right)^{-1}\widehat{X}'\mathbf{y} \tag{3}$$

The summary statistic data, which we do have, takes the form of a vector $\hat{\gamma}$ of length $m$, reporting the estimate of the effect of each SNP on lifespan. Also, for the disease traits, we have an $m \times d$ matrix $\widehat{\Gamma}$ of the estimated effects of each SNP on each disease,

$$\hat{\gamma} = \frac{1}{n}Z'\mathbf{y} \tag{4}$$

$$\widehat{\Gamma} = \frac{1}{n}Z'X \tag{5}$$

These estimates of the standardized effects come from the GWAS summary statistics and were calculated assuming a linear model for the relationship between each SNP and each trait (including lifespan). For example, each extra minor allele will increase or decrease the log-odds of a binary trait, or linearly increase or decrease continuous traits. We infer the standardized effect estimate of SNP $i$ on trait $t$, that is, $\widehat{\Gamma}_{it}$, from the observed (or imputed) Z-statistic and the sample size of the GWAS for that trait, $\widehat{\Gamma}_{it} \approx w_{it}/\sqrt{n_t}$.

To compute $\mathbf{b}$, we begin by rearranging $\widehat{X}$,

$$\begin{aligned}\widehat{X} &= Z(Z'Z)^{-1}Z'X \\ &= Z\left(\frac{1}{n}Z'Z\right)^{-1}\frac{1}{n}Z'X \\ &= Z \quad C^{-1} \quad \widehat{\Gamma}\end{aligned} \tag{6}$$

where $C = \frac{1}{n}Z'Z$ is the genetic correlation matrix due to LD. We then substitute this expression for $\widehat{X}$ into equation (3) yielding

$$\begin{aligned}\mathbf{b} &= \left(\widehat{\Gamma}'C^{-1}Z'ZC^{-1}\widehat{\Gamma}\right)^{-1}\widehat{\Gamma}'C^{-1}Z'\mathbf{y} \\ &= \left(\widehat{\Gamma}'C^{-1}\frac{1}{n}Z'ZC^{-1}\widehat{\Gamma}\right)^{-1}\widehat{\Gamma}'C^{-1}\frac{1}{n}Z'\mathbf{y} \\ &= \left(\widehat{\Gamma}'C^{-1}CC^{-1}\widehat{\Gamma}\right)^{-1}\widehat{\Gamma}'C^{-1}\widehat{\gamma} \\ &= \left(\widehat{\Gamma}'C^{-1}\widehat{\Gamma}\right)^{-1}\widehat{\Gamma}'C^{-1}\widehat{\gamma}\end{aligned} \tag{7}$$

This last equation gives us estimates for the causal effects in terms of quantities that we have access to.

The covariance of this estimator, assuming homoscedastic errors, is

$$\mathrm{Var}\left[\mathbf{b}\right] = \sigma^2\left(\widehat{X}'\widehat{X}\right)^{-1} = \hat{\sigma}^2\left(\widehat{\Gamma}'C^{-1}\widehat{\Gamma}\right)^{-1} \tag{8}$$

where

$$\hat{\sigma}^2 = \frac{1}{n}\left(\mathbf{y} - \widehat{X}\mathbf{b}\right)'\left(\mathbf{y} - \widehat{X}\mathbf{b}\right) \tag{9}$$

represents the variance of the outcome trait unexplained by the causal predictor, which in reality is very close to 1. The covariance estimator for heteroscedastic errors requires individual level data and hence not suitable for summary statistics-based MR estimation. Note that Burgess et al.[50] introduced a maximum likelihood-based MR method using only summary statistics. Since the lifespan study (UK Biobank) does not share samples with the disease trait studies, our formula (equation 7) yields the same result conditional on the SNP-trait estimates having little variance. To make sure that our method gave sensible results, we implemented the maximum likelihood-based MR method by Burgess et al.[50] and confirmed that we obtained indistinguishable solution to their likelihood function (equation 1 in their study) maximization (see Supplementary Table 9 for the causal estimate comparison and the between-trait correlation estimates ($\rho$ in their paper) are in Supplementary Table 10).

Instead of using every SNP, we use only those which are strong instruments, and we use only a subset of those strong instruments that are spread further apart

on the chromosomes in order that their genetic correlation is low. This separation allows us to approximate $C$ by the identity matrix $I$, which improves numerical stability and simplifies the equations. It gives us the expressions we estimate the causal effects and the covariance of those estimates, where $\hat{\gamma}$ and $\hat{\Gamma}'$ are computed only over a subset of the SNPs. Refer to this small set of uncorrelated strong instruments as $S$ to select the relevant subset of rows from $\hat{\Gamma}$ and $\hat{\gamma}$.

To select $S$, the subset of strong instruments where there is a low level of genetic correlation (LD) between instruments, we began with the SNP with the most significant $P$ value (across all traits) and then discarded any SNP in LD ($r^2 > 0.2$). Repeating this pruning procedure results in 6,443 independent SNPs used for the estimation of causal effects. To avoid weak instruments the standardized effects for a SNP-trait pair with $p_{t,i} > 10^{-5}$ was set to zero. Define $\hat{\gamma}_S$ as the effect estimate of the S SNPs on lifespan. Define $\hat{\Gamma}_S^*$ as the effect estimates of the S SNPs on the 11 selected traits, where $\hat{\Gamma}_S^*$ has been set to zero if $p_{it} < 10^{-5}$, and we can then simplify equations (7 and 8) to

$$\hat{\mathbf{b}} = \left(\hat{\Gamma}_S^{*'}\hat{\Gamma}_S^*\right)^{-1}\hat{\Gamma}_S^{*'}\hat{\gamma}_S \tag{10}$$

$$\text{Var}\left[\hat{\mathbf{b}}\right] = \hat{\sigma}^2\left(\hat{\Gamma}_S^{*'}\hat{\Gamma}_S^*\right)^{-1} \tag{11}$$

This expression for estimating $\hat{\mathbf{b}}$ is equivalent to using a linear regression between $\hat{\gamma}_S$ and $\hat{\Gamma}_S^*$ and using the estimated coefficients as the estimates of the per-trait causal effects. Instead of using all 58 non-lifespan traits, we used stepwise-selection, starting with an empty set of traits and adding or removing to maximize the AIC. Among AIC-selected variables we further removed those with multivariate $P$ value > 0.05, which resulted in 11 traits. In other words, 11 traits were identified to have a significant causal effect on lifespan. For the remainder of the method, computing the prior and the BF, only these 11 traits were used. In all these regression analyses we forced the intercept to be zero, but fitting the intercept too did not change the estimated causal effects (Supplementary Table 2).

To compute the BF for a given SNP, as described later, we require an estimate of the causal effects for these 11 traits that is independent of the estimated lifespan effect for that SNP, $\hat{\gamma}_i$. We therefore repeat the procedure for estimating the causal effects 22 times, where each chromosome in turn has been 'masked'. This gives us 22 vectors of causal effect estimates. When computing the prior for a SNP on a given chromosome, we use causal effect estimates computed using only the instruments from the other 21 chromosomes. This 'masking' process is to ensure that the prior for the longevity Z-statistic of a particular SNP is completely independent of its actual Z-statistic in the longevity study.

**Building association prior for lifespan.** The next step is to compute, for every SNP genome-wide, a prior for its effect on longevity as a function of its estimated effect on the 11 non-lifespan traits and the causal estimates of those 11 traits on lifespan. For each SNP $i$, we therefore took the truncated standardized effect sizes ($\hat{\Gamma}_{it}^*$) across the 11 lifespan-influencing traits and multiplied them by the estimated causal effect for the corresponding traits ($\hat{b}_t$, equation (10)). Note that only 77,963 SNPs had association $P$ value $< 10^{-5}$ for at least one of the 11 traits, thus for all other SNPs all $\hat{\Gamma}_{it}^*$ were set to zero, hence their prior effects to zero. The sum of these products, $\mu_i = \sum_{t=1}^d \hat{b}_t\hat{\Gamma}_{it}^*$, is our prior belief of the standardized effect of the SNP on lifespan. Given the independence of $\hat{b}_t$ and $\hat{\Gamma}_{it}^*$, we estimated the variance of the prior $\mu_i$ as

$$\text{Var}(\mu_i) = \text{tr}\left(\text{Var}(\hat{\mathbf{b}})\right)\text{tr}\left(\text{Var}(\hat{\Gamma}_{i,\circ})\right) + \hat{\mathbf{b}}'\text{Var}(\hat{\Gamma}_{i,\circ})\hat{\mathbf{b}} + \hat{\Gamma}'_{i,\circ}\text{Var}(\hat{\mathbf{b}})\hat{\Gamma}_{i,\circ} \tag{12}$$

where $\text{Var}[\hat{\mathbf{b}}]$ is estimated as $\hat{\sigma}^2\left(\hat{\Gamma}_S^{*'}\hat{\Gamma}_S^*\right)^{-1}$, as in equation 11, and $\text{Var}(\hat{\Gamma}_{i,t}) = 1/n_t$. It is interesting to note that our method is robust to misreported sample sizes as the relevant term in the prior, the product $\hat{\Gamma}_{ti} \times \hat{b}_t$, will be unchanged; a misreported sample size for a given study will affect both factors in that product, but the errors will cancel out and the prior will be unaffected.

Also, as mentioned earlier, most of the 2.3 million SNPs will have zero prior, $\mu_i = 0$, but the variance, $\text{Var}(\mu_i)$, will be non-zero and hence they do have a prior, albeit very weak.

**Bayes factors and their distribution under the null.** A conventional GWAS will consider how likely the observed lifespan or longevity z-statistic coming from a standard normal distribution, $N(0,1)$. When using standardized effect estimates, their distribution under the null is $(0, n_l^{-1})$, where $n_l$ is the sample size in the longevity study. The prior In Bayesian analysis, we also consider the likelihood under an alternative hypothesis to compare it to the likelihood under the null. While under the null distribution the prior has zero mean and zero variance, the prior distribution under the alternative hypothesis is $N(\mu_i, \text{Var}(\mu_i))$.

For each of the 2.3 million SNPs, we can now compare the effect estimate observed in the longevity GWAS to the prior computed above. The BF is defined as the ratio of the likelihoods of the observed longevity effect size estimate, $\hat{\gamma}_i$, under the two hypotheses. Considering the alternative hypothesis as a prior distribution on $\delta$, which takes the place of the true mean, we use integration to compute the

marginal likelihood,

$$\text{BF}_i = \frac{\int L(\hat{\gamma}_i; \delta, \text{Var}(\hat{\gamma}_i))L(\delta; \mu_i, \text{Var}(\mu_i))\text{d}\delta}{L(\hat{\gamma}_i; 0, \text{Var}(\hat{\gamma}_i))} \tag{13}$$

where $L(\gamma, \mu, \sigma^2)$ is the density of $\gamma$ under the corresponding Gaussian distribution

$$L(\gamma, \mu, \sigma^2) = \frac{1}{\sqrt{2\pi\sigma^2}}e^{-\frac{(\gamma-\mu)^2}{2\sigma^2}} \tag{14}$$

As shown elsewhere [http://www.cs.ubc.ca/~murphyk/Papers/bayesGauss.pdf], the ratio of above is equivalent to

$$\text{BF}_i = \frac{L(\hat{\gamma}_i; \mu_i, \text{Var}(\hat{\gamma}_i) + \text{Var}(\mu_i))}{L(\hat{\gamma}_i; 0, \text{Var}(\hat{\gamma}_i))} \tag{15}$$

To define a set of SNPs that survive 5% FDR we first needed to assign $P$ values to the obtained BFs. As standard procedure with any test statistic (BF is a test statistic), we used a permutation-based approach to calculate $P$ values corresponding to each BF (ref. 51). In practice, we generated BFs genome-wide for the same 2.3 million SNPs when the lifespan Z-statistics (and the corresponding $\hat{\gamma}_i$) were replaced by Z-statistics derived from GWAS scans with 1,000 random outcomes. We then calculated the resulting BFs for each of the 1,000 null sets of Z-statistics while keeping the same priors as before ($N(\mu_i, \text{Var}(\mu_i))$ for SNP $i$). In formula, the BF for SNP $i$ for the $k$-th random trait is defined as

$$\text{BF}_i^{(k)} = \frac{L(\hat{\gamma}_i^{(k)}; \mu_i, \text{Var}(\hat{\gamma}_i^{(k)}) + \text{Var}(\mu_i))}{L\left(\hat{\gamma}_i^{(k)}; 0, \text{Var}(\hat{\gamma}_i^{(k)})\right)} \tag{16}$$

where $\hat{\gamma}_i^{(k)} = \frac{1}{n}Z_i'y^{(k)}$ with $Z_i$ representing the standardized genotype vector for SNP $i$ and $y^{(k)}$ stands for the $k$-th random phenotype vector drawn from a standard normal distribution. This gave us a large set of 2.3 billion (2.3 M × 1,000) null BFs emerging from the exact same SNP set with the same priors to compute empirical $P$ values for each observed BF. These $P$ values were then subjected to Benjamini–Hochberg step-up procedure to select the largest set that survives 5% FDR. Using the null BFs, we also estimated the per comparison error rate[52] (under weak control) corresponding to our selection of 16 SNPs with logBF > 2.2857, which yielded 4.88% concordantly with the 5% FDR control. The rationale behind our FDR control procedure has been outlined and applied previously[53], however the estimation of the null BF distribution we proposed here is less arbitrary.

**Calculating effect of the 16 SNPs on different scales.** We followed the same protocol to calculate Cox proportionate HRs[54] for parent survival for the top 16 SNPs as was done by Joshi et al.[14] (see summary above). Analyses here used imputed allele dosages therefore subject counts (115,180/111,193 for mother/father) are slightly higher than reported by Joshi et al.[14], where subjects with missing genotypes were excluded. Effects (log(HR)) on maternal and paternal lifespan were meta-analysed using inverse-variance weighting[55]. We also used dead parents only to estimate the effect of these SNPs on lifespan in years-per-allele using linear regression. Another way to obtain effect estimates in terms of years of life lost is to multiply the log (HR) by 10 (ref. 56). Since in our study the parental genotypes were approximated with that of the offspring the observed effects are expected to be half of the effect one could have obtained had parental genotypes been used. To correct for this we multiplied the obtained per year effect by two in both analysis. We compared these two estimates and found remarkable concordance (Supplementary Table 11).

**Combining evidence from replication studies.** We used the following UK Biobank-independent studies for replication: the CHARGE study[10], the 90PLUS longevity study[11], a genome-wide survival study[12] and a recent disease-informed extreme longevity GWAS (iGWAS)[18] including two cohorts (NECS and 90PLUS). As effect sizes were available for the first three studies, we combined the summary statistics via the method of Lin et al.[47] accounting for sample overlap, and then converted the resulting summary statistics into one-sided $P$ values. Since effect sizes and directions are unknown for the iGWAS, we used the Fisher's combined probability test[22] to meta-analyse the aggregated $P$ values from the first three studies with the $P$ values from the two iGWAS cohorts. To account for the sample overlap we used 236,383 off-target SNPs (available or imputed in all studies) to estimate the null cumulative density function for the sum of the $-2\ln(P)$ values, instead of using a $\chi^2$ distribution as done in the standard Fisher's method. Note that for the iGWAS NECS study we used the best available proxies for 14 of the 16 SNPs ($r^2$ ranging 0.25–1, see Supplementary Table 12).

**Relationship with age-related genes and methylation probes.** To identify potential causal genes at the 16 discovered loci, whose expression levels may modulate lifespan, we applied Mendelian randomization. As described above (equations (10 and 11)), MR can be performed using summary statistics only. Such analysis elucidates whether the impact of SNPs on the expression level is proportional to their effects on lifespan. We thus combined eQTL data (in all GTEX tissues[23]) with the lifespan association results. To ensure instrument validity we interrogated only those genes that had more than five independent eQTL SNPs.

Next, we asked whether SNPs associated[26] with at least one of the 1497 age-correlated[24] gene expressions are enriched in lower than expected lifespan association $P$ values. Analogously, we also searched for SNPs associated[27,28] with at least one of the 353 age-associated CpG sites[29] and compared their lifespan-association $P$ values to a random set of the same size. This analysis is meant to test whether most age-associated biomarkers (expression or methylation) are causally linked to lifespan.

**Depletion of lifespan-shortening alleles in older people.** If these 16 SNPs are truly associated with lifespan, their life-shortening alleles should be depleted in older individuals in general population cohorts, since people carrying these alleles would have less chance to survive in the cohort at older ages due to selection bias. We tested this hypothesis by modelling age as the outcome in a linear regression with regressors such as SNP allele dosage, gender, 15 ancestry principal components and batch indicator in 120,000 genetically white British participants of the UK Biobank. We then asked what fraction of the one-sided $P$ values fell below 0.05 and compared it to the expected number through the binomial test.

**LXS mouse data.** Male mice from 41 ILSXISS (LXS) recombinant inbred strains were maintained *ad libitum* feeding[30]. Median lifespan was calculated to represent longevity across strains. Microarray data from prefrontal cortex of LXS mice with the average age of 72 days were generated by Dr Michael Miles, and downloaded from GeneNetwork.org (GN Accession: GN130).

**Caloric restriction gene expression data analysis.** For each species, Affymetrix raw .CEL files were downloaded from GEO database[57] and preprocessed using robust multi-array averaging (RMA) normalization[58]. The quality of datasets was checked with principal component analysis (PCA) of the samples, gender and other batches. Differential gene expression analysis between controls and CR samples was performed using limma R package (R software 3.2)[59]. For multiple testing correction, a FDR <0.1 was chosen. The normalized microarray expression values of interested genes are shown on log2 scale.

**Data availability.** The disease GWAS summary statistics are publicly available and the sources are provided in Supplementary Table 1. The resulting Bayesian association summary statistics can be downloaded from http://wp.unil.ch/sgg/bayesian-lifespan-gwas/.

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

## Acknowledgements

This research has been conducted using the UK Biobank resource. We thank the CHARGE Consortium, in particular Andrew Johnson and Joanne Murabito, for the CHARGE lifespan association summary statistics. We are grateful to Peter Vollenweider and Sven Bergmann for the critical reading of the manuscript. Z.K. received financial support from the Swiss National Science Foundation (31003A-143914). Z.K., B.D., J.A. and M.R.-R. received support from SystemsX.ch (51RTP0_151019). J.A. and R.W.W. received support from EPFL, NIH (R01AG043930), SystemsX.ch (SySX.ch 2013/153) and SNSF (310030B-160318). M.R.R. received support from Etat de Vaud and SNSF (31003A-153341).

## Author contributions

Z.K designed the experiments, A.F.M., P.K.J., E.P., A.K., H.L., V.S., M.L., R.P.J.B., S.R. and Z.K. analysed the data, H.L. performed experiments, A.R., M.B., B.D., R.W.W., M.R.-R., F.P., V.R., J.A., J.F.W. and Z.K. oversaw the work, A.F.M., P.K.J., J.A., J.F.W. and Z.K. wrote the paper.

## Additional information

**Competing interests:** The authors declare no competing financial interests.

