## [Peer Review File · Nature Communications]

Reviewers' comments:

Reviewer #1 (Remarks to the Author):

Review of McDaid et al.

The authors performed a gene association study to identify 18 SNPs associated with lifespan using data from UK Biobank. Most of these SNPs are novel. The approach was to use information from prior studies of diseases in order to calculate weighting coefficients using the Mendelian randomization approach. Validation studies using independent cohorts replicate some of the lead SNPs. Bioinformatic analyses of the 18 lead SNPs were performed to identify linked genes, how the SNPs might affect expression of these genes and how changes in the activity of these genes might affect lifespan. The identification of longevity-associated SNPs using Mendelian randomization is new, and the results for this main analysis are solid. However, the replication and expression analyses are problematic.

My main concerns are:

1. Conceptual novelty. If a SNP is associated with a disease and the disease is associated with increased morbidity, doesn't it follow that the SNP will show an association with normal lifespan given enough data? It is not surprising that SNPs that increase one's chance of coronary artery disease, Type 2 Diabetes and schizophrenia, etc. are associated with normal longevity. It is admirable that this study found 18 SNPs with significant association with normal longevity, but eventually all of the disease-related SNPs will show an association if the amount of longevity data is at a sufficient level.

The disease GWAS returned two surprising results: 1) education attainment was the strongest factor for increasing longevity and 2) major depressive disorder increased, rather than decreased, longevity. I recommend that both these results be discussed in more depth. Education attainment likely indicates access to better health care, rather than having a biological mechanism such as reduced risk for diseases of morbidity. Is there a way to rationalize the positive effect of major depressive disorder?

2. Replication in independent studies. A weak point in the paper is that the replication results are over sold. Two out of 18 lead SNPs were validated in the CHARGE and EU longevity consortia using the Bonferroni threshold. Evidence that 6/18 lead SNPs had $P < .05$, that 6/18 is more than expected by chance and that 16/18 had effects in the same direction are only weak evidence that there is true signal amongst the 18 lead SNPs. The interpretation of the replication study should be more rigorous and conservative.

3. Mouse expression studies (p11, line 380-389). The mouse expression analysis is flawed.

a. The analysis compares expression in different BXD or LXS mouse strains. This is not the same as showing the effects on lifespan of raising or lowering expression in one strain.

b. Does expression of the three genes in Sup. Figure 5 act as QTLs for lifespan in the mouse strains?

c. How many genes in the genome show differences in expression between the different mouse strains? How surprising is it that some of the eight genes analyzed would show differences in expression in the mouse strains given the overall number of genes that differ in expression between the strains?

d. What were the criteria to select 8 genes for analysis out of all of the genes near the 18 lead SNPs?

f. p11, line 385 "Expression of AMIGO1 and RGS12 showed P-values < 0.05 (132 - 172 days of lifespan increase / expression unit, Pone-sided = 0.006, 0.037, respectively)." I don't understand where these values come from. RGS12 does not show a significant p value for either the red or brown dots. AMIGO1 shows a significant p value only for the red dots.

g. Supplemental Figure 5. What are red and brown dots? What is the y axis? What are the units of the X axis?

h. P11, line 393 CDKN2B and LAT mouse knockout results are not valid comparisons. The knockouts might die because of the trivial reason that the genes are essential, not because the knockout is mimicking the longevity effect in humans.

P8 line 242. "none of their 14 top hits reached nominal significance level in our study". The P values in Supplemental Table 2 seems to indicate that all 14 are significant.

P8 Line 266. "Despite the winner's curse bias, as a general trend these SNPs seemed to have larger effects on extreme longevity, especially at the APOC1 locus." The data for extreme longevity vs lifespan should be shown.

Table 1. Define effect size.

Table 2. Column headings should be defined explicitly.

Figure 6. Give effect size not just p values. What is fold-change in expression for the A allele, and how much does expression of this gene change per year?

Supplemental Table 1. Column headings should be defined.

Supplemental Table 2. What are P and UKBB Both?

Supplemental Table 3. Beta, P and N SNPs should be defined.

Supplemental Table 4. Outcome, Beta should be defined.

Supplemental Table 5. This is the wrong table. Text indicates that it has data about 35 mouse strains, but Table has data on three human SNPs. Column headings should be defined.

Supplemental Figure 6. What are the units for the y axis? What is H? Where are values for mouse and fly? What are the names of the genes in the other species? Where are data for LAT expression?

Reviewer #2 (Remarks to the Author):

The paper by McDaid et al. report the association of several SNPs with human 'lifespan' based on a pre-informed GWAS on parental age at death. They subsequently show that several of the genes associated with these SNPs are also associated with differential gene expression with age in human peripheral blood and with lifespan in mice.

I really like the approach taken by the authors and I think it could really help to move the field of genetics of human lifespan forward. However, some important details regarding the analyses are missing and the use of terminology is not always correct.

Major comments:

1. The authors use the term 'lifespan' throughout the document. However, sometimes they actually refer to parental age at death (or parental lifespan), which is not the same as the lifespan of an individual. This should be changed accordingly.

See for example:

Line 104: "associated with lifespan"

Line 115: "SNP-phenotype effect sizes and the trait-lifespan"

Line 118: "lifespan in a general population cohort"

Line 131: "lifespan (see Online Methods)"

Line 143: "Causal effects of GWAS traits on lifespan"
Figure 1

2. It is well-known that some of the phenotypes identified to be associated with 'lifespan' show an opposite association with mortality later in life (>85 years) as compared to middle age, see for an example van Vliet et al. *Gerontol A Biol Sci Med Sci*. 2010. I think this should be taken into account in the analyses performed by the authors.

3. There is some important information lacking concerning the replication studies. The authors mention that "SNP were tested for replication using summary statistics from the combined CHARGE and EU longevity studies". They have to describe this in more detail in the Methods section. For example, which summary measures were used (Beta and SE's or P-values)? Moreover, the results of the individual studies should, at least, be shown in a Supplementary Table to determine if the associations are not driven by only one of the two studies.

4. The results of the look-up in the GWAS of Walter et al. and Fortney et al. should be shown in a Supplementary Table (lines 238-242 and 247-250).

5. It is unclear how the authors determined the genetic correlation between the GWAS on parental age at death and the combined CHARGE and EU longevity GWAS / survival GWAS. They should explain this in more detail in the Methods section. The fact that the most important loci from both studies ($P < 10^{-5}$) are not replicated between them, with the exception of ApoE, makes me question if the provided number is actually correct. It seems that parental age at death is a suboptimal proxy for longevity and, thus, healthy aging.

6. How do the authors explain the fact that three out of the four SNPs that show an age-related eQTL effect could not be replicated in the CHARGE and EU longevity GWAS, survival GWAS, and the extreme longevity study? It seems that these SNPs might actually not be that relevant for lifespan. This is further confirmed by the fact that the expression of these genes is not associated with parental age at death in the CoLaus study. Why have the authors decided to take these genes forward for the analyses in model organisms?

7. Why would the authors perform the MR-analysis described from line 352 to 359 if they already showed that the lead-SNPs at the identified loci are no eQTL for the genes they are looking at? The authors should explain the reasoning behind this analysis in more detail or remove it from the manuscript.

8. I think the part on the model organisms is a really nice addition, but it is currently too condensed and the majority of results are not shown in a Supplementary Table or Figure. On the other hand, the Tables and Figures that have been created (Supplementary Table 5, Figure 5 and 6) are unclear and lack detail in the description (especially Supplementary Figure 6; how did the authors combine the fly and mice data here and why is LAT missing from this Figure). In addition, the selection of genes taken forward to this stage is questionable (see point 6 and 7).

9. The authors should explain why they would look at the expression of the genes in liver in mice. How can they be certain that this is comparable to expression of the gene in human peripheral blood? Do they have evidence (for example from GTEx) that liver is a good proxy for peripheral blood?

10. The format of the Supplementary Tables and Figures is really sloppy and some are almost impossible to read. For example, it is unclear what the column-headings in Supplementary Table 1 actually represent. In addition, three of the panels in Supplementary Figure 1 contain some strange

lines on the left side of the forest plots.

Minor comments:

- Line 30: The authors refer to the CHRNA3/5 locus as a locus robustly associated with human lifespan. However, I do not agree with this. The locus has only been found in two papers based on the same population and have not (yet) been validated in a large independent lifespan-related study. This is exemplified by the fact that the authors were unable to replicate this locus in the combined CHARGE and EU longevity studies. The authors should therefore not mention this locus as robustly-associated here. On the other hand, the locus near EBF1 has been replicated in an independent longevity GWAS (Zeng et al. Scientific Reports 2016) and could therefore be added here.

- Line 61: the "a" is missing between "have" and "strong".

- Line 85: The authors mention that one could identify "longevity-associated" loci using survival analysis. However, this is not the case, since one is looking at a decrease or increase in frequency per year, which is not the same as longevity. They should change this into "lifespan-associated".

- Line 104: Was the study on parental age at death able to find association of the locus near EBF1? If not, the authors should also mention this here for consistency.

- Line: 187: The authors should also provide a lambda belonging to Supplementary Figure 3.

- Line 204 to 206: This is a strange comment. The authors selected SNPs with a P-value $< 10^{-5}$ to be considered associated with a trait and not $< 5 \times 10^{-8}$, so why would this observation be worth mentioning?

- Line 307: "EBRF1" should be "EBF1".

- Line 327: Please add "peripheral blood according to" between "in" and "GTEx" for sake of clarity.

- Line 345-350: I do not see the added value of this analyses and I think it should be removed from the manuscript. If the authors want to keep them in they should show the results of these analyses in a Supplementary Table or Figure.

- Why was C-reactive protein not included in the list of age-related traits? It is well-known that this trait is associated with lifespan.

- The authors should refer to rs4420638 as APOE and not APOC1 throughout the manuscript. It is well-known that this SNP tags the effect of ApoE e4.

- Supplementary Table 2: "Walters" should be "Walter".

- I think Supplementary Table 5 is missing (described from line 380-385). Please add this Table.

Reviewer #3 (Remarks to the Author):

The manuscript by McDaid et al. describes a GWAS-like Bayesian framework to identify loci associated with age of parental death through increasing disease risk. They apply their method to data obtained

from the UK Biobank study, which contains genotype and lifespan information for over 100,000 individuals. Through combining this data with previously performed GWAS, they first derived priors for association of all HapMAP SNPs with longevity using a Mendelian Randomization-esque framework, then combined this information with the SNP/parental longevity associations to calculate posteriors of each SNP being associated with longevity. They then work to show that several of their genes may influence lifespan in model organisms and propose mechanisms of action.

Previous studies have found only a few genetic variants that robustly associated with longevity (an area the authors must know is somewhat contentious). To improve discovery power for longevity variants, this paper uses a Bayesian approach to prioritize variants that are associated with disease known to influence lifespan. This is simultaneously a strength and the weakness of the paper. Specifically, it is not surprising that traits associated with premature death -- for example, Coronary Artery Disease (CAD) and causally related traits to CAD like low-density lipoprotein cholesterol -- would be associated with longevity (and that risk alleles here shorten life span). Even though this causal path is known, the paper missed at least one mark where this case is fairly strong: e.g., in contrast to AMIGO1, SORT1 is by far the stronger candidate and story (see below).

Another challenge is that the paper brings together several analysis, but with unclear narrative. Some results did not seem relevant to the central story (e.g., the LD score regression analysis). Other analysis components were sensible to mention, however the results were either negative, the evidence supporting them was quite weak, or they were confusing (the gene expression analysis in model systems). Thus, it was a challenge to deduce what the new contributions were from the authors that was backed with convincing evidence about the genetics of longevity.

Specific comments are provided below.

Major Comments

1. The overall impact of this study is unclear.

a. As stated by the authors in their introduction, it has already been established that certain diseases play a major role in shortening longevity. In addition, through GWAS and other means, loci influencing risk to those disease have been found. If a disease shortens longevity, then it seems like most loci contributing to it will influence longevity. What is the significance of picking out only a subset of these loci through complex methods and pointing out that they contribute to longevity? The authors do not provide a specific rationale or impact. If the authors believe the basic inference from the MR experiment, does it not follow that most (if not all) loci that modulate a biomarker then cause shortening of lifespan?

b. Given that these loci are established (i.e., genome-wide significant) for their respective trait, the observation that most of the discovered loci 'replicate' in independent datasets for longevity is good, however doesn't feel that surprising. What is the new biological insight revealed here? Additional discussion and clarification of the goal and impact of this study would greatly strengthen the report.

2. Concerns about generation of the priors, and the priors themselves.

2a. Validity of the selected features. Essentially, the authors are performing a type of feature selection (machine learning) to determine which risk factors go into their downstream analysis. But how do the authors know that the selected features are best, the weight for those features are accurate, and that the models selected are valid and generalizable? Typically, in the "machine learning" space, cross-validation, true holdouts, etc. would be used to demonstrate the robustness, accuracy, goodness of fit (neither over, nor under-fitting). While it is appreciated that this isn't a central focus of the paper, it

is crucial to the applicability of the approach in general terms. But also specifically here: if the priors are over fit, the subsequent association analysis will require less evidence for sites with nominal associations (because the prior is stronger), increasing false positives. The authors should perform cross-validation, or a similar approach, to show that the priors would remain stable if different, but similar, disease traits were used.

Note that the chromosome holdout experiment does not address this issue (this addresses a different question, not on the features selected).

2b. Several correlated, and possibly causally connected traits, such as cholesterol, BMI and Type 2 diabetes, are risk factors selected by MR analysis. A previous paper on multivariate MR (Burgess and Thompson, 2015, PMID 25632051) indicates that causal relationships between risk factors can cause widely varying effect estimates when regression methods are used. Isn't the model double counting here, in the selection? In addition, what happens with SNPs that have opposing effects on multiple traits? The complexities here are numerous, and are not very well described in the methods, but impacts interpretation of the features selected and if the model is good or overfit (see comment #2a, above).

2c. A quantitative measurement of the similarity in the causal-effect (α) values estimated in each of the 22 leave-one out chromosome sets should be provided to reinforce that this method is robust.

2d. While authors state that they are not using the MR results in a causal framework, they indirectly are, in interpretation. Some of those could indeed make sense, too. The paper calls these "causal" effects - and I'm not certain I agree with that. Robustness, sensitivity, confounding - these things have not been worked through on these causal effect estimates. There a bit of an art form and effort to making strong instruments to build a causal case (risk score analysis, sensitivity, heterogeneity, robustness analyses), and unsure if that has been convincingly shown here.

2e. What is potentially confusing or hard to wrap around is the result on educational attainment. The 'causal graph' is unclear. I know the authors will say that the /specific/ features don't matter; but an implication that *could* be made is that educational attainment increases lifespan - which clearly doesn't make a lot of sense, unless educational attainment itself was correlated with other factors that did have an impact on lifespan (alcohol use, poverty, etc.). I am vaguely aware of papers in this area (Marioni et al 2016, PNAS), so given that this is the most substantial contributor, it may be worth having some discussions on the strengths and limitations here.

3. Concerns about the section on the model-systems 'validation' experiments. Overall, this component seemed unconvincing, undermining the message that these loci are significant effectors of longevity.

3a. The clearest example of that is rs646776. This is the strongest eQTL in the genome (for liver), and the underlying, previously implicated gene is SORT1, not AMIGO1 as the authors state, for LDL-C. See Musunuru 2010, PMID: 20686566.

The clearest hypothesis is that the effect on longevity is mediated through LDL-C, which increases risk for CAD (which this variant is also strongly associated with), and premature death from myocardial infarction. Yet, the authors fold this into a case where the expression is correlated (in blood) with another gene to a much lesser extent.

Given the above, it is unclear what the purpose of this analysis is, and if it can be interpreted in the way the authors intend: that this variant changes gene expression of these genes, which then impacts lifespan (at least in one case, if this hypothesis were true, I wouldn't pick out AMIGO as the causally

related gene!).

3b. The tissue types under consideration are also fairly scattered. In the first paragraph of the expression in model organism section, the authors state that their analysis will be done in blood and brain. However, in the next paragraph, they do their analysis in liver, with no explanation. Can we learn what the authors want to infer from this experiment, given that the tissues are different? If so, why?

3d. Furthermore, no reports for the *SULT1A1* and *CHRNA5* were reported. Does that mean that 2/3 of the lookups were inconsistent with your hypothesis?

3e. The last section, there are several experiments from lookups the authors state themselves admit are "inconclusive". As the authors state, inconclusive aging trends were observed across different tissues in model organisms. In addition, most genes did not show signal in the correlation analysis, seen in lines 380-389.

Except for caloric restriction data sets, which is theoretically interesting, but pretty hard to interpret (and there's no statistics provided to do this).

3f. The data for *AMIGO1* and *RGS12* are pretty weak effects. Why should this result be convincing, given the issues above? How often would we see this level of difference picking a random gene?

3g. The only gene that appeared to have strong effects in Supplementary Figure 5 is *RBM6*. However, given the issue of gene selection (*AMIGO1* isn't the causal gene) and the tissue discordance, it is pretty challenging to evaluate how much positive, additional evidence on longevity this study affords. The relationship between the CR diet and expression in brain or blood seems to be only source of signal; given that this is a non-trivial perturbation, this also makes interpretation incredibly challenging.

4. The overall flow of the manuscript.

4a. The message of the section "Genetics of disease, lifespan, extreme longevity and selection bias" is unclear. Overall, it read like a detour from the overall message of the paper of identifying and verifying loci associated with lifespan. How do these analyses connect to the central message or narrative the paper, which felt more like (at least my take on the paper): "disease-associations for phenotypes related to premature death associates with longevity"? If the authors believe this section is important, it should be put into a broader context and/or moved closer to the end.

4b. The connection between disease and lifespan has already been demonstrated, so we are unclear of the importance of the LD-score regression showing this, especially given its limitation as noted by the authors. A comparison with previous estimates, or clarification of the goal of the analysis, could be helpful.

4c. Importantly, methods describing the data set used in the analysis on lines 263-275 should be explicitly stated. Moreover, what set of SNPs were used for this analysis?

5. It seems like the approach (generally) is: identifying a subset of SNPs for testing, then boosting individuals ones given the MR data. It wasn't immediately clear, but are the authors controlling 5% error for 72,735 test? That seems far too lenient.

6. Did you include population stratification components when associating age with your SNPs?

7. The MR gene expression experiment was not particularly detailed. What is the hypothesis? Were the SNPs eQTLs for multiple genes? Why was brain the focus?

Minor Comments

8. Make clear in Figure 1 that alphas are generated using all chromosomes except the one that the SNP is found on. Also, define variables in MR section of figure, or remove them.

9. Give a more thorough explanation of the risk factor selection process, especially what was meant when the authors say they removed traits with multivariate P-value >0.05 . Was this done as part of the stepwise regression or afterwards?

10. Discuss reasons why several of the 18 SNPs didn't even replicate direction of effect in various other data sets.

11. In discussion, it sounds as if alcohol consumption is known to influence lifespan. A citation for this would be helpful.

12. Supplementary figures could stand for some work:

12a. In supplementary figure 5, the difference between the red and maroon dots needs to be specified in the figure caption.

12b. suppl fig 1: plots are not on the same x axis scale nor include the same features nor in the same order. Plus, no x axis label

12c. suppl fig 3: no axis labels. if based on 1000 randomizations, how do you get p-values upwards of $1E-09$?

13. It is unclear upon reading just the main text whether the authors performed the analysis on lines 387-389, or whether it appeared in the cited paper.

14. The supplementary tables need explanations. It was hard to interpret their results without keys.

15. Figure 2 doesn't have an x axis label.

16. line 288: "fitter" not exactly what you mean. Stick with ascertainment explanation (so as not to equivocate with population genetic selection).

17. The multivariate method sets the intercept to zero, which assumes that the causal effects are not directionally biased, and the authors state that inclusion "did not change the causal effect estimates". Can we see the data for that?

18. It is not immediately clear what the point of the two-sentence paragraph starting at line 391 is. The authors should provide an interpretation of the stated results.

Reviewer #4 (Remarks to the Author):

Overview

This paper identifies SNPs with a causal connection to human longevity working through twelve traits. Joint causal effects of each trait are estimated via Mendelian randomization (MR), a form of instrumental variables.

The overall framework of regression links $b_{i,t}$ from SNP i to trait t followed by links a_t from trait t to longevity is a reasonable choice. Then $\mu_i = \sum_{t=1}^T b_{i,t}a_t$ is a reasonable measure of the longevity effect of SNP i .

The paper has mathematical errors and computational errors. In most places the math needed to check up on the results is missing as well as the logic to support it. There could therefore be many additional undetected errors.

Larger points

1. The article lacks a mathematical presentation of the way the data were handled.

There are many places where the reader just has to guess what was done. In some instances it is easy to guess, in other cases it is not really possible. Even with easy guesses, readers cannot be sure of getting them all correctly.

The notes below are in reference to the 'Online Methods' section. That section should be self contained, but is vague. Part of the problem seems to be that the description is made in a circular order where the narrative depends on facts to be revealed later on. After several readings, it is not clear what the authors have done nor why. Much of the problem is that necessary formulas are not given.

- (a) Line 601 refers to Martingale residuals. Say exactly which ones are used. The book by Therneau and Grambsch mentions four kinds. What fraction of the lifetimes were missing and had to be estimated? Plugging in guesses for 5% of the lifetimes would have very different implications than plugging in guesses for 80% of them.
- (b) Introduce mathematical notation for the Biobank analysis (lines 598–613). Say what the effects were. Maybe estimates of SNP effect on parental longevity? The meta-analysis treats the mothers' and fathers' longevity as independent. Is that reasonable? Their environments would be strongly correlated. Smoking habits (or second hand smoke) would affect both. Using the average of the parents' ages would eliminate the need for meta-analysis and an assumption of independence. That might be better.
- (c) Lines 615–622. What is the extent of the overlap between the two replication studies? Give the number of common cases as well as the numbers in each study. How exactly did the combined GWAS adjust for that overlap? Use mathematical notation.
- (d) Lines 648–653. Were summary statistics imputed for disease GWAs, longevity GWAs, or both? Why was that small study of only 3781 individuals used to define correlations among 2.1 million SNPs? (Maybe it was the only one with enough sequencing.) What formula was used to do the imputation? The imputed Z values could now have quite strong correlations among themselves. How does the later analysis account for their dependence? Maybe highly correlated Z's never co-occur because of LDA thinning, or maybe correlated Z's get treated as if they were independent, double counting information.

- (e) Lines 658–686. Instrumental variables can be tricky to use and several authors have pointed out problems with using them in Mendelian randomization. There are also multiple notions of MR (sometimes describing linkage equilibrium, sometimes describing equal probability of inheriting maternal or paternal alleles). This section should lay out in mathematical notation what the response variable is, what the predictor is, what the instrumental variable is, and then explain why the regression coefficient could be causal in this case. It should also address any reasons why causality might not hold.

Is the causal association established for the coefficients a , for the coefficients b , or both? For instance, the causal effect of education presumably includes any downstream consequences of education such as higher income. How do we know that education is the causal variable instead of something upstream of education that correlates with it?

If a is causal (education causally affects longevity) but the link $b_{t,i}$ from SNP i to education (trait t) is not causal (just a correlation), how do we interpret the contribution $a_t b_{t,i}$? Is there still anything causal in it?

It is usual for instrumental variables to have at least as many instruments as there are regression variables getting a causal interpretation. So what are the instrumental variables for the multivariate causal regression? (They should be named in this section where the causal claim comes.)

- (f) Lines 674–681. I’m completely lost here to the point where it is hard to even formulate a question. Are the p -values on SNPs? Are they from all GWAS or just the trait ones? The paragraph refers to SNPs with non-zero effects but we never know for sure which ones have zero effects and which ones have non-zero effects. Presumably that is based on an estimate, but how exactly? There should be mathematical descriptions. Why divide a Z statistic by an effective sample size, when Z statistics are ordinarily already normalized quantities with a standard error (including a sample size) in the denominator? Why are sample sizes misreported? Can we model the process by which they are misreported? There should be a formula for the effective sample size and an explanation of the logic behind that formula.
- (g) Lines 683–688. Introduce a mathematical symbol for the standardized effects of a SNP pair with appropriate subscripts. Was it $b_{t,i}$ or something else? Use mathematical formulas to describe how summary statistics are used to estimate causal effects. What in fact were the summary statistics? Explain why the logic from reference 40 applies to the present situation (and address or at least acknowledge any caveats). The textual description that follows is not clear.
- (h) Lines 702–703. How exactly was $\text{Var}(a)$ estimated? Was it simply the usual variance formula from the final regression after running AIC and removing nonsignificant predictors? Or did it account for the model search process involving AIC and predictor removal?
- (i) The online methods should describe the multivariate out of sample MR analysis used to include or exclude each study in mathematical notation. Introduce symbols for the variables and parameters used.
- (j) Lines 737–742. This material is not clear. What are the assumptions that make this a prior for a Z statistic? How is the longevity effect size γ_i computed? Is $\gamma = \mu_i$ or is it normalized some way? What is $\text{SE}(\mu_i)$? Is it the same as $\text{Var}(\mu_i)$ or is there a different meaning?
- Say clearly what the null and alternative distributions are, why they have different variances, and what randomness those variances incorporate.
- (k) Lines 753–760. The analysis pipeline has numerous steps. The letter Z is used in several places for different quantities. Where exactly are the Z ’s getting replaced by random ones, one thousand times? What steps in the analysis get recomputed? For instance, do

the imputations with the new random Z s? Or are these Z s from a later stage? Introduce mathematical notation to make it precisely clear what randomization was done, what distribution the random elements came from, and what was computed with them.

It is not clear what the BH procedure was applied to or why.

- (l) Lines 764–766. Does this mean that the same analysis was done on this data as on the Biobank data, or that the same analysis was done as the Joshi reference did? Or maybe both. Were the martingale residuals and survival times used again, or were hazard ratios used? Once again, is it reasonable to assume that the maternal and paternal data are independent? Why would there not be within family correlations in longevity, possibly for non-genetic reasons? How is $10 \log(\text{HR})$ reasonable as ‘years of live’? Reference 42 is cited, but why are we to believe it applies here?

- (m) How do permutations enter the picture? The main article describes permutations of Bayes factors, perhaps 2×10^9 of them. The online methods section does not talk about permutations. It is not very common to permute Bayes factors. What is the underlying logic and how does it apply to the variables in this problem?

Is it possible that permutations were applied to Bayes factors to generate something like a permutation p -value for the Bayes factor to feed into Benjamini-Hochberg? That is quite an unusual approach if that is what was done.

It can be quite laborious to write in a mathematically clear way, but doing so disambiguates the wording and helps to clarify implicit assumptions as well as expose errors. Significant time savings can be had by doing that in LaTeX.

Some of the problems above, such as correlated imputations and the variance of a might not have any good solution in the statistical literature. Instead of an internal proof of validity, one can then appeal to external validation in other data sets that confirm the present paper’s findings. It is still necessary to be precise about what was computed and why. To the extent that problems remain in the data analysis, it becomes more difficult to interpret cases where the present method fails to confirm a discovery in the literature. The failed confirmations could be due to problems in the present analysis.

2. (a) Around line 729 the paper has

$$\text{Var}(\mu_i) = \text{tr}(\text{Var}(a)) + a' * a + b'_{0,i} * \text{Var}(a) * b_{0,i}$$

for $\mu_i = \sum_{t=1}^T a_t * b_{t,i}$. The article does not say what $b_{0,i}$ is here. It might be an hypothesized value for b or maybe it is about some trait 0. Maybe it is an expected value of b , but then why doesn’t that depend on t ? It does not seem that this formula could be correct. The article says that a and b are independent, so they should be random for that to be meaningful. But then the expression above for $\text{Var}(\mu_i)$ itself becomes random which does not seem to fit the rest of the paper. Also it is not clear how $\text{tr}(\text{Var}(a))$ could enter the equation without being b somehow being in there too. To see why, suppose that all the $b_{t,i}$ are always zero. Then the expression above could have $\text{Var}(\mu_i) \neq 0$ even though $\mu_i = 0$ always.

Some algebra shows that for independent random vectors a and b of the same dimension,

$$\text{Var}(a^\top b) = \mathbb{E}(b)^\top \text{Var}(a) \mathbb{E}(b) + \mathbb{E}(a)^\top \text{Var}(b) \mathbb{E}(a) + \text{tr}(\text{Var}(a) \text{Var}(b)).$$

- (b) What is meant by variance in that expression and how is it computed? For example, is variance meant to be sampling uncertainty in those quantities, or are any other sources of uncertainty included? The discussion above mentions some issues about $\text{Var}(a)$. Are there similar issues for $\text{Var}(b)$?

- (c) Explain why a and b are independent. Here a comes from a regression of longevity on traits and b comes from a regression of traits on SNPs.

The fact that so many of the other formulas are simply missing makes it very likely that the analysis has many errors like this one.

3. The effects in Figure 2 are in SDs of lifespan. It would be more interpretable to use years of lifespan. Then the authors can compare their estimate of the effect on lifespan to other parts of the literature, or future researchers can compare to this paper. Similarly, readers can judge whether the given effects are large enough to be interesting but not so large as to be unreasonable.
4. Figure 4 should have a more meaningful axis. For Biobank, the effects are mostly between .005 to .01. If that would be in years, then it would be roughly two to three days. It says that it is in units of standardized effect. If that would be standard deviation of lifetimes we get maybe 30 to 45 days if lifetimes have a standard deviation of about 15 years. Are the units comparable to those in Figure 2? If so then education level at 0.2 is about 20 to 40 times the effect of a typical SNP in Figure 4, which is interesting but we still cannot relate it to anything outside of Biobank. Both figures should use years, or possibly days, of extra life.

The vertical axis of Figure 4 should be given an interpretation. The label says it is a standardized effect for people over 90 versus controls. What does 0.02 mean here? Maybe the SNP gives a 2% greater chance of living to 90. Or maybe if people have (hypothetically) a 30% chance of living to 90 then people with the SNP have a $30 \times 1.02 = 30.6$ percent chance. Or maybe the odds ratio

$$\frac{\Pr(\text{live to 90}|\text{minor LPA})}{\Pr(\text{don't live to 90}|\text{minor LPA})} / \frac{\Pr(\text{live to 90}|\text{major LPA})}{\Pr(\text{don't live to 90}|\text{major LPA})} = 1.02,$$

or maybe the sampling uncertainty of the odds or odds ratio or log odds ratio is involved.

It should be possible for a reader to judge whether 0.02 is too small to be interesting, too large to be credible, or somewhere in between and comparable to other findings.

5. At line 240, 13 of 18 SNPs agree in effect direction, with $p = 0.015$. That matches a binomial tail probability assuming that a direction match has probability 0.5 independently for each of the 18 SNPs. This is a plausible assumption, but might not be right. Maybe there are correlations that make the SNPs match in sign more often than 50% of the time. The authors should check the fraction of matching signs. If it would be 56% then getting 13 matches would only yield $p = 0.049$. This sign match does not appear to be a strong confirmation.
6. Lines 385–389. Describe how these p -values were computed. Name the method and the logic to support that method. For instance with RGS12, p -values of 0.095 and 0.4 combine to 0.037. So something that is insignificant twice in a row (and once not even close) is now significant. Checking with Fisher's statistic $\Pr(\chi_{(4)}^2 \geq -2 \log(.095 \times .04)) \doteq 0.16$ which is not significant. Maybe the authors just multiplied the two p -values which does not yield a correct p -value (that result is close to the reported one and might arise by rounding error). The scatterplot in Supplemental Figure 5 does not show a very strong pattern for RGS12 and Fisher's p -value is not very small.

Smaller points

1. line 592: thorough \rightarrow through
2. line 382–383: I am guessing that for 3 strains of mice there was no high fat treatment arm. Is that how it went?

Reviewers' comments:

Reviewer #1 (Remarks to the Author):

The revised manuscript addressed all of my concerns and is suitable for publication in Nature Communications.

Reviewer #2 (Remarks to the Author):

The authors should be aware that the 90PLUS cohort used in the Fortney et al. paper is exactly the same sample as was used for the EU longevity study. Hence, it might not be the best option to mention them both in the text and Table 2 (and consider them as independent).

Reviewer #5 (Remarks to the Author):

This is a very thoughtful approach and an innovative way to integrate prior knowledge on SNP-disease associations and has the potential to greatly influence future work in the field. I found the overall approach and initial outcomes to be quite interesting and well described. The authors are able to demonstrate global replication success in the CHARGE/EU longevity studies. However, they are not able to comment on the replicability of individual SNP associations. All of the content in the replication, expression, and model system analyses are interesting but fail to definitively demonstrate the validity of the findings.

The replication results are promising but remain overstated. I appreciate the difficulty in identifying appropriate populations for replication meta-analysis given the uniqueness of the UK biobank – and I think that the approach of looking across multiple similar studies is reasonable. However, the results read as a long laundry list of partial replications in disparate studies – including the section on replication in your analysis of other published longevity findings and the expression analyses - and it is quite difficult to evaluate the overall replication of any individual SNP association. Many of the relevant statistics are nicely provided in Table 2 and, from this, one can make some inferences regarding the replication across multiple sources of each of these findings – but this is not evident from the text, which talks in general terms about the pairwise numbers of replicated findings between studies. These observations are not formally combined with statistics or even presented in a manner that enables one to understand how frequently each SNP association replicated across the multiple analyses that were performed.

The gene expression analysis (and methylation analysis?) is not sufficiently described – in the results or in the methods – with enough detail to understand what was done or to evaluate its legitimacy. This is especially true given the general readership of Nature Communications.

I find the caloric restriction study to be too speculative – caloric restriction is not a proxy for longevity and I find these results to be over-interpreted.

It would be interesting to consider how this approach - integrating SNPs based on disease associations - can help to understand genetic cause of co-morbidities (or interactive effects across SNPs causal to different diseases) and how these might influence lifespan.

REVIEWERS' COMMENTS:

Reviewer #5 (Remarks to the Author):

With the replication effort performed by meta-analysis, I am now comfortable that the strength of the findings support publication of this manuscript.

I still find the gene expression and methylation analyses to be speculative and not strictly relevant – but the authors have now done a reasonable job of representing them as such.

One small point:

Perhaps my point with regard to the caloric restriction work was not clear. There is a robust literature that demonstrates a causal affect of caloric restriction on longevity in animal models. However, caloric restriction has a multitude of effects that are not directly related to longevity. Given this, you cannot make the claim that your analysis of gene-specific differential expression changes in CR animals provides any evidence that these genes are causal to the impact of CR on longevity. (Lines 399-401). The analysis is fine but the interpretation is far to strong. This is correlative, not causal.

Reviewer #1

The authors performed a gene association study to identify 18 SNPs associated with lifespan using data from UK Biobank. Most of these SNPs are novel. The approach was to use information from prior studies of diseases in order to calculate weighting coefficients using the Mendelian randomization approach. Validation studies using independent cohorts replicate some of the lead SNPs. Bioinformatic analyses of the 18 lead SNPs were performed to identify linked genes, how the SNPs might affect expression of these genes and how changes in the activity of these genes might affect lifespan. The identification of longevity-associated SNPs using Mendelian randomization is new, and the results for this main analysis are solid.

We thank the reviewer this positive evaluation.

However, the replication and expression analyses are problematic.

We agree with the reviewer's assessment and strengthened the replication analysis and trimmed down the expression follow-up analysis to the most solid core results.

My main concerns are:

1. Conceptual novelty. If a SNP is associated with a disease and the disease is associated with increased morbidity, doesn't it follow that the SNP will show an association with normal lifespan given enough data? It is not surprising that SNPs that increase one's chance of coronary artery disease, Type 2 Diabetes and schizophrenia, etc. are associated with normal longevity. It is admirable that this study found 18 SNPs with significant association with normal longevity, but eventually all of the disease-related SNPs will show an association if the amount of longevity data is at a sufficient level.

We believe that simply pinpointing lifespan-associated SNPs in the genome is not a very useful exercise, as most of the genome is associated to complex traits to some level (a key assumption of successfully applied mixed effect models). Stating that all disease-associated markers are also lifespan-linked is probably valid, but only to some extent. Our paper goes far beyond these trivial statements:

- 1. We can *estimate the effect of these SNPs* on parental lifespan. Our Table 2 also includes a directly interpretable measure of the effect in years of life lost.**
- 2. While prior estimates of effect size based on the SNPs contribution to diseases correlate reasonably ($r=0.24$) with actual effect on lifespan, *for most SNPs there is an enormous gap between disease and lifespan effect*. This is partially due to not having large enough disease GWAS studies (to boost the accuracy of prior calculation), and partially to not covering many diseases or other risk factors that are key for lifespan. This experiment is crucial to demonstrate how little we know about lifespan, and how much more complex lifespan is than a simple summation of life-threatening diseases.**

3. As you can see from our top hit list – apart from FTO and APOE – the SNPs we pick up do not coincide with SNPs with the strongest disease-association. Thus the naive “a disease SNP is a lifespan SNP” approach could not even rank disease SNPs in terms of importance.
4. Importantly, 3 of the 16 hits are not associated with any of the 12 traits at genome-wide significant level. These would have not been discovered with the simple disease SNP=longevity SNP argument. Moreover, in the future with more disease GWAS available our method will be able to discover more and more pleiotropic variants that do not pass strict genome-wide significant threshold for any of the studied disease traits, but have tiny effect on several traits at the same time.
5. SNPs have pleiotropic effects, thus knowing the effect of a SNP on one disease does not necessarily tell us the directionality of the effect on lifespan. Even looking at our top hits we see a rich pleiotropy (shown in the revised **Supplementary Figure 4**):

We agree that these arguments were not laid out clearly in the original submission and hence we have added these points to the Discussion:

“One could argue that any SNP linked to a trait causally affecting lifespan will eventually be associated with lifespan. Our study points far beyond this basic statement: First, we provide effect size estimates of the identified SNPs on various scales. Second, three of our top hits are not

associated at genome-wide significant level with any individual disease and only two of them are top hit for some disease. Third, most of the 16 loci exhibit pleiotropic effect (Supplementary Figure 3) and the actual effect size estimates of the discovered SNPs are quite far from both the priors and their individual effect on a trait (multiplied by the impact of the trait on lifespan) (Supplementary Figures 4-5)."

The disease GWAS returned two surprising results: 1) education attainment was the strongest factor for increasing longevity and 2) major depressive disorder increased, rather than decreased, longevity. I recommend that both these results be discussed in more depth. Education attainment likely indicates access to better health care, rather than having a biological mechanism such as reduced risk for diseases of morbidity. Is there a way to rationalize the positive effect of major depressive disorder?

It is important to point out that the larger European MDD study did not show significant impact on lifespan. Upon excluding the Chinese MDD study, the MDD trait was no longer selected, leaving us with 11 (instead of 12) traits, slightly changing the priors. We re-ran the entire analysis with the new priors, which resulted in two borderline top hits' failure to pass the 5% FDR threshold, leaving us with 16 loci. We also interpreted the meaning of educational attainment's involvement in longevity:

"Educational attainment may have shared genetics with socio-economic status, which is a more plausible underlying cause."

And included the Marioni et al 2016 paper (recommended by reviewer #3) to further support this strong link.

2. Replication in independent studies. A weak point in the paper is that the replication results are over sold. Two out of 18 lead SNPs were validated in the CHARGE and EU longevity consortia using the Bonferroni threshold. Evidence that 6/18 lead SNPs had $P < .05$, that 6/18 is more than expected by chance and that 16/18 had effects in the same direction are only weak evidence that there is true signal amongst the 18 lead SNPs. The interpretation of the replication study should be more rigorous and conservative.

We fully agree with the reviewer, our choice was based on statistical power – essentially we do not have a large enough replication sample with good power to replicate most of the findings. We performed *power calculations* to illustrate that the studies used for replication are massively underpowered, but that this is the best one can do with the currently available data. We describe it now in the results:

"Given the large sample size of the UK Biobank compared to other previous independent studies we first assessed power to replicate our findings in the combined CHARGE studies. First, we performed power calculations assuming a replication study with 14,000 samples selected from the top and bottom 5% tail of a general lifespan distribution and a SNP explaining ~1% of lifespan variability. These calculations showed that apart from the top two hits (*APOE* and *CHRNA3/5*) we have only 16% (for the *FTO* variant) to 73% (for the SNP near *BSND*) power to replicate our findings with a P-

value below 0.05/16 in a study very similar to the combined CHARGE data set. For this reason we call our analysis as a *look-up*, given that strict replication is impossible due to power considerations."

It is all the more convincing that despite the low power we still managed to replicate the majority of the findings.

Furthermore, We have obtained replication data from another publication¹ for two cohorts (NECS and 90PLUS). These results strengthen our replication success in the 90PLUS study, five of our 16 SNPs reach a P-value surviving Bonferroni correction (0.05/16), plus a further two in the NECS study. These new lookups have been added to Table 2. We added these new results and a more conservative summary sentence to the manuscript:

"Remarkably, six of our top hits (in/near *BSND*, *CELSR2*, *LPA*, *CDKN2BAS*, *FTO* and *APOC1*) survived Bonferroni correction (0.05/16) in the 90PLUS cohort of the recent disease-informed extreme longevity GWAS¹ (Table 2). For the NECS cohort in this study there were only 243,000 SNPs available so we chose the best available proxy for our 16 top hits to assess replication performance (Supplementary Table 3). In this replication cohort, seven of the 16 hits survived Bonferroni correction.

In summary, 12 out of the 16 hits replicate in at least one of the replication studies with nominal significant one-sided P-value and 9 of these (in/near *BSND*, *CELSR2*, *FTH1P5*, *LPA*, *CDKN2BAS*, *CHRNA5*, *NPIP8*, *FTO* and *APOC1*) survive Bonferroni correction (0.05/16)."

3. Mouse expression studies (p11, line 380-389). The mouse expression analysis is flawed.

a. The analysis compares expression in different BXD or LXS mouse strains. This is not the same as showing the effects on lifespan of raising or lowering expression in one strain.

b. Does expression of the three genes in Sup. Figure 5 act as QTLs for lifespan in the mouse strains?

c. How many genes in the genome show differences in expression between the different mouse strains? How surprising is it that some of the eight genes analyzed would show differences in expression in the mouse strains given the overall number of genes that differ in expression between the strains?

d. What were the criteria to select 8 genes for analysis out of all of the genes near the 18 lead SNPs?

f. p11, line 385 "Expression of AMIGO1 and RGS12 showed P-values <0.05 (132 - 172 days of lifespan increase / expression unit, one-sided =0.006, 0.037, respectively)." I don't understand where these values come from. RGS12 does not show a significant p value for either the red or brown dots. AMIGO1 shows a significant p value only for the red dots.

g. Supplemental Figure 5. What are red and brown dots? What is the y axis? What are the units of the X axis?

h. P11, line 393 CDKN2B and LAT mouse knockout results are not valid comparisons. The knockouts might die because of the trivial reason that the genes are essential, not

because the knockout is mimicking the longevity effect in humans.

It is a valid point and now we reran the mouse expression vs lifespan association analysis correcting for genetic background (population structure), using mixed models (via the EMMA software²). The new analysis confirmed that the impressively strong association between *RBM6* expression at 72 days of age with lifespan was not due to unaccounted population stratification issue. We updated the results:

“The significant correlation persisted even after correcting for population structure through mixed effect models² ($P=1.1 \times 10^{-4}$). The other two genes showed no significant association with lifespan.”

This is not unexpected as the LXS cohort is an F2 derived family rather than an admixed cohort. Relatedness does not differ among strains more than expected by random segregation. Moreover, the vast majority of genes show no significant association with lifespan, thus seeing such an extreme deviation as *RBM6* is very significantly sticking out of the haystack.

Since the follow-up human analysis has been reduced to the most stringent ones, we do not have *RGS12* and *AMIGO* as candidates, only three genes (*RBM6*, *CHRNA5* and *SULT1A1*) emerged from the expression-lifespan MR experiment.

We also reran the caloric restriction comparison analysis in mice focusing only on brain tissues, as the three causal genes were emerging from MR analysis using brain eQTLs. This analysis revealed a very strong down-regulation of the *SULT1A1* gene in neocortex (the only available brain tissue) upon caloric restriction, concordant with our finding that low expression may be causally extending human lifespan.

“Notably, *SULT1A1* expression levels are down-regulated upon CR diet in mouse (GSE11291), similarly our human study showed lower expression may be linked to extended lifespan (Supplementary Figure 10).”

Following the reviewer’s comments, we focused only on the clearest three genes that emerged from the MR analysis and sought replication only in the same (brain) tissue, where they showed causal effect in human.

P8 line 242. “none of their 14 top hits reached nominal significance level in our study”. The P values in Supplemental Table 2 seems to indicate that all 14 are significant.

The reviewer looked at the column that represents the P-value of the cited study, not in our study. We have now updated that table (now Table S4) and the last column is the Z-statistic in our study. It is clear that none of those Z-statistics are larger than 1.95, thus none of the SNPs are nominally significantly associated in our study.

P8 Line 266. “Despite the winner’s curse bias, as a general trend these SNPs seemed to have larger effects on extreme longevity, especially at the APOC1 locus.” The data for extreme longevity vs lifespan should be shown.

The data is shown in the Figure and in Table 2 (UKBB Z-stat, Walters Z-stat, CHARGE Z-stat, Fortney 90 PLUS, Fortney NECS columns describe our UK Biobank based study and all replication studies).

Table 1. Define effect size.

Done.

Table 2. Column headings should be defined explicitly.

All column names are now explained in the legend.

Figure 6. Give effect size not just p values. What is fold-change in expression for the A allele, and how much does expression of this gene change per year?

This Figure is gone to focus on the most solid results only.

Supplemental Table 1. Column headings should be defined.

Added in the second tab.

Supplemental Table 2. What are P and UKBB Both?

We expanded the column labels.

Supplemental Table 3. Beta, P and N SNPs should be defined.

We expanded the column labels.

Supplemental Table 4. Outcome, Beta should be defined.

This analysis is removed as was only tangentially related to our follow-up.

Supplemental Table 5. This is the wrong table. Text indicates that it has data about 35 mouse strains, but Table has data on three human SNPs. Column headings should be defined.

This is now supplementary Table 6 and the text is modified: this shows the correlation between the expression of the three focal genes (in mouse brain) and lifespan in mice.

Supplemental Figure 6. What are the units for the y axis? What is H? Where are values for mouse and fly? What are the names of the genes in the other species? Where are data for LAT expression?

This is now Figure S10 and expression is (as always) in arbitrary units, labels are expanded. No relevant tissue specific expression data was found for fly/worm.

Reviewer #2

The paper by McDaid et al. report the association of several SNPs with human 'lifespan' based on a pre-informed GWAS on parental age at death. They subsequently show that several of the genes associated with these SNPs are also associated with differential gene expression with age in human peripheral blood and with lifespan in mice.

I really like the approach taken by the authors and I think it could really help to move the field of genetics of human lifespan forward. However, some important details regarding the analyses are missing and the use of terminology is not always correct.

Major comments:

1. The authors use the term 'lifespan' throughout the document. However, sometimes they actually refer to parental age at death (or parental lifespan), which is not the same as the lifespan of an individual. This should be changed accordingly.

See for example:

Line 104: "associated with lifespan"

Line 115: "SNP-phenotype effect sizes and the trait-lifespan"

Line 118: "lifespan in a general population cohort"

Line 131: "lifespan (see Online Methods)"

Line 143: "Causal effects of GWAS traits on lifespan"

Figure 1

We changed at several instances, but it often breaks the flow of sentences, so we also added a sentence to the beginning of the Results to clarify:

"For simplicity, in the following, we will use the term lifespan to refer to parental lifespan."

2. It is well-known that some of the phenotypes identified to be associated with 'lifespan' show an opposite association with mortality later in life (>85 years) as compared to middle age, see for an example van Vliet et al. *Gerontol A Biol Sci Med Sci.* 2010. I think this should be taken into account in the analyses performed by the authors.

It is a good point, we were not aware of this result, we added the reference and a comment that this age range represents only a small fraction of the Biobank samples' parents:

"Furthermore, some risk factors (lipids, blood pressure, obesity and diabetes) have shown opposite effect on lifespan when focusing on longevity only beyond 85 years ³, but less than 15% of UK Biobank parental lifespans fell into this category, hence this phenomenon could only mildly worsen our prior, which was based on younger populations."

3. There is some important information lacking concerning the replication studies. The authors mention that "SNP were tested for replication using summary statistics from the combined CHARGE and EU longevity studies". They have to describe this in more detail in the Methods section. For example, which summary measures were used (Beta and SE's or P-values)? Moreover, the results of the individual studies should, at least, be

shown in a Supplementary Table to determine if the associations are not driven by only one of the two studies.

We have now given a detailed explanation how to two studies were combined and SNPs showing significant Cochran heterogeneity P-value (<0.1%) were dropped:

“We meta-analysed these two studies, via inverse-variance weighted fixed effect meta-analysis adjusted for correlated test statistics⁴ due to the fact that there are a small number of individuals that are in both studies, to construct a single GWAS.”

4. The results of the look-up in the GWAS of Walter et al. and Fortney et al. should be shown in a Supplementary Table (lines 238-242 and 247-250).

Thank you for the comment, this was indeed missing. We added now these look-up results to Table 2.

5. It is unclear how the authors determined the genetic correlation between the GWAS on parental age at death and the combined CHARGE and EU longevity GWAS / survival GWAS. They should explain this in more detail in the Methods section. The fact that the most important loci from both studies ($P < 10^{-5}$) are not replicated between them, with the exception of ApoE, makes me question if the provided number is actually correct. It seems that parental age at death is a suboptimal proxy for longevity and, thus, healthy aging.

We explained that it was calculated using the well-known LD score regression method:

“To check to what extent the genomic landscape of lifespan and extreme longevity overlap, we applied LD-score regression⁵.”

6. How do the authors explain the fact that three out of the four SNPs that show an age-related eQTL effect could not be replicated in the CHARGE and EU longevity GWAS, survival GWAS, and the extreme longevity study? It seems that these SNPs might actually not be that relevant for lifespan. This is further confirmed by the fact that the expression of these genes is not associated with parental age at death in the CoLaus study. Why have the authors decided to take these genes forward for the analyses in model organisms?

We do not have a large enough replication sample with good power to replicate most of the findings. To demonstrate this we performed *power calculations* to illustrate that the studies used for replication are massively underpowered, but that this is the best one can do with the currently available data. We describe it now in the results:

“Given the large sample size of the UK Biobank compared to other previous independent studies we first assessed power to replicate our findings in the combined CHARGE studies. First, we performed power calculations assuming a replication study with 14,000 samples selected from the top and bottom 5% tail of a general lifespan distribution and a SNP explaining ~1% of lifespan variability. These calculations showed that apart from the top two hits (*APOE* and *CHRNA3/5*) we have only

16% (for the *FTO* variant) to 73% (for the SNP near *BSND*) power to replicate our findings with a P-value below 0.05/16 in a study very similar to the combined CHARGE data set. For this reason we call our analysis as a *look-up*, given that strict replication is impossible due to power considerations."

It is all the more convincing that despite the low power we still managed to replicate the majority of the findings.

Furthermore, We have obtained replication data from another publication¹ for two cohorts (NECS and 90PLUS). These results strengthen our replication success in the 90PLUS study, five of our 16 SNPs reach a P-value surviving Bonferroni correction (0.05/16), plus a further two in the NECS study. These new lookups have been added to Table 2. We added these new results and a more conservative summary sentence to the manuscript:

"Remarkably, six of our top hits (in/near *BSND*, *CELSR2*, *LPA*, *CDKN2BAS*, *FTO* and *APOC1*) survived Bonferroni correction (0.05/16) in the 90PLUS cohort of the recent disease-informed extreme longevity GWAS¹ (Table 2). For the NECS cohort in this study there were only 243,000 SNPs available so we chose the best available proxy for our 16 top hits to assess replication performance (Supplementary Table 3). In this replication cohort, seven of the 16 hits survived Bonferroni correction.

In summary, 12 out of the 16 hits replicate in at least one of the replication studies with nominal significant one-sided P-value and 9 of these (in/near *BSND*, *CELSR2*, *FTH1P5*, *LPA*, *CDKN2BAS*, *CHRNA5*, *NPIP8*, *FTO* and *APOC1*) survive Bonferroni correction (0.05/16)."

Finally, we restricted our follow-up analysis to those with evidence for causal implication, not just co-localisation of eQTL, lifespan-QTL signals. This yielded three key genes only, which we now followed up, two of which showed convincing evidence in mice.

7. Why would the authors perform the MR-analysis described from line 352 to 359 if they already showed that the lead-SNPs at the identified loci are no eQTL for the genes they are looking at? The authors should explain the reasoning behind this analysis in more detail or remove it from the manuscript.

As mentioned above we simplified the analysis and dropped the weaker co-localisation analysis and performed only rigorous MR analysis, which pinpointed three key genes:

"We applied MR to test whether the expression of certain genes at the 16 lifespan-associated loci may be causally driving lifespan. We investigated 91 gene-tissue pairs (equivalent to 57 independent tests) at the 16 loci that had more than five independent eQTL SNPs in a GTEx⁶ tissue. Three genes (*SULT1A1*, *CHRNA5* and *RBM6*) showed significant negative causal effects (all in brain) on lifespan (see Supplementary Table 5)."

8. I think the part on the model organisms is a really nice addition, but it is currently too condense and the majority of results are not shown in a Supplementary Table or Figure.

On the other hand, the Tables and Figures that have been created (Supplementary Table 5, Figure 5 and 6) are unclear and lack detail in the description (especially Supplementary Figure 6; how did the authors combine the fly and mice data here and why is LAT missing from this Figure). In addition, the selection of genes taken forward to this stage is questionable (see point 6 and 7).

Only the most convincing animal data (caloric restriction and lifespan vs expression in brain at 72 days) are kept now, also the gene list is reduced to the three most robust ones. We made the respective Figures and Tables clearer, explained the table column labels and figure labels.

9. The authors should explain why they would look at the expression of the genes in liver in mice. How can they be certain that this is comparable to expression of the gene in human peripheral blood? Do they have evidence (for example from GTEx) that liver is a good proxy for peripheral blood?

Indeed, switching tissues/organs is not a clean approach. Instead we only present now results that are produced in the same tissue as the discovery, namely brain.

10. The format of the Supplementary Tables and Figures is really sloppy and some are almost impossible to read. For example, it is unclear what the column-headings in Supplementary Table 1 actually represent. In addition, three of the panels in Supplementary Figure 1 contain some strange lines on the left side of the forest plots.

Sorry for this sloppiness, we clean them up now.

Minor comments:

- Line 30: The authors refer to the CHRNA3/5 locus as a locus robustly associated with human lifespan. However, I do not agree with this. The locus has only been found in two papers based on the same population and have not (yet) been validated in a large independent lifespan-related study. This is exemplified by the fact that the authors were unable to replicate this locus in the combined CHARGE and EU longevity studies. The authors should therefore not mention this locus as robustly-associated here. On the other hand, the locus near EBF1 has been replicated in an independent longevity GWAS (Zeng et al. Scientific Reports 2016) and could therefore be added here.

We do not see how that paper by Zeng et al. [<https://www.ncbi.nlm.nih.gov/pmc/articles/PMC4766491/>] confirms *EBF1*, as none of their top hits lie on chromosome 5. But agree, that CHNRA3/5 is yet to be confirmed, so we do not call it robustly-replicating in the revised version.

- Line 61: the “a” is missing between “have” and “strong”.

This sentence has been rephrased to shorten the introduction.

- Line 85: The authors mention that one could identify “longevity-associated” loci using survival analysis. However, this is not the case, since one is looking at a decrease or increase in frequency per year, which is not the same as longevity. They should change this into “lifespan-associated”.

Thank you, it is now changed.

- Line 104: Was the study on parental age at death able to find association of the locus near EBF1? If not, the authors should also mention this here for consistency.

We added *EBF1* to the sentence, no association was found.

- Line: 187: The authors should also provide a lambda belonging to Supplementary Figure 3.

We added the genomic lambda to the text:

"In the QQ-plot, the permutation based BF P-values show good adherence to the null distribution for ~99% of the selected SNPs with genomic lambda of 1.002 (Supplementary Figure 7)."

- Line 204 to 206: This is a strange comment. The authors selected SNPs with a P-value $< 10^{-5}$ to be considered associated with a trait and not $< 5 \times 10^{-8}$, so why would this observation be worth mentioning?

The P-value threshold for selecting instruments for estimating the causal effects to derive priors for each SNP does not play any role in the calculation of the BF. Of course SNPs with some evidence for disease association have stronger priors and hence have more chance to be picked up. The key point here is that some yet to be disease-associated-SNPs can be picked up in our study.

- Line 307: "EBRF1" should be "EBF1".

Thank you.

- Line 327: Please add "peripheral blood according to" between "in" and "GTEx" for sake of clarity.

This sentence is now changed and clearer. For consistency we only use GTEx data.

- Line 345-350: I do not see the added value of this analyses and I think it should be removed from the manuscript. If the authors want to keep them in they should show the results of these analyses in a Supplementary Table or Figure.

We think this is interesting information to know that probably age-related expression or methylation biomarkers are mostly not causally linked to lifespan. Since it is five lines only, we prefer to keep it.

- Why was C-reactive protein not included in the list of age-related traits? It is well-known that this trait is associated with lifespan.

We checked this study, but to our surprise it did not show strong multivariate causal effect on lifespan.

- The authors should refer to rs4420638 as APOE and not APOC1 throughout the manuscript. It is well-known that this SNP tags the effect of ApoE e4.

We refer to it now as “APOE/APOC1”.

- Supplementary Table 2: “Walters” should be “Walter”.

Corrected.

- I think Supplementary Table 5 is missing (described from line 380-385). Please add this Table.

This is now Table S6 and has been updated to show the association.

Reviewer #3

The manuscript by McDaid et al. describes a GWAS-like Bayesian framework to identify loci associated with age of parental death through increasing disease risk. They apply their method to data obtained from the UK Biobank study, which contains genotype and lifespan information for over 100,000 individuals. Through combining this data with previously performed GWAS, they first derived priors for association of all HapMAP SNPs with longevity using a Mendelian Randomization-esque framework, then combined this information with the SNP/parental longevity associations to calculate posteriors of each SNP being associated with longevity. They then work to show that several of their genes may influence lifespan in model organisms and propose mechanisms of action.

Previous studies have found only a few genetic variants that robustly associated with longevity (an area the authors must know is somewhat contentious). To improve discovery power for longevity variants, this paper uses a Bayesian approach to prioritize variants that are associated with disease known to influence lifespan. This is simultaneously a strength and the weakness of the paper. Specifically, it is not surprising that traits associated with premature death -- for example, Coronary Artery Disease (CAD) and causally related traits to CAD like low-density lipoprotein cholesterol -- would be associated with longevity (and that risk alleles here shorten life span).

We believe that simply pinpointing lifespan-associated SNPs in the genome is not a very useful exercise, as most of the genome is associated to complex traits to some level (a key assumption of successfully applied mixed effect models). Stating that all disease-associated markers are also lifespan-linked is probably valid, but only to some extent. Our paper goes far beyond these trivial statements:

- 1. We can estimate the effect of these SNPs on parental lifespan. Our Table 2 also includes a directly interpretable measure of the effect in years of life lost.**
- 2. While prior estimates of effect size based on the SNPs contribution to diseases correlate reasonably ($r=0.24$) with actual effect on lifespan, for most SNPs there is an enormous gap between disease and lifespan effect. This is partially due to not having large enough disease GWAS studies (to boost the accuracy of prior calculation), and partially to not covering many diseases or other risk factors that are key for lifespan. This experiment is crucial to demonstrate how little we know about lifespan, and how much more complex lifespan is than a simple summation of life-threatening diseases.**
- 3. As you can see from our top hit list – apart from FTO and APOE – the SNPs we pick up do not coincide with SNPs with the strongest disease-association. Thus the naive “a disease SNP is a lifespan SNP” approach could not even rank disease SNPs in terms of importance.**
- 4. Importantly, 3 of the 16 hits are not associated with any of the 12 traits at genome-wide significant level. These would have not been discovered with the simple disease SNP=longevity SNP argument. Moreover, in the future**

with more disease GWAS available our method will be able to discover more and more pleiotropic variants that do not pass strict genome-wide significant threshold for any of the studied disease traits, but have tiny effect on several traits at the same time.

SNPs have pleiotropic effects, thus knowing the effect of a SNP on one disease does not necessarily tell us the directionality of the effect on lifespan. Even looking at our top hits we see a rich pleiotropy (shown in the revised Supplementary Figure 4).

We agree that these arguments were not laid out clearly in the original submission and hence we have added these points to the Discussion:

“One could argue that any SNP linked to a trait causally affecting lifespan will eventually be associated with lifespan. Our study points far beyond this basic statement: First, we provide effect size estimates of the identified SNPs on various scales. Second, three of our top hits are not associated at genome-wide significant level with any individual disease and only two of them are top hit for some disease. Third, most of the 16 loci exhibit pleiotropic effect (**Supplementary Figure 3**) and the actual effect size estimates of the discovered SNPs are quite far from both the priors and their individual effect on a trait (multiplied by the impact of the trait on lifespan) (**Supplementary Figures 4-5**).”

Even though this causal path is known, the paper missed at least one mark where this case is fairly strong: e.g., in contrast to AMIGO1, SORT1 is by far the stronger candidate and story (see below).

See below, where we discuss this in details.

Another challenge is that the paper brings together several analysis, but with unclear narrative. Some results did not seem relevant to the central story (e.g., the LD score regression analysis). Other analysis components were sensible to mention, however the results were either negative, the evidence supporting them was quite weak, or they were confusing (the gene expression analysis in model systems). Thus, it was a challenge to deduce what the new contributions were from the authors that was backed with convincing evidence about the genetics of longevity.

We fully agree with the review and trimmed the follow-up experiments, focusing on the most robust ones. Following the comments the reviewer, (1) we removed mouse KO experiment results and nominally significant results. (2) We removed the simple overlap analysis between the 16 lifespan hits and age-related expression QTL SNPs, as these do not represent solid causal implications of those genes in lifespan modulation. (3) We focused only on the clearest three genes that emerged from the MR analysis with multiple strong instruments and sought replication only in the same (brain) tissue, where they showed causal effect in human.

Specific comments are provided below.

Major Comments

1. The overall impact of this study is unclear.

a. As stated by the authors in their introduction, it has already been established that certain diseases play a major role in shortening longevity. In addition, through GWAS and other means, loci influencing risk to those disease have been found. If a disease shortens longevity, then it seems like most loci contributing to it will influence longevity. What is the significance of picking out only a subset of these loci through complex methods and pointing out that they contribute to longevity? The authors do not provide a specific rationale or impact. If the authors believe the basic inference from the MR experiment, does it not follow that most (if not all) loci that modulate a biomarker then cause shortening of lifespan?

b. Given that these loci are established (i.e., genome-wide significant) for their respective trait, the observation that most of the discovered loci 'replicate' in independent datasets for longevity is good, however doesn't feel that surprising. What is the new biological insight revealed here? Additional discussion and clarification of the goal and impact of this study would greatly strengthen the report.

We addressed both comments above and added several points to the Discussion. The key argument is that while the emerging loci themselves may not be that surprising, we can estimate their actual effect on lifespan precisely and found an enormous gap compared to the scenario where they simply act through diseases. Second, many of these SNPs have pleiotropic effect on several traits, again the impact of such SNPs on lifespan is not straightforward. Third, we would like to stress that mortality and morbidity have different genetic basis. While disease-predisposing SNPs are directly impacting morbidity, this does not necessarily affect mortality. These processes appear to be far more subtle and one of the major outcomes of our paper is the observation how surprisingly little (~25%) mortality and general disease morbidity share in terms of genetics. It is also reflected in the fact that the ranking of 16 discovered variants (in terms of their effect) is completely different between lifespan and among diseases. Moreover, many risk factors for morbidity can be reduced via medical care/intervention and do not translate to risk for mortality. This is why our priors are very far from perfect, but still good enough to improve power substantially. We clearly set out the aims in the updated manuscript:

"The key aims of our study are to (i) discover new variants impacting mortality and doing this by developing a mechanistic model to estimate morbidity priors for each SNP; (ii) understand through which disease-predisposition they act; (iii) reveal the discrepancy between the life-shortening effect of a SNP and its effect on various life-shortening diseases; (iv) shed light on transcriptome biomarkers that may be causally involved in this process; (v) elucidate whether these processes are shared between mice and human. "

In terms of broader impact, we are tackling here the most important human traits of all, being alive.

2. Concerns about generation of the priors, and the priors themselves.

2a. Validity of the selected features. Essentially, the authors are performing a type of feature selection (machine learning) to determine which risk factors go into their downstream analysis. But how do the authors know that the selected features are best, the weight for those features are accurate, and that the models selected are valid and generalizable? Typically, in the “machine learning” space, cross-validation, true holdouts, etc. would be used to demonstrate the robustness, accuracy, goodness of fit (neither over, nor under-fitting). While it is appreciated that this isn’t a central focus of the paper, it is crucial to the applicability of the approach in general terms. But also specifically here: if the priors are over fit, the subsequent association analysis will require less evidence for sites with nominal associations (because the prior is stronger), increasing false positives. The authors should perform cross-validation, or a similar approach, to show that the priors would remain stable if different, but similar, disease traits were used.

Note that the chromosome holdout experiment does not address this issue (this addresses a different question, not on the features selected).

We performed *cross-validation experiment* to show that the priors are stable. We described these analyses:

“In order to provide further evidence that these causal estimates yield robust priors, we performed MR using instruments only from chromosomes 1 to 6 and another MR analysis with instruments only from chromosomes 7 to 22. Although the selected traits did not perfectly agree (Supplementary Table 2), both the causal effect sizes and the resulting priors were highly concordant (Supplementary Figure 2).”

The prior calculation can also be viewed as a prediction exercise: Comparing SNP effects on various disease traits (predictors) with those on lifespan (dependent variable), using 21 chromosomes only, we try to predict what lifespan effect we would see for SNPs on the 22nd chromosome given the predictors. *This justification holds regardless of whether the causal effects are correctly estimated, and whether there is any causal relationship between those traits and lifespan or not.*

Finally, we also emphasized that *even if the causal effect estimates and hence the priors are incorrect, our findings remain valid* thanks to the permutation procedure. This has been added to the text:

“While many of the obtained causal effect estimates seem plausible and in line with literature evidence, we do not claim that these causal effect estimates are unbiased, neither that no instruments violated MR assumptions. This step was necessary only to obtain sensible priors for the downstream analysis. If these estimates are wrong, our prior is less informative, yielding fewer discoveries.”

2b. Several correlated, and possibly causally connected traits, such as cholesterol, BMI and Type 2 diabetes, are risk factors selected by MR analysis. A previous paper on multivariate MR (Burgess and Thompson, 2015, PMID 25632051) indicates that causal relationships between risk factors can cause widely varying effect estimates when regression methods are used. Isn't the model double counting here, in the selection? In addition, what happens with SNPs that have opposing effects on multiple traits? The complexities here are numerous, and are not very well described in the methods, but impacts interpretation of the features selected and if the model is good or overfit (see comment #2a, above).

The regression-based method presented in (Burgess and Thompson, 2015⁷) is a stepwise regression method, while we apply multivariate regression in one step. The difference is that the method they refer to calculates the conditional effect and our gives the joint effect. This explains why the conditional regression-based estimates (in Burgess and Thompson, 2015⁷) are consistently downward biased.

This was not clearly outlined in the original Methods section, but we have now expanded it substantially and the equations are explicitly stated. We show that our regression-based estimate is derived from the 2 stage least-square MR estimator. Further, we have implemented the likelihood-based methods of Burgess and Thompson, 2015⁷, which yielded identical results to ours:

"Note that Burgess et al.⁷ introduced a maximum likelihood (ML) based MR method using only summary statistics. Since the lifespan study (UK Biobank) does not share samples with the disease trait studies, our formula (Eq. 2) yields the same result conditional on the SNP-trait estimates having little variance. To make sure that our method gave sensible results, we implemented the ML-based MR method by Burgess et al.⁷ and confirmed that we obtained indistinguishable solution to their likelihood function (Eq. 1 in their study) maximisation (see **Supplementary Table 9** for the causal estimate comparison and the between-trait correlation estimates (ρ in their paper) are in **Supplementary Table 10**)."

2c. A quantitative measurement of the similarity in the causal-effect (alpha) values estimated in each of the 22 leave-one out chromosome sets should be provided to reinforce that this method is robust.

Figure 2 shows that all leave-one-out estimates fall within the confidence interval of the overall (using all chromosomes) estimate, thus none of them are statistically significantly different, except the leave-chr15-out for smoking, which is due to the fact that the biggest hit (near *CHRNA3/5*) is on chromosome 15.

2d. While authors state that they are not using the MR results in a causal framework, they indirectly are, in interpretation. Some of those could indeed make sense, too. The paper calls these "causal" effects - and I'm not certain I agree with that. Robustness, sensitivity, confounding - these things have not been worked through on these causal effect estimates. There a bit of an art form and effort to making strong instruments to build a causal case (risk score analysis, sensitivity, heterogeneity, robustness analyses), and unsure if that has been convincingly shown here.

We agree that ensuring that an MR analysis yields true causal estimates is extremely complicated. We did our best to perform as many sanity checks as we could think of. We have now performed an out-of sample causal effect selection (see above). We also verified that none of the SNPs have strong impact on lifespan independent of these 11 traits, which would violate the third MR assumption:

“In addition, we examined the MR residuals to ensure that there was no instrument with indication to violate the third MR assumption, i.e. that the instrument must be independent of disease-lifespan confounders. If such violation occurs, the instrument would have a disease-independent (pleiotropic) impact on lifespan and hence the observed SNP-lifespan effect would deviate substantially from the prior effect. We observed only a slight deviation for one instrument (a SNP near *APOE*, rs7256200) on the residual qq-plot for all the 22 leave-one-chromosome-out estimation (**Supplementary Figure 3**). Due to the large number of instruments, excluding this single SNP does not change noticeably the results.”

Importantly, the prior calculation can also be viewed as a prediction exercise: Comparing SNP effects on various disease traits (predictors) with those on lifespan (dependent variable), using 21 chromosomes only, we try to predict what lifespan effect we would see for SNPs on the 22nd chromosome given the predictors. *This justification holds regardless of whether the causal effects are correctly estimated, and whether there is any causal relationship between those traits and lifespan or not.*

On top of it, *even if these estimates are completely off and overfitted, our permutation procedure excludes any possibility of observing false associations due to incorrect prior specifications.*

“While many of the obtained causal effect estimates seem plausible and in line with literature evidence, we do not claim that these causal effect estimates are unbiased, neither that no instruments violated MR assumptions. This step was necessary only to obtain sensible priors for the downstream analysis. If these estimates are wrong, our prior is less informative, yielding fewer discoveries.”

2e. What is potentially confusing or hard to wrap around is the result on educational attainment. The ‘causal graph’ is unclear. I know the authors will say that the /specific/ features don’t matter; but an implication that *could* be made is that educational attainment increases lifespan - which clearly doesn’t make a lot of sense, unless educational attainment itself was correlated with other factors that did have an impact on lifespan (alcohol use, poverty, etc.). I am vaguely aware of papers in this area (Marioni et al 2016, PNAS), so given that this is the most substantial contributor, it may be worth having some discussions on the strengths and limitations here.

We also interpreted the meaning of educational attainment's involvement in longevity and included the Marioni et al 2016 paper to further support this strong link:

"Educational attainment may have shared genetics with socio-economic status, which is a more plausible underlying cause. In line with our finding, a recent study demonstrated that genetic risk score for education level is a strong predictor of longevity⁸."

As it is the case with many MR analyses, we might identify a trait that is genetically strongly correlated with the true causal risk factor.

3. Concerns about the section on the model-systems 'validation' experiments. Overall, this component seemed unconvincing, undermining the message that these loci are significant effectors of longevity.

3a. The clearest example of that is rs646776. This is the strongest eQTL in the genome (for liver), and the underlying, previously implicated gene is *SORT1*, not *AMIGO1* as the authors state, for LDL-C. See Musunuru 2010, PMID: 20686566.

The clearest hypothesis is that the effect on longevity is mediated through LDL-C, which increases risk for CAD (which this variant is also strongly associated with), and premature death from myocardial infarction. Yet, the authors fold this into a case where the expression is correlated (in blood) with another gene to a much lesser extent.

First, we agree with the reviewer that *AMIGO1* is probably not a causally implicated gene. It was not obtained by MR analysis, but simply by reporting the overlap between age-related gene expression QTLs and the previously 18 top hits. We have removed all overlap-based analyses, as such an approach is not statistically justifiable to yield causally implied genes.

On the other hand, based on our study, *SORT1* cannot be justified to have liver expression levels causally linked to lifespan. The reason for this is that *SORT1* has a much stronger eQTL in the liver, which does not happen to be associated with lifespan. Provided that this stronger eQTL is a valid instrument, it excludes strong causality (of course within the range of our statistical power). Using MR analysis, we could confirm the significant causal effect estimate of *SORT1* expression (in liver) on LDL. Thus while we could replicate the findings of the cited study quoted by the reviewer, it seems that the same MR approach cannot confirm any gene causally implicated in lifespan modulation in this region. We added these observations to the Discussion:

"For example, in case of the lifespan-associated rs646776 at the 1p13.3 locus (near *CELSR2*) we could not detect causally implicated mediator gene in any tissue. However, a previous study⁹ showed that this variant might exert its effect on LDL cholesterol levels through modulating the expression of *SORT1* in the liver. Our MR analysis confirms this gene as the best candidate for LDL, but not for lifespan: The reason for this is that *SORT1* has a much stronger eQTL SNP in liver (rs7528419), which is not associated with lifespan in our study. Thus unless rs7528419 is an invalid

instrument, our observation excludes *SORT1* as causal gene for lifespan, despite being a strong candidate for LDL."

Importantly, we toned down causal claims:

"While many of the obtained causal effect estimates seem plausible and in line with literature evidence, we do not claim that these causal effect estimates are unbiased, neither that no instruments violated MR assumptions. This step was necessary only to obtain sensible priors for the downstream analysis. If these estimates are wrong, our prior is less informative, yielding fewer discoveries."

Given the above, it is unclear what the purpose of this analysis is, and if it can be interpreted in the way the authors intend: that this variant changes gene expression of these genes, which then impacts lifespan (at least in one case, if this hypothesis were true, I wouldn't pick out AMIGO as the causally related gene!).

We do not pick AMIGO either, it came from a less stringent non-MR based (simple co-localisation analysis) the analysis is much more careful now and does not reveal any gene at this locus.

3b. The tissue types under consideration are also fairly scattered. In the first paragraph of the expression in model organism section, the authors state that their analysis will be done in blood and brain. However, in the next paragraph, they do their analysis in liver, with no explanation. Can we learn what the authors want to infer from this experiment, given that the tissues are different? If so, why?

This is correct and we only follow-up findings in mouse brain tissue where we had robust MR estimates.

3d. Furthermore, no reports for the *SULT1A1* and *CHRNA5* were reported. Does that mean that 2/3 of the lookups were inconsistent with your hypothesis?

We simply could not find any significant association for some genes in model organisms, either due to power issue, or different mechanisms. We now list all results (be it negative) in the supplement (Table S6 and Fig S10). We have found an interesting down-regulation of *SULT1A1* in mouse brain upon caloric restriction diet, which is in concordance with our observation in human:

"Notably, *SULT1A1* expression levels are down-regulated upon CR diet in mouse (GSE11291), similarly our human study showed lower expression may be linked to extended lifespan (Supplementary Figure 10)."

3e. The last section, there are several experiments from lookups the authors state themselves admit are "inconclusive". As the authors state, inconclusive aging trends were observed across different tissues in model organisms. In addition, most genes did not show signal in the correlation analysis, seen in lines 380-389.

Except for caloric restriction data sets, which is theoretically interesting, but pretty hard to interpret (and there's no statistics provided to do this).

We followed the reviewer's advice and only keep the CR experiment and the mouse expression vs lifespan in the correct (brain) tissue. The rationale for the CR experiment is that CR lowers *SULT1A1* expression, which may in turn extend lifespan. *SULT1A1* expression can also be regulated by SNPs and those eQTL SNPs are also longevity-associated with proportional effect.

3f. The data for AMIGO1 and RGS12 are pretty weak effects. Why should this result be convincing, given the issues above? How often would we see this level of difference picking a random gene?

We removed those results. We have also reran the analysis using mixed effect models, which removed most possible confounding yielding well-calibrated P-values.

3g. The only gene that appeared to have strong effects in Supplementary Figure 5 is RBM6. However, given the issue of gene selection (AMIGO1 isn't the causal gene) and the tissue discordance, it is pretty challenging to evaluate how much positive, additional evidence on longevity this study affords. The relationship between the CR diet and expression in brain or blood seems to be only source of signal; given that this is a non-trivial perturbation, this also makes interpretation incredibly challenging.

We kept RBM6 only and the CR experiment, as was suggested.

4. The overall flow of the manuscript.

4a. The message of the section "Genetics of disease, lifespan, extreme longevity and selection bias" is unclear. Overall, it read like a detour from the overall message of the paper of identifying and verifying loci associated with lifespan. How do these analyses connect to the central message or narrative the paper, which felt more like (at least my take on the paper): "disease-associations for phenotypes related to premature death associates with longevity"? If the authors believe this section is important, it should be put into a broader context and/or moved closer to the end.

We have reduced this section to a single 8-line paragraph and moved it to the end of the Results. This contributes to the overarching message of the paper how longevity, lifespan and morbidity differ in terms of genetic architecture.

4b. The connection between disease and lifespan has already been demonstrated, so we are unclear of the importance of the LD-score regression showing this, especially given its limitation as noted by the authors. A comparison with previous estimates, or clarification of the goal of the analysis, could be helpful.

The importance is to show how relatively little the 11 key diseases explain about lifespan genetics. We made it clear in the updated text:

"Second, the squared genetic correlation between lifespan and the (causal effect) weighted combination of 11 traits/diseases examined in this study is surprisingly low 5.8% (Supplementary Figure 6), but in line with the one 7.7% lifespan variance explained by the 11 traits in the multivariate MR model. Examining the effect of each of the 16 top hits on the 11 disease-related traits revealed that part of the loci exhibit antagonistic pleiotropy (Supplementary Figure 4)."

4c. Importantly, methods describing the data set used in the analysis on lines 263-275 should be explicitly stated. Moreover, what set of SNPs were used for this analysis?

We rewrote this paragraph and move closer to the presented data (Figure 4):

“Looking at Figure 4 we noticed that some of these 16 SNPs seemed to have larger effects on extreme longevity (*APOC1*, *LPA* and *FTO*). There are several explanations for this: (i) Markers related to late onset diseases (such as Alzheimer’s) are bound to be less pronounced effect earlier. (ii) Inflated extreme longevity effects can also be put down to the fact that testing the extremes of a distribution vs. the whole distribution by definition yields larger effects. (iii) Moreover, the UK Biobank study used offspring genotypes as proxy for parental genotypes, which results in estimates biased towards zero. On the other hand, we also observe a handful of SNPs (e.g. in *CHRNA5*, *LPL*) impacting lifespan only in the normal range, but with negligible effect on extreme lifespan.”

5. It seems like the approach (generally) is: identifying a subset of SNPs for testing, then boosting individuals ones given the MR data. It wasn’t immediately clear, but are the authors controlling 5% error for 72,735 test? That seems far too lenient.

We failed to explain it clearly: we generated prior distribution for all SNPs (even if 96% of them had zero mean, they had informative variance) and BFs were calculated for all 2.3M SNPs, consequently FDR was controlled at 5% including all these SNPs, not only those with non-zero prior.

We rewrote the misleading paragraph of the Discussion:

“When using non-informative priors only two loci could be identified¹⁰, but our approach revealed 16 loci associated with lifespan at 5% false discovery rate. Moreover, if we were to use the standard association P-values, but reduce multiple testing burden to only 4% of the genome that had non-zero prior assigned, still only three of the 16 loci would have been recovered at 5% FDR (Supplementary Table 7).”

We also added to the Methods that:

“For each of the 2.3 million SNPs, we can now compare the effect estimate observed in the longevity GWAS to the prior computed above.”

6. Did you include population stratification components when associating age with your SNPs?

Yes, we did. Now added to the text:

“We tested this hypothesis by modelling age as the outcome in a linear regression with regressors such as SNP allele dosage, gender, 15 ancestry principal components and batch indicator in 120,000 genetically white British participants of the UK Biobank.”

7. The MR gene expression experiment was not particularly detailed. What is the hypothesis? Were the SNPs eQTLs for multiple genes? Why was brain the focus?

We aimed to find genes whose expression may be causally modulating lifespan. For this we used genes, whose expression levels have more than 5 independent

and strong instruments in a given tissue. We carried out the experiment only at the 16 lifespan-associated loci to reduce multiple testing burden and looked at all 26 tissues available in GTEx. We did not focus on brain, but since it has the largest sample size in GTEx, this is where we had the highest power to detect strong causal effects.

Minor Comments

8. Make clear in Figure 1 that alphas are generated using all chromosomes except the one that the SNP is found on. Also, define variables in MR section of figure, or remove them.

We added this information to the legend.

9. Give a more thorough explanation of the risk factor selection process, especially what was meant when the authors say they removed traits with multivariate P-value >0.05. Was this done as part of the stepwise regression or afterwards?

This was done after the step-wise selection, it is now explained in the Methods.

10. Discuss reasons why several of the 18 SNPs didn't even replicate direction of effect in various other data sets.

We added power analysis that explains the non-replication. We performed *power calculations* to illustrate that the studies used for replication are massively underpowered, but that this is the best one can do with the currently available data. We describe it now in the results:

"Given the large sample size of the UK Biobank compared to other previous independent studies we first assessed power to replicate our findings in the combined CHARGE studies. First, we performed power calculations assuming a replication study with 14,000 samples selected from the top and bottom 5% tail of a general lifespan distribution and a SNP explaining ~1% of lifespan variability. These calculations showed that apart from the top two hits (*APOE* and *CHRNA3/5*) we have only 16% (for the *FTO* variant) to 73% (for the SNP near *BSND*) power to replicate our findings with a P-value below 0.05/16 in a study very similar to the combined CHARGE data set. For this reason we call our analysis as a *look-up*, given that strict replication is impossible due to power considerations."

It is all the more convincing that despite the low power we still managed to replicate the majority of the findings.

11. In discussion, it sounds as if alcohol consumption is known to influence lifespan. A citation for this would be helpful.

The reference is now added.

12. Supplementary figures could stand for some work:

12a. In supplementary figure 5, the difference between the red and maroon dots needs to be specified in the figure caption.

This is now redrawn exploiting only brain data (Figure S9).

12b. suppl fig 1: plots are not on the same x axis scale nor include the same features nor in the same order. Plus, no x axis label

We added axis labels and set the same axis limits, although identical axis limits are not ideal, due to some outliers present in some plots.

12c. suppl fig 3: no axis labels. if based on 1000 randomizations, how do you get p-values upwards of 1E-09?

Labels added now. The P-values are based on comparing the observed BF to 2.3 billion null BFs, this is why they can be as low as 1 divided by 2.3 billion ~1E-9. See for reference for example Bertolino et al. ¹¹.

13. It is unclear upon reading just the main text whether the authors performed the analysis on lines 387-389, or whether it appeared in the cited paper.

We performed the analysis on the publicly available expression data.

14. The supplementary tables need explanations. It was hard to interpret their results without keys.

We added now much more explicit column labels and legends to some of them in the second tab.

15. Figure 2 doesn't have an x axis label.

Added now.

16. line 288: "fitter" not exactly what you mean. Stick with ascertainment explanation (so as not to equivocate with population genetic selection).

Good point. We changed it to "in better health".

17. The multivariate method sets the intercept to zero, which assumes that the causal effects are not directionally biased, and the authors state that inclusion "did not change the causal effect estimates". Can we see the data for that?

We present these results in Supplementary Table 2.

18. It is not immediately clear what the point of the two-sentence paragraph starting at line 391 is. The authors should provide an interpretation of the stated results.

This paragraph has been removed now.

Reviewer #4

Overview

This paper identifies SNPs with a causal connection to human longevity working through twelve traits. Joint causal effects of each trait are estimated via Mendelian randomization (MR), a form of instrumental variables.

The overall framework of regression links $b_{i,t}$ from SNP i to trait t followed by links a_t from trait t to longevity is a reasonable choice. Then $\mu_i = \sum_{t=1}^T b_{i,t}a_t$ is a reasonable measure of the longevity effect of SNP i .

The paper has mathematical errors and computational errors. In most places the math needed to check up on the results is missing as well as the logic to support it. There could therefore be many additional undetected errors.

We would like point out that objectively reading the reviewer's comments, we could not identify a single mathematical/computational error. Almost all of the comments are questions, or when criticism is raised it is often due to misunderstandings. We realize that this is due to the fact that (a) the reviewer may not be as familiar with the GWAS field as with other aspects of statistics; (b) the reviewer may not have read the rest of the paper in the same depth as the Methods; (c) the reviewer may not have given an in-depth reading of the references listed in the methods (which is a lot, hence understandable); and (d) importantly, due to us not explaining the methods with sufficient clarity. We can certainly improve the last point and we did our best to stay concise, but give ample information to follow the methods.

Larger points

1. The article lacks a mathematical presentation of the way the data were handled.

There are many places where the reader just has to guess what was done. In some instances it is easy to guess, in other cases it is not really possible. Even with easy guesses, readers cannot be sure of getting them all correctly.

The notes below are in reference to the 'Online Methods' section. That section should be self contained, but is vague. Part of the problem seems to be that the description is made in a circular order where the narrative depends on facts to be revealed later on. After several readings, it is not clear what the authors have done nor why. Much of the problem is that necessary formulas are not given.

We agree that not sufficient details were provided in the Methods section, especially for someone not very familiar with the vast literature of the GWAS methodologies. We first provide now a concise summary of the methods at the beginning of the Results section:

For each SNP, we compute a novel test statistic, based on a Bayes Factor. With simulations, we draw a large sample of how these statistics would be distributed if the null hypothesis were true for every SNP, i.e. no effect of any SNP on lifespan. This allows us to compute a P-value for each observed test statistic, comparing the observed test statistic to the large sample of "null" test statistics. In this framework, the goal is to compute a novel test statistic which is more powerful than

the conventional lifespan Z-statistic of $\frac{\hat{\beta}}{\sqrt{\text{var}(\hat{\beta})}}$. Given two point hypotheses, the likelihood ratio test is the most powerful test¹². We use a Bayes Factor as the likelihood ratio. For each SNP, the two hypotheses of interest are the null hypothesis of no effect on lifespan and an alternative hypothesis, which we refer to as a *prior*. The prior is a prediction of the lifespan effect size, based on the effect sizes of the same SNP for other disease-related traits and an (out-of-sample) estimate of the causal effect of these traits on lifespan. The steps taken, in order, are: 1) Impute Z-statistics for all SNPs of interest in lifespan and in other traits of interest. 2) Estimate the causal effects of traits on lifespan using Mendelian randomization. 3) Build the prior for each SNP. 4) Compute the BF for each SNP. 5) Sample the 'null' BFs. 6) Compute P-values by comparing the observed BFs to the 'null' BFs. All these steps are explained in details in the **Methods**.“

Second, we have added now all equations that are necessary to understand and implement our approach.

(a) Line 601 refers to Martingale residuals. Say exactly which ones are used. The book by Therneau and Grambsch mentions four kinds. What fraction of the lifetimes were missing and had to be estimated? Plugging in guesses for 5% of the lifetimes would have very different implications than plugging in guesses for 80% of them.

We did not specify such details on purpose, as this part of the study is identical to Joshi et al. 2016¹⁰. Given the space limitation we face and that all study details can be looked up in that study, we decided to limit ourselves to the most crucial details for the wide audience. In brief Joshi et al used the Martingale residuals defined as follows:

$$\hat{M}_i = \delta_i - \hat{\Lambda}_0(\tau_i) e^{\hat{\gamma}_1 Z_{i1} + \dots + \hat{\gamma}_k Z_{ik}}$$

where δ_i and τ_i are the event indicator (1—died, 0—survived at the end of follow-up) and follow-up time of the *i*th individual, $\hat{\gamma}_1 \dots \hat{\gamma}_k$ are the estimates of the coefficients of the covariates Z_{i1}, \dots, Z_{ik} in the model from equation

$$h(x) = h_0(x) e^{\beta X + \gamma_1 Z_1 + \dots + \gamma_k Z_k}$$

We had to plug in the “guesses” for 23% of father’s age of death and 40% of mother’s age of death. Again, this is identical data to Joshi et al. published in the same journal, we do not think that such details need to be reintroduced in our paper again.

(b) Introduce mathematical notation for the Biobank analysis (lines 598–613). Say what the effects were. Maybe estimates of SNP effect on parental longevity? The meta-analysis treats the mothers’ and fathers’ longevity as independent. Is that

reasonable? Their environments would be strongly correlated. Smoking habits (or second hand smoke) would affect both. Using the average of the parents' ages would eliminate the need for meta-analysis and an assumption of independence. That might be better.

The entire analysis described between lines 598-613 is identical to that of Joshi et al. 2016. We only gave a brief recap of how it was carried out and do not think that all details need to be explained again.

The effect size for maternal and paternal lifespan association had a correlation of 0.11 across all SNPs. This is not unexpected, exactly as the reviewer points out it is probably due to the shared environment. Assuming that the majority of SNPs have no detectable effect on lifespan, this is a reliable estimate of the correlation between the test statistics under the null. The work by Lin and Sullivan⁴ generalize the meta-analysis to correlated effect estimates. The optimal weights are defined in Eq. (4) of that paper. In case of similar SE of the two estimates, the weights remain equal, but the variance of the combined effect size inflates by a factor of (1+rho), where rho is the correlation of the effect estimates under the null (0.11 in our case). We added a brief reference to the meta-analysis method to the Methods section. Again, this is standard in the GWAS field and we do not see any reason to explain it more.

"In our study we used the combined association results for both parents via inverse-variance weighted fixed effect meta-analysis adjusted for correlated test statistics⁴. The correlation between the maternal and paternal lifespan effect sizes was observed to be 0.11 under the null."

Lines 615–622. What is the extent of the overlap between the two replication studies? Give the number of common cases as well as the numbers in each study. How exactly did the combined GWAS adjust for that overlap? Use mathematical notation.

The exact same way as above, we can calculate the combined effect. The only difference is that this time the correlation between effect estimates emerges from the fact of having overlapping samples.

To ensure that the estimate was correct, we performed this analysis two ways: once when estimating the correlation directly from the test statistics (as before) and once when estimating it from the actual sample overlap (since the correlation was due to sample overlap). The EU90 (Deelen et al.) study has 5,406 cases and 15,112 controls. The CHARGE (Broer et al.) study has 7,729 cases and 16,121 controls. The effective sample sizes (see e.g. Han & Eskin¹³), $\frac{1}{\frac{1}{\#cases} + \frac{1}{\#controls}}$, are 3981.6 and 5224.3 respectively.

This allows us to convert each Z-statistic to a standardized effect estimate $\hat{\beta}_i^{EU} = \frac{z_i^{EU}}{\sqrt{3981.6}}$ and $\hat{\beta}_i^{US} = \frac{z_i^{US}}{\sqrt{5224.3}}$; this conversion places the two estimates on the same 'scale', with known variance, and they can therefore be meaningfully combined. Two studies ("Rotterdam Study 1" and "Rotterdam Study 2") are present in both meta-analyses (EU and CHARGE), with effective sample sizes of 512.5 and 48.3 respectively,

for a total effective sample size of 560.8 in the overlap. The resulting theoretical correlation (0.143, from Eq 7 of Lin and Sullivan⁴) is similar to the observed correlation between the two sets of Z-statistics (Pearson correlation of 0.152, Spearman correlation of 0.146). With this correlation and the known variances of the two estimators, we know the covariance matrix and can compute the optimal weights (their Equation 4). These weights give us a single joint estimator (their Equation 3) with known variance (their Equation 5), which can then be converted to a z-statistic and p-value, which we use as the combined CHARGE longevity association statistic.

- (c) Lines 648–653. Were summary statistics imputed for disease GWAs, longevity GWAs, or both? Why was that small study of only 3781 individuals used to define correlations among 2.1 million SNPs? (Maybe it was the only one with enough sequencing.) What formula was used to do the imputation? The imputed Z values could now have quite strong correlations among themselves. How does the later analysis account for their dependence? Maybe highly correlated Z's never co-occur because of LDA thinning, or maybe correlated Z's get treated as if they were independent, double counting information.

The Z-statistics for 2.3 million HapMap SNPs were imputed in all GWASs, disease and lifespan, where the Z-statistic was not already present. Many publicly available reference panels have fewer than 1,000 individuals, therefore 3,781 is quite large. The formula for imputation has now been made more explicit in our latest draft. The imputed Z-statistics will have strong correlation, however 'true' Z-statistics will also have strong correlation due to linkage disequilibrium and this is a very standard issue for all GWASes. The imputation method does not introduce any extra correlation that must be corrected for. Summary statistic imputation is a standard procedure in the GWAS field and a plethora of methods exists (e.g. ¹⁴), it has been shown to be very robust already for reference panel sizes of 500 samples and does not alter pair-wise correlation between imputed test statistics.

In our Methods section, we now show how the MR estimation could take account of this correlation, however we use only well-pruned SNPs for this estimation and therefore correlation between Z-statistics does not arise here.

- (d) Lines 658–686. Instrumental variables can be tricky to use and several authors have pointed out problems with using them in Mendelian randomization. There are also multiple notions of MR (sometimes describing linkage equilibrium, sometimes describing equal probability of inheriting maternal or paternal alleles). This section should lay out in mathematical notation what the response variable is, what the predictor is, what the instrumental variable is, and then explain why the regression coefficient could be causal in this case. It should also address any reasons why causality might not hold.

We have substantially expanded the Methods section to answer all these questions. By definition, the instruments for MR are SNPs that strongly associate ($P < 10^{-5}$) with

the risk factors (predictors). We then demonstrate in the Methods that the regression coefficients are (under some conditions) equivalent to the 2-stage least square MR estimate. As often, it is very difficult to ensure that all MR assumptions hold. Instrument strength is ensured by the selection of the SNPs. We also examined if the computed priors are robust to the selection of instruments (Results):

"In order to provide further evidence that these causal estimates yield robust priors, we performed MR using instruments only from chromosomes 1 to 6 and another MR analysis with instruments only from chromosomes 7 to 22. Although the selected traits did not perfectly agree (Supplementary Table 2), both the causal effect sizes and the resulting priors were highly concordant (Supplementary Figure 2)."

We also verified whether any of the instruments has effect on lifespan that is independent from any of the 11 selected disease-related traits, i.e. violating the third MR assumption (Results):

"In addition, we examined the MR residuals to ensure that there was no instrument with indication to violate the third MR assumption, i.e. that the instrument must be independent of disease-lifespan confounders. If such violation occurs, the instrument would have a disease-independent (pleiotropic) impact on lifespan and hence the observed SNP-lifespan effect would deviate substantially from the prior effect. We observed only a slight deviation for one instrument (a SNP near *APOE*, rs7256200) on the residual qq-plot for all the 22 leave-one-chromosome-out estimation (Supplementary Figure 3). Due to the large number of instruments, excluding this single SNP does not change noticeably the results."

Is the causal association established for the coefficients a , for the coefficients b , or both? For instance, the causal effect of education presumably includes any downstream consequences of education such as higher income. How do we know that education is the causal variable instead of something upstream of education that correlates with it?

The effect of the SNP on the trait ($b_{t,i}$ in the original manuscript, $\Gamma_{t,i}$ in the revised m/s) is by definition causal (or in strong LD with a SNP that is causal) – this is widely accepted in the GWAS literature. We are born with our DNA sequence, if anything correlates with a SNP dosage it can only be the consequence of the SNP (neither confounding, nor reverse causality is possible).

Education is a very good example, indeed. We added some discussion on it in the revised Discussion section. If a variable, such as high income, a consequence of education and that is casual, education can be viewed as causal, as in a hypothetical randomized control trial by improving education would lead to higher income, which in turn would improve lifespan. Of course, would have been better to include GWAS results on income into the causal inference to improve our priors, but no such robust study is publicly available to date. On the other hand if there is another variable (e.g. being born in an area with no good schools and low socio-economic status) correlated with education level in a mutually causal way, it is very hard to disentangle whether

the effect we see for education is truly education or the other trait, or both. This is why we added to the discussion that

“The strong link between education level and lifespan has been seen in other studies¹⁵, but its interpretation is not straightforward. Educational attainment may have shared genetics with socio-economic status, which is a more plausible underlying cause.”

If a is causal (education causally affects longevity) but the link $b_{t,i}$ from SNP i to education (trait t) is not causal (just a correlation), how do we interpret the contribution $a_t b_{t,i}$? Is there still anything causal in it?

As explained above, SNP-trait correlation directly translates to causality (or the SNP being in LD with the causal variant).

It is usual for instrumental variables to have at least as many instruments as there are regression variables getting a causal interpretation. So what are the instrumental variables for the multivariate causal regression? (They should be named in this section where the causal claim comes.)

By definition of MR (applied to GWAS), instrumental variables are SNPs. We used various subsets of the initially selected 6,443 instrumental variables (instruments for at least one of the 58 traits). Here is the list of the number of instruments with non-zero effect on each of the 11 traits:

Study name	#instruments
Education	480
LDL	417
BMI	315
Smoking	15
CAD	67
T2D	114
Schizophrenia	611
Height	2693
Triglycerides	335
HDL	424
Fasting Glucose	64

As can be seen there are much more instruments than total traits considered. Even though we refer to these as “causal” estimates, our method does not yield false positive results. It is explained in the Results:

“While many of the obtained causal effect estimates seem plausible and in line with literature evidence, we do not claim that these causal effect estimates are unbiased, neither that no instruments violated MR assumptions. This step was necessary only to obtain sensible priors for the downstream analysis. If these estimates are wrong, our prior is less informative, yielding fewer discoveries.”

Moreover, the prior calculation can also be viewed as a prediction exercise: Comparing SNP effects on various disease traits (predictors) with those on lifespan (dependent variable) using 21 chromosomes only, we try to predict what lifespan effect we'd see for SNPs on the 22nd chromosome given the predictors.

- (e) Lines 674–681. I'm completely lost here to the point where it is hard to even formulate a question. Are the p -values on SNPs? Are they from all GWAS or just the trait ones?

We completely reworked the description to make it clearer. Each SNP for each study (both disease-related traits and lifespan) has a P-value assigned.

- (f) The paragraph refers to SNPs with non-zero effects but we never know for sure which ones have zero effects and which ones have non-zero effects. Presumably that is based on an estimate, but how exactly? There should be mathematical descriptions.

Indeed, we do not know, which ones have exactly zero effect. This is why we apply the 10^{-5} threshold on the association P-values and assume that those with $P > 10^{-5}$ have probably either zero or very small effect. This cutoff is to ensure that we do not use too weak instruments for the Mendelian randomization, which would bias the causal estimates to the observational correlation. It is safer to err on the side of caution even if we lose some instruments. We added mathematical description about the effect truncation process (i.e. setting the effect to 0 if there is not enough evidence for the opposite).

- (g) Why divide a Z statistic by an effective sample size, when Z statistics are ordinarily already normalized quantities with a standard error (including a sample size) in the denominator? Why are sample sizes misreported? Can we model the process by which they are misreported? There should be a formula for the effective sample size and an explanation of the logic behind that formula.

If the genotype and phenotype have been standardized to mean zero and variance one, the resulting linear regression effect estimate will have variance $(1 - r^2)/n$, under the null, where n is the sample size and r^2 is the explained variance of the model. Since in GWAS r^2 is typically smaller than 1%, we simplify the variance to $1/n$. The regression coefficient emerging from such a regression is referred to as the 'standardized effect estimate' and can be multiplied by \sqrt{n} to create the Z-statistic, i.e. a test statistic with mean zero and variance one under the null. The Z-statistic can then be converted to a two-sided P-value. We essentially reverse engineer the standardized effect size from the P-value and the sample size using the formula $\Phi^{-1}(P/2)/\sqrt{n}$.

The non-standardised effect size cannot be deduced, as it requires the knowledge of the variance of the disease trait and the genotype in each meta-analysis study, which we most often do not know. The 'standardized effect estimate' is useful in many

contexts as it is an unbiased estimate of the true effect and represents the square root of the explained variance (the approximation holds for small effects, typical to GWAS, otherwise the standardized effect is $\frac{r}{\sqrt{1-r^2}}$, where r^2 is the explained variance). Given two different studies of the same trait, but with different sample sizes, the 'standardized effect estimates' can be compared to each other while the two Z-statistics cannot be meaningfully compared to each other, as they are not on the same scale.

Sample sizes may be misreported as studies sometimes report the total sample size, not the one for which they have both genotype and phenotype (and covariate) data available. Sometimes cohorts report sample sizes before QC procedure. This cannot be modeled and as we point out it is not a concern to us, as the contribution of a disease trait to the prior effect size of a SNP is $\hat{b}_t \hat{\Gamma}_{it}^*$, where \hat{b}_t is the causal effect estimate for trait t and $\hat{\Gamma}_{it}^*$ is the standardised effect estimate for SNP i on trait t . If the provided sample size for trait t was n_t , but the real one is m_t , the real $\hat{\Gamma}_{it}^*$ should be $\sqrt{n_t/m_t}$ times the original $\hat{\Gamma}_{it}^*$, but the causal effect estimate would then become $\sqrt{m_t/n_t}$ times the old estimate. Thus the two factors cancel out.

- (h) Lines 683–688. Introduce a mathematical symbol for the standardized effects of a SNP pair with appropriate subscripts. Was it $b_{t,i}$ or something else? Use mathematical formulas to describe how summary statistics are used to estimate causal effects. What in fact were the summary statistics? Explain why the logic from reference 40 applies to the present situation (and address or at least acknowledge any caveats). The textual description that follows is not clear.

Indeed, the standardized effects were denoted by $b_{t,i}$. The Online Methods have been rewritten, with consistent notation. The standardized effect estimate of SNP i on trait t is now denoted by $\hat{\Gamma}_{it}$. By summary statistics we mean the Z-statistic from the linear regression.

We added now the definition of “summary statistics”:

“Note that through the whole paper we utilise only association summary statistics, i.e. the Z-statistics from the regression (coefficient estimate divided by its standard error), and not actual genetic data, except for the UK Biobank.”

Reference (40) [Burgess et al 2013 Gen Epi] is a pioneering paper in the field of Mendelian Randomisation applied to the case when only summary statistics are available, but not the actual genetic and phenotypic data. As we now describe in the Methods the two estimators yield close estimates:

“Note that Burgess et al.⁷ introduced a maximum likelihood (ML) based MR method using only summary statistics. Since the lifespan study (UK Biobank) does not share samples with the disease trait studies, our formula (Eq. 2) yields the same result conditional on the SNP-trait estimates having little variance. To make sure that our method gave sensible results, we implemented the ML-based MR method by Burgess et al.⁷ and confirmed that we obtained indistinguishable solution to their likelihood function (Eq. 1 in their study) maximisation (see **Supplementary Table 9** for the causal estimate comparison and the between-trait correlation estimates (ρ in their paper) are in **Supplementary Table 10**).”

- (i) Lines 702–703. How exactly was $\text{Var}(a)$ estimated? Was it simply the usual variance formula from the final regression after running AIC and removing nonsignificant predictors? Or did it account for the model search process involving AIC and predictor removal?

The former guess is correct: The variance of the causal effect arises from the final regression of the 11 variables. It is now fully derived in the latest manuscript. The estimate of the vector of causal effects, $\hat{\mathbf{b}}$ in the revised m/s, is $(\hat{\Gamma}_S^{*'} \hat{\Gamma}_S^*)^{-1} \hat{\Gamma}_S^{*'} \hat{\gamma}_S$, using instruments that are independent of each other, and the variance is $\text{Var}[\hat{\mathbf{b}}] \approx (\hat{\Gamma}_S^{*'} \hat{\Gamma}_S^*)^{-1}$.

- (j) The online methods should describe the multivariate out of sample MR analysis used to include or exclude each study in mathematical notation. Introduce symbols for the variables and parameters used.

The out-of-sample analysis does not involve removing studies. It involves removing instruments (SNPs) from each chromosome in turn. The simulation method we use to compute P-values, described in detail later in the manuscript, assumes that the causal effect estimates ($\hat{\mathbf{b}}_i$) are independent of the observed parental lifespan z-statistic in UK Biobank ($\hat{\gamma}_i$). The simple way to maintain this independence is to use two different lifespan studies, one to estimate the causal effects and another to provide the observed z-statistic used in the Bayes Factor. A more powerful approach to ensure this independence, which is the approach we take, is to use the largest study available and to compute the causal effect estimates 22 times, where each of the 22 chromosomes is masked in turn. We then compute the prior for a SNP using causal effects estimates derived from the other 21 chromosomes. Z-statistics are independent from chromosome to chromosome.

- (k) Lines 737–742. This material is not clear. What are the assumptions that make this a prior for a Z statistic? How is the longevity effect size γ_i computed? Is $\gamma = \mu_i$ or is it normalized some way? What is $\text{SE}(\mu_i)$? Is it the same as $\text{Var}(\mu_i)$ or is there a different meaning?

SE() has been removed and replaced with Var, for consistency.

Say clearly what the null and alternative distributions are, why they have different variances, and what randomness those variances incorporate.

We now added this clearly to the text:

“While under the null distribution the prior has zero mean and zero variance, the prior distribution under the alternative hypothesis is $N(\mu_i, \text{Var}(\mu_i))$.”

We also refer explicitly to the derivation of the resulting likelihood function integration in the numerator: [http://www.cs.ubc.ca/~murphyk/Papers/bayesGauss.pdf]

- (l) Lines 753–760. The analysis pipeline has numerous steps. The letter Z is used in several places for different quantities. Where exactly are the Z 's getting replaced by random ones, one thousand times? What steps in the analysis get recomputed? For instance, do the imputations with the new random Z s? Or are these Z s from a later stage? Introduce mathematical notation to make it precisely clear what randomization was done, what distribution the random elements came from, and what was computed with them.

It is not clear what the BH procedure was applied to or why.

To generate a sample of what the Bayes Factors would be, if the null hypothesis of zero effect on lifespan were true for all SNPs, we performed 1000 GWASs with 1000 random phenotypes to take the place of the true lifespan phenotype for the same 2.3M SNPs. This gave us samples of Z-statistics, which had the appropriate correlation between nearby SNPs due to linkage disequilibrium. The Bayes Factors are then computed with new, randomly-derived, estimates of the standardized effect on lifespan ($\hat{\gamma}_i$), while retaining the existing prior (μ_i) for each SNP i .

Under the “true” lifespan phenotype, just as under these 1000 “null” phenotypes, the priors are independent of the UK Biobank lifespan effect estimate, and therefore there is no need to regenerate the priors and we can reuse the priors. In the “null” Bayes Factors, BF_i , only the $\hat{\gamma}_i$ needs to be replaced.

This procedure ensures that the local correlation between test statistic (Z-stat) is preserved, the priors for each SNP are identical in the null and real experiment. The procedure thus simulates the genome-wide distribution of Bayes factors (2.3M x 1,000 = 2.3 billion) for GWAS studies with random outcome traits, which enables us to assess how extreme the observed BFs in this distribution. This allows turning each BF to a P-value, as done in frequentist statistics (P-value is the probability to observe as extreme or more extreme test statistic in case no associations were real). Once P-values are obtained false discovery rates can be attributed to any selection of SNPs using for example the Benjamini-Hochberg (BH) procedure. BH, as all FDR controlling procedures, are applied to test statistic P-values. We extended the text to:

“In order to define a set of SNPs that survive 5% false discovery rate (FDR) we first needed to assign P-values to the obtained BFs. As standard procedure with any test statistics (BF is a test statistic), we used a permutation-based approach to calculate P-values corresponding to each BF_i . In practice, we generated BFs genome-wide for the same 2.3 million SNPs when the lifespan Z-statistics (and the corresponding $\hat{\gamma}_i$) were replaced by Z-statistics derived from GWAS scans with 1,000 random outcomes. We then calculated the resulting BFs for each of the 1,000 null sets of Z-statistics while keeping the same priors as before ($N(\mu_i, Var(\mu_i))$ for SNP i). In formula, the Bayes factor for SNP i for the k -th random trait is defined as

$$BF_i^{(k)} = \frac{L(\hat{\gamma}_i^{(k)}; \mu_i, \text{Var}(\hat{\gamma}_i^{(k)}) + \text{Var}(\mu_i))}{L(\hat{\gamma}_i^{(k)}; 0, \text{Var}(\hat{\gamma}_i^{(k)}))}$$

where $\hat{\gamma}_i^{(k)} = \frac{1}{n} \mathbf{Z}_i' \mathbf{y}^{(k)}$ with \mathbf{Z}_i representing the standardised genotype vector for SNP i and $\mathbf{y}^{(k)}$ stands for the k -th random phenotype vector drawn from a standard normal distribution. This gave us a large set of 2.3 billion (2.3M x 1,000) null BFs emerging from the exact same SNP set with the same priors in order to compute empirical P-values for each observed BF. These P-values were then subjected to Benjamini-Hochberg step-up procedure to select the largest set that survives 5% FDR. Using the null BFs, we also estimated the per comparison error rate (PCER)¹⁷ (under weak control) corresponding to our selection of 16 SNPs with $\log BF > 2.2857$, which yielded 4.88% concordantly with the 5% FDR control. The rationale behind our FDR control procedure has been outlined and applied previously¹¹, however the estimation of the null BF distribution we proposed here is less arbitrary.”

- (m) Lines 764–766. Does this mean that the same analysis was done on this data as on the Biobank data, or that the same analysis was done as the Joshi reference did? Or maybe both.

The analysis was almost identical to what Joshi did and applied to the same UK Biobank data with identical settings (Peter Joshi is co-author of our paper, so this is ensured). As explained in the text, the only difference was that in our reanalysis, specifically done only for the 16 top hits, missing genotypes were imputed as described in the UK Biobank paper¹⁸.

- (n) Were the martingale residuals and survival times used again, or were hazard ratios used? Once again, is it reasonable to assume that the maternal and paternal data are independent? Why would there not be within family correlations in longevity, possibly for non-genetic reasons? How is $10\log(\text{HR})$ reasonable as ‘years of live’? Reference 42 is cited, but why are we to believe it applies here?

The HRs were meta-analysed, not the martingale residuals. The correlation between the maternal and paternal data is 0.11. They are combined using the same method by Lin and Sullivan as described above⁴.

- (o) How do permutations enter the picture? The main article describes permutations of Bayes factors, perhaps 2×10^9 of them. The online methods section does not talk about permutations. It is not very common to permute Bayes factors. What is the underlying logic and how does it apply to the variables in this problem?

We do not permute the BFs, but the (UK Biobank) individuals in the association analysis. The reason for this is to gather large number of null Z-statistics, which can be combined with the same priors for each SNP to yield null BFs, 1000 of them for each SNP. We expanded this section in the Methods, see quoted text above.

Is it possible that permutations were applied to Bayes factors to generate something like a permutation p -value for the Bayes factor to feed into Benjamini-Hochberg? That is quite an unusual approach if that is what was done.

Correct, that is the approach we took. At a high level, our approach is a conventional frequentist approach to compute P-values by comparing the observed test statistics (in our case, computed as Bayes Factors) to a simulation of what the test statistics would be if the null hypothesis were true. As a result of this, we can control the error rate (e.g. FDR) even if the causal effect estimates are of low quality. Low-quality estimates would lead to a loss of power, but would not compromise the error rates.

It can be quite laborious to write in a mathematically clear way, but doing so disambiguates the wording and helps to clarify implicit assumptions as well as expose errors. Significant time savings can be had by doing that in LaTeX.

We wish we could have used LaTeX too, would have been much easier, but Nature journals do not accept LaTeX version for final publication.

Some of the problems above, such as correlated imputations and the variance of a might not have any good solution in the statistical literature. Instead of an internal proof of validity, one can then appeal to external validation in other data sets that confirm the present paper's findings. It is still necessary to be precise about what was computed and why. To the extent that problems remain in the data analysis, it becomes more difficult to interpret cases where the present method fails to confirm a discovery in the literature. The failed confirmations could be due to problems in the present analysis.

We agree that validation is crucial as an extra confirmation that the method is solid. We have now included one more study in the validation and 9 of the 16 SNPs replicate with $P < 0.05/16$ in at least one of the validation studies. This is very impressive result in our view because as we have now included a power analysis to show that for most SNPs replication power is very low due to the small available replication sample size.

2. (a) Around line 729 the paper has

$$\text{Var}(\mu_i) = \text{tr}(\text{Var}(a)) + a' * a + b'_{0,i} * \text{Var}(a) * b_{0,i}$$

for $\mu_i = \sum_{t=1}^T a_t * b_{t,i}$. The article does not say what $b_{0,i}$ is here. It might be an hypothesized value for b or maybe it is about some trait 0. Maybe it is an expected value of b , but then why doesn't that depend on t ? It does not seem that this formula could be correct. The article says that a and b are independent, so they should be random for that to be meaningful. But then the expression above for $\text{Var}(\mu_i)$ itself becomes random which does not seem to fit the rest of the paper. Also it is not clear how $\text{tr}(\text{Var}(a))$ could enter the equation without being b somehow being in there too. To see why, suppose that all the $b_{t,i}$ are always zero. Then the expression above could have $\text{Var}(\mu_i) \neq 0$ even though $\mu_i = 0$ always. Some algebra shows that for independent random vectors a and b of the same dimension,

$$\text{Var}(a^T b) = E(b)^T \text{Var}(a) E(b) + E(a)^T \text{Var}(b) E(a) + \text{tr}(\text{Var}(a) \text{Var}(b)).$$

Our latest manuscript has an equation exactly like this, with \hat{b} and $\hat{\Gamma}$ taking the place of a and b .

- (b) What is meant by variance in that expression and how is it computed? For example, is variance meant to be sampling uncertainty in those quantities, or are any other sources of uncertainty included? The discussion above mentions some issues about $\text{Var}(a)$. Are there similar issues for $\text{Var}(b)$?

If all 12 GWASs (11 disease + lifespan) were repeated in a different set of people, then the Z-statistics would be different, and the estimates of the causal effects, \hat{b} would be different. The prior is a product of two random vectors, where each vector has known covariance, and therefore we can compute the variance of the product.

- (c) Explain why a and b are independent. Here a comes from a regression of longevity on traits and b comes from a regression of traits on SNPs.

The coefficients apply to the Z-statistics on chromosome c are estimated in the other 21 chromosomes. Therefore, they would be independent of each other if the GWASs were repeated again. For a SNP on chr22, a is estimated using chromosomes 1-21 and b is estimated using only this SNP on chr22. The two estimates are generated using independent data.

The fact that so many of the other formulas are simply missing makes it very likely that the analysis has many errors like this one.

We respectfully disagree with the reviewer, none of the comments revealed any errors. We agree that many more equations could have been laid out with full details - which we have rectified -, but we claim that the reviewer spotted no actual error.

3. The effects in Figure 2 are in SDs of lifespan. It would be more interpretable to use years of lifespan. Then the authors can compare their estimate of the effect on lifespan to other parts of the literature, or future researchers can compare to this paper. Similarly, readers can judge whether the given effects are large enough to be interesting but not so large as to be unreasonable.

It is conventional in many contexts to consider standardized effect sizes, where the original phenotypes and genotypes have been standardized to have mean zero and variance one. This allows for easier comparison across different studies that might use different units of measurement. The choice to sometimes use transformed variables (such as Martingale residuals instead of actual age of death in years, or log

transformation of BMI due to skewness) is to ensure model assumptions (e.g. residual normality) or improve statistical power.

To address the reviewers comment, we agree that it is a good idea to show the effect on the most interpretable outcome (years of life lost per allele) and now we have reanalyzed the UK Biobank data to derive these estimates for the 16 top hits. However, we cannot obtain effect estimates on year-lost scale, which is indeed the most interpretable, for any of the replication studies (especially since almost all of them are case-control studies). These values are inserted into Table 2 and it is clear that none of the risk alleles reduce lifespan by more than 7 months.

4. Figure 4 should have a more meaningful axis. For Biobank, the effects are mostly between .005 to .01. If that would be in years, then it would be roughly two to three days. It says that it is in units of standardized effect. If that would be standard deviation of lifetimes we get maybe 30 to 45 days if lifetimes have a standard deviation of about 15 years. Are the units comparable to those in Figure 2? If so then education level at 0.2 is about 20 to 40 times the effect of a typical SNP in Figure 4, which is interesting but we still cannot relate it to anything outside of Biobank. Both figures should use years, or possibly days, of extra life.

The vertical axis of Figure 4 should be given an interpretation. The label says it is a standardized effect for people over 90 versus controls. What does 0.02 mean here? Maybe the SNP gives a 2% greater chance of living to 90. Or maybe if people have (hypothetically) a 30% chance of living to 90 then people with the SNP have a $30 \times 1.02 = 30.6$ percent chance. Or maybe the odds ratio

$$\frac{\Pr(\text{live to 90}|\text{minor LPA}) \Pr(\text{live to 90}|\text{major LPA})}{\Pr(\text{don't live to 90}|\text{minor LPA}) \Pr(\text{don't live to 90}|\text{major LPA})} = 1.02,$$
or maybe the sampling uncertainty of the odds or odds ratio or log odds ratio is involved.

It should be possible for a reader to judge whether 0.02 is too small to be interesting, too large to be credible, or somewhere in between and comparable to other findings.

As mentioned above, we cannot turn the estimates to years of life lost for any other study except the UKBiobank, which we provide in Table 2. These effects range between 1 and 7 month of life lost for the top 16 SNPs. For the other studies we do not have access to the individual study data, hence cannot calculate it: the combined CHARGE + EU90 study presented in Figure 4 (y-axis) represents the combination of two case-control studies (with slightly different case and control definition), we could report ORs, but we do not have that available in the EU90 study, also those are not comparable to the UKBB study effects on the Martingale residuals or Cox-regression HRs. This plot is only meant to show the concordance in effect direction and relative strength - admittedly standardized effects are hard to interpret.

5. At line 240, 13 of 18 SNPs agree in effect direction, with $p = 0.015$. That matches a binomial tail probability assuming that a direction match has probability 0.5 independently for each of the 18 SNPs. This is a plausible assumption, but might not be right. Maybe there are correlations that make the SNPs match in sign more often than

50% of the time. The authors should check the fraction of matching signs. If it would be 56% then getting 13 matches would only yield $p = 0.049$. This sign match does not appear to be a strong confirmation.

There is absolutely no correlation, as the set of SNPs are selected from the FDR hits after pruning for their correlation (on top of it, they are more than one Mb away in the genome).

In our latest manuscript, we have only 16 hits instead of 18 as we removed a non-European MDD study from the causal effect estimation, and we still have 13 (out of 16) in the correct direction. As a result, the significance only has increased.

6. Lines 385–389. Describe how these p -values were computed. Name the method and the logic to support that method. For instance with RGS12, p -values of 0.095 and 0.4 combine to 0.037. So something that is insignificant twice in a row (and once not even close) is now significant.

Checking with Fisher's statistic $\Pr(\chi^2_{(4)} \geq -2 \log(.095 \times .04)) \doteq 0.16$ which is not significant. Maybe the authors just multiplied the two p -values which does not yield a correct p -value (that result is close to the reported one and might arise by rounding error). The scatterplot in Supplemental Figure 5 does not show a very strong pattern for RGS12 and Fisher's p -value is not very small.

This analysis is gone, as we do not follow up those genes in liver, but the three focal genes in the correct tissue (brain). Still, we insist that the computation was correct. First, we stated in those lines 385–389 that we computed a one-sided P-value, thus it is the half of the two-sided P-value given the correct direction (positive correlation). Second, our computation was based on inverse-variance weighting of the effect size, not by the Fisher omnibus test (which is less powerful and ignores directionality): $\rho_1 = 0.287$, $SE_1 = 0.172$; $\rho_2 = 0.154$, $SE_2 = 0.182$. The inverse-variance weighting meta-analysis gives then $\rho_{meta} = 0.224$, $SE_{meta} = 0.125$, $P_{meta} = 0.073$. Hence the one-sided meta-P-value is 0.0364, exactly as we stated in the original paper.

Smaller points

1. line 592: thorough → through

Thank you, corrected.

2. line 382–383: I am guessing that for 3 strains of mice there was no high fat treatment arm. Isthathow it went?

Yes, exactly, but we do not use the BDX lines in the revised manuscript, as we replicated expression findings in the same tissue as the human expression MR, which was brain.

References

1. Fortney, K. *et al.* Genome-Wide Scan Informed by Age-Related Disease Identifies Loci for Exceptional Human Longevity. *PLoS Genet* **11**, e1005728 (2015).
2. Kang, H.M. *et al.* Efficient control of population structure in model organism association mapping. *Genetics* **178**, 1709-23 (2008).
3. van Vliet, P., Oleksik, A.M., van Heemst, D., de Craen, A.J. & Westendorp, R.G. Dynamics of traditional metabolic risk factors associate with specific causes of death in old age. *J Gerontol A Biol Sci Med Sci* **65**, 488-94 (2010).
4. Lin, D.Y. & Sullivan, P.F. Meta-analysis of genome-wide association studies with overlapping subjects. *Am J Hum Genet* **85**, 862-72 (2009).
5. Bulik-Sullivan, B. *et al.* An atlas of genetic correlations across human diseases and traits. *Nat Genet* **47**, 1236-41 (2015).
6. Consortium, G.T. Human genomics. The Genotype-Tissue Expression (GTEx) pilot analysis: multitissue gene regulation in humans. *Science* **348**, 648-60 (2015).
7. Burgess, S. & Thompson, S.G. Multivariable Mendelian randomization: the use of pleiotropic genetic variants to estimate causal effects. *Am J Epidemiol* **181**, 251-60 (2015).
8. Marioni, R.E. *et al.* Genetic variants linked to education predict longevity. *Proc Natl Acad Sci U S A* **113**, 13366-13371 (2016).
9. Musunuru, K. *et al.* From noncoding variant to phenotype via SORT1 at the 1p13 cholesterol locus. *Nature* **466**, 714-9 (2010).
10. Joshi, P.K. *et al.* Variants near CHRNA3/5 and APOE have age- and sex-related effects on human lifespan. *Nat Commun* **7**, 11174 (2016).
11. Bertolino, F., Cabras, S., Castellanos, M.E. & Racugno, W. Unscaled Bayes factors for multiple hypothesis testing in microarray experiments. *Statistical Methods in Medical Research* **24**, 1030-1043 (2015).
12. Neyman, J. On the problem of the most efficient tests of statistical hypotheses. *Philosophical Transactions of the Royal Society of London Series a-Containing Papers of a Mathematical or Physical Character* **231**, 289-337 (1933).
13. Han, B. & Eskin, E. Random-effects model aimed at discovering associations in meta-analysis of genome-wide association studies. *Am J Hum Genet* **88**, 586-98 (2011).
14. Pasaniuc, B. *et al.* Fast and accurate imputation of summary statistics enhances evidence of functional enrichment. *Bioinformatics* **30**, 2906-14 (2014).
15. Marioni, R.E. *et al.* Genetic variants linked to education predict longevity. *Proc Natl Acad Sci U S A* (2016).
16. Wen, X. Robust Bayesian FDR Control Using Bayes Factors, with Applications to Multi-tissue eQTL Discovery. *Statistics in Biosciences*, 1-22 (2016).
17. Dudoit, S., Shaffer, J.P. & Boldrick, J.C. Multiple hypothesis testing in microarray experiments. *Statistical Science* **18**, 71-103 (2003).

18. Sudlow, C. *et al.* UK biobank: an open access resource for identifying the causes of a wide range of complex diseases of middle and old age. *PLoS Med* **12**, e1001779 (2015).

Reviewer #1

The revised manuscript addressed all of my concerns and is suitable for publication in Nature Communications.

We thank the reviewer for this evaluation.

Reviewer #2

The authors should be aware that the 90PLUS cohort used in the Fortney et al. paper is exactly the same sample as was used for the EU longevity study. Hence, it might not be the best option to mention them both in the text and Table 2 (and consider them as independent).

Indeed, the 90PLUS study is the same as the EU longevity study. This is why we did not meta-analyse the outcomes from the Fortney et al. ¹and the original study (Deelen et al. ²). The Fortney et al. study is a disease informed GWAS (iGWAS) based on the summary statistics of the Deelen et al. study, so their results partially overlap. We therefore used off-target SNPs to estimate the null distribution of the Fisher's combined P-value (sum of $-\ln(P)$) distribution to account for overlap of traits and samples. We added a new section to the Methods:

"Combining evidence from replication studies

We used the following UK Biobank-independent studies for replication: the CHARGE study³, the 90PLUS longevity study², a genome-wide survival study⁴ and a recent disease-informed extreme longevity GWAS (iGWAS)¹ including two cohorts (NECS and 90PLUS). As effect sizes were available for the first three studies, we combined the summary statistics via the method of Lin et al.⁵ accounting for sample overlap, and then converted the resulting summary statistics into one-sided P-values. Since effect sizes and directions are unknown for the iGWAS, we used the Fisher's combined probability test⁶ to meta-analyse the aggregated P-values from the first three studies with the P-values from the two iGWAS cohorts. To account for the sample overlap we used 236,383 off-target SNPs (available/imputed in all studies) to estimate the null cumulative density function for the sum of the $-2\ln(P)$ values, instead of using a chi-square distribution as done in the standard Fisher method. Note that for the iGWAS NECS study we used the best available proxies for 14 of the 16 SNPs (r^2 ranging 0.25-1, see Supplementary Table 12)."

Reviewer #5

This is a very thoughtful approach and an innovative way to integrate prior knowledge on SNP-disease associations and has the potential to greatly influence future work in the field. I found the overall approach and initial outcomes to be quite interesting and well described. The authors are able to

demonstrate global replication success in the CHARGE/EU longevity studies.

We thank the reviewer for this very positive comment.

However, they are not able to comment on the replicability of individual SNP associations. All of the content in the replication, expression, and model system analyses are interesting but fail to definitively demonstrate the validity of the findings.

As the reviewer pointed out below, it is extremely hard to find a replication study comparable to the size of the UK Biobank. We provided power calculations to demonstrate that with the currently available independent studies we have only 16-73% (*post hoc*) power to replicate these findings with P-value below 0.05/16. Most publications based on the UK Biobank face the same issue, e.g. Joshi et al.⁷. We felt the only confirmation we can perform is to (a) look up our findings in all the largest available published studies and (b) find the most relevant animal experiments published (dietary restriction, see below) or done in-house (mouse lifespan vs expression at 72 days of age). We now provide more rigorous assessment of replication combining all replication studies accounting for their overlap.

The replication results are promising but remain overstated. I appreciate the difficulty in identifying appropriate populations for replication meta-analysis given the uniqueness of the UK biobank – and I think that the approach of looking across multiple similar studies is reasonable. However, the results read as a long laundry list of partial replications in disparate studies – including the section on replication in your analysis of other published longevity findings and the expression analyses - and it is quite difficult to evaluate the overall replication of any individual SNP association. Many of the relevant statistics are nicely provided in Table 2 and, from this, one can make some inferences regarding the replication across multiple sources of each of these findings – but this is not evident from the text, which talks in general terms about the pairwise numbers of replicated findings between studies. These observations are not formally combined with statistics or even presented in a manner that enables one to understand how frequently each SNP association replicated across the multiple analyses that were performed.

We have now formally meta-analysed all the presented studies, accounting for sample overlap-driven summary statistic similarity [see above reply to Reviewer 2]. We now only present the combined results in Table 2 and the detailed (per replication study P-values) in Supplementary Table 3. Also simplified the description of the replication results as follows:

“The 16 significant SNPs were tested against summary statistics from all available replication studies: the CHARGE study³, the EU longevity², a genome-wide survival study⁴ and a recent disease-informed extreme longevity GWAS¹ including two cohorts. We combined evidence at the

P-value level and combined P-values using the Fisher's method while accounting for study overlap (see **Methods**). Notably, 11 of the 16 SNPs can be declared significant based on its combined replication P-value at 5% FDR (**Table 2** and **Supplementary Table 3**)."

The gene expression analysis (and methylation analysis?) is not sufficiently described – in the results or in the methods – with enough detail to understand what was done or to evaluate its legitimacy. This is especially true given the general readership of *Nature Communications*.

Given that the gene expression and methylation experiment did not yield any meaningful insight we decided to be brief, also to follow the suggestion of Reviewer #3. We have now slightly expanded it hoping that it is understandable for the broad audience of *Nature Communications*:

"We applied MR to test whether the expression of certain genes at the 16 lifespan-associated loci may be causally driving lifespan. The rationale behind MR is that if a gene expression in a given tissue is causally related to lifespan, SNPs modulating its expression must have an effect on lifespan proportional to the effect on the expression. We investigated 91 gene-tissue pairs (equivalent to 57 independent tests) at the 16 loci that had more than five independent eQTL SNPs in a GTEx⁸ tissue to be sure that the signal is not driven by pleiotropy. Three genes (*SULT1A1*, *CHRNA5* and *RBM6*) showed significant negative causal effects (all in brain) on lifespan, i.e. lower expression extending lifespan (see **Supplementary Table 5**).

If the expression of a gene correlates with chronological age, it may be an indicator for the biological age and may be linked to lifespan⁹. This hypothesis was formally confirmed for age-associated methylation levels¹⁰. If the expression of age-correlated genes tends to modulate lifespan, SNPs regulating their expression may influence lifespan through the altered expression. To test this hypothesis, we explored whether SNPs that at least mildly regulate peripheral blood expression level¹¹ ($P < 10^{-4}$) of age-associated genes⁹ are enriched for lower than expected longevity P-values. This set of SNPs, however, showed no significant enrichment in low P-values. Similarly, we found no significant enrichment of lower than expected lifespan P-values among SNPs that are methylation QTLs¹² ($P < 10^{-4}$) for at least one of the 353 age-associated methylation CpG sites¹³. These results indicate that many of the age-correlated biomarkers may simply change in response to aging or are driven by mechanisms that underlie aging. "

I find the caloric restriction study to be too speculative – caloric restriction is not a proxy for longevity and I find these results to be over-interpreted.

We agree with the reviewer that in humans the impact of CR on longevity cannot be fully established yet. However, calorie restriction (CR), defined as a reduction of about 20-40% of the regular caloric intake, can extend lifespan in a large range of organisms along the evolutionary scale. In *C. elegans*, different regimens of CR lead to extended lifespan via processes alternatively dependent on AMPK/aak-2 and FoxO/daf-16 or other nematode longevity pathways, including ubiquinone synthesis¹⁴. Similarly, CR extends lifespan in natural populations of *D. melanogaster*, independently of the adaption to laboratory growth conditions¹⁵. In rodents, the first

evidence the CR could positively impact on lifespan and healthspan emerged already in the 1930s¹⁶, and it was shown to increase mouse lifespan even when started in adulthood¹⁷. Recently, in-depth reanalysis of the effects of CR in nonhuman primates from two parallel studies performed in the in the late 1980s has confirmed that CR improves health and survival also in rhesus monkeys¹⁸. In humans, evidence based on the improvements of primary and secondary aging traits, such as fasting insulin level, body temperature, metabolic rates, inflammation markers, elasticity of the cardiac left ventricle, indicates that CR exerts beneficial effects in humans as in laboratory animals, and prevents the development of age-associated health complications^{19,20}.

We added a shorted version of this argument to the manuscript:

“Caloric restriction (CR), defined as a reduction of about 20-40% of the regular caloric intake, can extend lifespan in a large range of organisms (*C. elegans*,¹⁴ *D. melanogaster*¹⁵, mice¹⁷, rhesus monkeys¹⁸). In humans, the evidence is less direct and based on the improvements of primary and secondary aging traits (fasting insulin level, body temperature, metabolic rates, inflammation markers, elasticity of the cardiac left ventricle)^{19,20}. We therefore asked whether CR exerts its effect in animal models through inducing expression change in any of the three focal genes.”

It would be interesting to consider how this approach - integrating SNPs based on disease associations - can help to understand genetic cause of co-morbidities (or interactive effects across SNPs causal to different diseases) and how these might influence lifespan.

We agree that our approach facilitates the discovery of SNPs that tend to act through disease predisposition. This is a central topic of our future research. While impacting co-morbidities played an important role in the discovery of these 16 SNPs, we would like to point out that they have co-morbidity independent pleiotropic effects on lifespan. Still to give more details on the pleiotropic impact of these SNPs on co-morbidities, we have now moved (what used to be) Fig S4 to the main manuscript (and moved the original Fig 4 to the supplement, as it did not yield insight beyond what is already stated in Table 2). The new Figure 5 reveals various pleiotropy patterns, which we now describe in the Discussion:

“Third, most of the 16 loci exhibit pleiotropic effects (Figure 5): For example, the *APOE/APOC1* variant (rs4420638) decreases lifespan primarily through (beyond the well-known effect on Alzheimer predisposition) increasing LDL levels, but surprisingly protects from type 2 diabetes. The *CHRNA3/5* SNP (rs951266) mainly impacts lifespan through smoking and schizophrenia predisposition; the *SNX29* SNP rs729583 seems to have little impact on most of the 11 traits; the *FTO* SNP rs9939973 essentially exerts its effect on BMI and its downstream traits.”

Furthermore, these 16 SNPs do not show interaction effect.

We have also looked at the pair-wise genetic correlation of the 11 co-morbidities and lifespan (see heatmap below) to illustrate how much genetics they share with lifespan globally. However, we feel that we lack clear interpretation of such a global result and hence decided not to include it in the manuscript. It is subject to future investigations to see the network of conditional causal effects of these traits on each other and on lifespan.

References

1. Fortney, K. *et al.* Genome-Wide Scan Informed by Age-Related Disease Identifies Loci for Exceptional Human Longevity. *PLoS Genet* **11**, e1005728 (2015).
2. Deelen, J. *et al.* Genome-wide association meta-analysis of human longevity identifies a novel locus conferring survival beyond 90 years of age. *Hum Mol Genet* **23**, 4420-32 (2014).
3. Broer, L. *et al.* GWAS of longevity in CHARGE consortium confirms APOE and FOXO3 candidacy. *J Gerontol A Biol Sci Med Sci* **70**, 110-8 (2015).
4. Walter, S. *et al.* A genome-wide association study of aging. *Neurobiol Aging* **32**, 2109 e15-28 (2011).
5. Lin, D.Y. & Sullivan, P.F. Meta-analysis of genome-wide association studies with overlapping subjects. *Am J Hum Genet* **85**, 862-72 (2009).

6. Fisher, F.M.a.R.A. Questions and answers. *The American Statistician* **2**, 30-31 (1948).
7. Joshi, P.K. *et al.* Variants near CHRNA3/5 and APOE have age- and sex-related effects on human lifespan. *Nat Commun* **7**, 11174 (2016).
8. Consortium, G.T. Human genomics. The Genotype-Tissue Expression (GTEx) pilot analysis: multitissue gene regulation in humans. *Science* **348**, 648-60 (2015).
9. Peters, M.J. *et al.* The transcriptional landscape of age in human peripheral blood. *Nat Commun* **6**, 8570 (2015).
10. Chen, B.H. *et al.* DNA methylation-based measures of biological age: meta-analysis predicting time to death. *Aging (Albany NY)* **8**, 1844-1865 (2016).
11. Westra, H.J. *et al.* Systematic identification of trans eQTLs as putative drivers of known disease associations. *Nat Genet* **45**, 1238-43 (2013).
12. Grundberg, E. *et al.* Global analysis of DNA methylation variation in adipose tissue from twins reveals links to disease-associated variants in distal regulatory elements. *Am J Hum Genet* **93**, 876-90 (2013).
13. Horvath, S. DNA methylation age of human tissues and cell types. *Genome Biol* **14**, R115 (2013).
14. Greer, E.L. & Brunet, A. Different dietary restriction regimens extend lifespan by both independent and overlapping genetic pathways in *C. elegans*. *Aging Cell* **8**, 113-27 (2009).
15. Metaxakis, A. & Partridge, L. Dietary restriction extends lifespan in wild-derived populations of *Drosophila melanogaster*. *PLoS One* **8**, e74681 (2013).
16. McCay, C.M., Crowell, M.F. & Maynard, L.A. The effect of retarded growth upon the length of life span and upon the ultimate body size. 1935. *Nutrition* **5**, 155-71; discussion 172 (1989).
17. Weindruch, R. & Walford, R.L. Dietary restriction in mice beginning at 1 year of age: effect on life-span and spontaneous cancer incidence. *Science* **215**, 1415-8 (1982).
18. Mattison, J.A. *et al.* Caloric restriction improves health and survival of rhesus monkeys. *Nat Commun* **8**, 14063 (2017).
19. Heilbronn, L.K. *et al.* Effect of 6-month calorie restriction on biomarkers of longevity, metabolic adaptation, and oxidative stress in overweight individuals: a randomized controlled trial. *JAMA* **295**, 1539-48 (2006).
20. Holloszy, J.O. & Fontana, L. Caloric restriction in humans. *Exp Gerontol* **42**, 709-12 (2007).